# Full-Batch Gradient Descent Outperforms One-Pass SGD: Sample Complexity Separation in Single-Index Learning

**Filip Kovačević** [1]  **Hong Chang Ji** [2]  **Denny Wu** [*3]  **Mahdi Soltanolkotabi** [*4]  **Marco Mondelli** [*1]

## Abstract

It is folklore that reusing training data more than once can improve the statistical efficiency of gradient-based learning. While this phenomenon has been extensively studied in linear regression, the benefit of multi-pass gradient descent (GD, which reuses all the data) over one-pass stochastic gradient descent (online SGD, which uses each data point only once) is not well-understood in nonlinear and non-convex settings, except for a loss modification mechanism achieved by the first two passes on the data. In this work, we consider learning a $d$-dimensional single-index model with a quadratic activation, for which it is known that one-pass SGD requires $n \gtrsim d \log d$ samples to achieve weak recovery. We first show that this $\log d$ factor in the sample complexity persists for full-batch spherical GD on the correlation loss; however, by simply truncating the activation, full-batch GD exhibits a favorable optimization landscape at $n \simeq d$ samples, thereby outperforming one-pass SGD (with the same activation) in statistical efficiency. We complement this result with a trajectory analysis of full-batch GD on the squared loss from small initialization, showing that $n \gtrsim d$ samples and $T \gtrsim \log d$ gradient steps suffice to achieve strong (exact) recovery.

## 1. Introduction

A fundamental question in machine learning is when and how reusing training data improves the statistical efficiency of gradient descent (GD). On the empirical side, recent works compare the scaling laws of single-epoch versus multi-epoch pretraining of large language models as training enters the "data-constrained" regime (Muennighoff et al., 2023; Yan et al., 2025). On the theoretical side, existing results have mostly focused on linear regression, where multi-pass stochastic gradient descent (SGD) can achieve better loss scaling than its one-pass counterpart under "hard" source conditions (Pillaud-Vivien et al., 2018; Lin et al., 2025). For *nonlinear* models and non-convex settings, however, the presence and mechanisms of analogous multi-pass speedups remain poorly understood. One notable exception is the study of the SQ-CSQ gap in single-index learning, where reusing each data point twice is shown to induce a form of label preprocessing that can improve the sample complexity of SGD (Dandi et al., 2024b; Lee et al., 2024; Arnaboldi et al., 2024). That said, this explanation is insufficient for two reasons: (i) these works only consider repeating the batch exactly twice, and moreover, the same statistical gain is already present in one-pass SGD with a modified loss function; (ii) this mechanism does not apply to learning "hard" target functions for which one must analyze GD training *beyond finitely many passes* over the data to establish a statistical separation — examples include all *even* single-index models, such as phase retrieval (Fienup, 1982). Understanding the benefit of data reuse in this setting requires a refined analysis of the *multi-pass* learning dynamics over a diverging time horizon.

In this work, we study the optimization and sample complexity of gradient-based learning in a *Gaussian single-index model*, a standard sandbox for understanding non-convex feature learning in shallow neural networks (Ba et al., 2022; Bietti et al., 2022; Berthier et al., 2025). Specifically, we assume access to $n$ i.i.d. samples $(x_i, y_i)_{i=1}^n$, where

$$x_i \sim \mathcal{N}(0, I_d), \quad y_i = \sigma(\langle x_i, \theta^\star \rangle), \quad \|\theta^\star\| = 1. \quad (1.1)$$

Equation (1.1) defines a single-index model, where the label $y_i$ depends on a one-dimensional projection (specified by the target direction $\theta^\star$) of the $d$-dimensional input $x_i$. We consider an activation (or, as commonly referred to in the statistical literature, a *link function*) $\sigma : \mathbb{R} \to \mathbb{R}$ which is either quadratic or truncated quadratic; for these even link functions, the aforementioned two-pass loss-modification analysis cannot establish a sample complexity separation — see Dandi et al. (2024b, Section 3). We note that this learning problem is closely connected to *phase retrieval*

---

[*]Equal contribution  [1]Institute of Science and Technology Austria [2]Sung Kyun Kwan University [3]New York University and Flatiron Institute [4]University of Southern California. Correspondence to: Filip Kovačević <filip.kovacevic@ista.ac.at>.

*Proceedings of the 43rd International Conference on Machine Learning*, Seoul, South Korea. PMLR 306, 2026. Copyright 2026 by the author(s).

(Fienup, 1982; Candes et al., 2015; Mondelli & Montanari, 2018; Fannjiang & Strohmer, 2020). Moreover, when $\sigma(z) = z^2$, it is a special case of a quadratic neural network, whose optimization behavior has been extensively studied (Soltanolkotabi et al., 2018; Sarao Mannelli et al., 2020b; Martin et al., 2024; Ben Arous et al., 2025).

Phase retrieval provides a natural nonlinear setting to compare the sample efficiency of one-pass versus multi-pass (or full-batch) updates. First, we know that information theoretically, $n \gtrsim d$ samples are necessary and sufficient to learn this single-index target function (Mondelli & Montanari, 2018; Barbier et al., 2019). However, moving to gradient-based algorithms, prior works have proved that one-pass (online) SGD succeeds with $n \gtrsim d \log d$ samples (Ben Arous et al., 2021; Tan & Vershynin, 2023). In fact, the same sample complexity $n \gtrsim d \log d$ is required for all activations having information exponent 2, which includes any even $\sigma$. While establishing a hardness result that holds for all batch sizes and step-size schedules is challenging, it has been argued that the $\log d$ factor is *necessary* for weak recovery under an "optimal" stable step size $\eta \simeq 1/d$ — see the lower bound in Ben Arous et al. (2021, Theorem 1.4). This presents a logarithmic gap from the information theoretic limit. In contrast, in the full-batch setting where all data points are reused at each step, iterative algorithms beyond gradient descent (i.e., spectral estimators (Lu & Li, 2020; Mondelli & Montanari, 2018) then refined via approximate message passing (Donoho et al., 2009; Mondelli & Venkataramanan, 2021; Feng et al., 2022)) have been shown to succeed with the better sample complexity $n \gtrsim d$, and they can even achieve *strong recovery* (see (2.1) for a definition) at finite $n/d$ (Maillard et al., 2020; Mondelli & Venkataramanan, 2021). These findings motivate the following questions:

1. *Can standard full-batch GD learn* (1.1) *with linear-in-$d$ sample complexity?*

2. *If so, what iteration complexity is required to achieve weak and strong recovery?*

Notably, most existing analyses of full-batch GD (and its variants) still require $n \gtrsim d \operatorname{polylog} d$, and therefore do not imply a statistical separation from one-pass SGD (Netrapalli et al., 2013; Candes et al., 2015; Sun et al., 2018; Chen et al., 2019). If we focus on the squared loss and quadratic activation, there is evidence that the optimization *landscape* can be benign at finite $n/d$ in certain regions (Sarao Mannelli et al., 2020b;a; Cai et al., 2023). However, this does not cover the commonly-studied setting of spherical gradient updates on the correlation loss (Ben Arous et al., 2021; Abbe et al., 2023; Damian et al., 2023), where the $\log d$ barrier is expected to be unavoidable for one-pass SGD due to the lower bound in Ben Arous et al. (2021). Furthermore, these

landscape analyses alone $(i)$ cannot sharply characterize the optimization time of GD, and $(ii)$ do not address whether strong recovery is attainable. The goal of this work is to rigorously show when full-batch GD can achieve weak and strong recovery with linear-in-$d$ sample complexity, and to characterize the corresponding iteration complexity.

### 1.1. Our contributions

We aim to develop a refined understanding of when full-batch GD achieves linear-in-$d$ sample complexity for learning the single-index model in (1.1). We first consider the same algorithmic setup as Ben Arous et al. (2021) — minimizing the correlation loss over the sphere — and present both a negative and a positive result in Section 3:

- For the quadratic activation $\sigma(z) = z^2$, we show that when $n \ll d \log d$, spherical gradient flow on the full dataset achieves only trivial performance, indicating that full-batch updates offer no statistical advantage over their one-pass counterpart.

- Next, we show that the sample complexity can be improved via a simple modification: by truncating the quadratic nonlinearity, full-batch spherical gradient flow achieves weak recovery with $n \gtrsim d$ samples, whereas one-pass SGD still requires $n \gtrsim d \log d$ for the same truncated link function.

To prove the positive result, we establish a uniform BBP phase transition (Baik et al., 2005) at finite $n/d$ for the Hessian along the optimization trajectory. This implies a benign empirical loss landscape and enables us to invoke the stable manifold theorem (see, e.g., Lee et al. (2016)) to show convergence to an informative direction with non-trivial overlap. Altogether, this provides theoretical evidence that full-batch updates with data reuse outperform online updates in a non-convex feature learning problem.

The above spherical gradient analysis based on landscape properties does not characterize the convergence rate of the learning algorithm. To sharply quantify the iteration complexity, in Section 4 we study the full-batch (Euclidean) GD dynamics on the squared loss from small initialization — a setting also inspired by recent work on gradient-based feature learning (Stöger & Soltanolkotabi, 2021; Boursier et al., 2022; Ren et al., 2025). We prove that, with a truncated quadratic activation, *strong recovery* can be achieved with $n \gtrsim d$ samples and $T \gtrsim \log d$ gradient steps. This stands in sharp contrast to the unbounded quadratic setting, where the local landscape at finite $n/d$ remains highly non-convex and thus resists a proof of exact recovery (Cai et al., 2023; Liu et al., 2024). To our knowledge, this is the first strong recovery and convergence rate guarantee for full-batch GD (without algorithmic or loss modifications) in the information theoretically optimal proportional $n, d$ regime.

## 1.2. Related work

**Learning single- and multi-index models.** The sample complexity of learning Gaussian single-index models has been widely studied in the feature learning literature. A common conclusion is that gradient-based training requires polynomial-in-$d$ many samples (Soltanolkotabi, 2017; Dudeja & Hsu, 2018; Ben Arous et al., 2021; Bietti et al., 2022; Damian et al., 2023; Glasgow et al., 2025), where the exponent depends on $k \in \mathbb{N}$ which is the *information exponent* (or *generative exponent*) of the activation function $\sigma$ (Damian et al., 2024; Joshi et al., 2025). The quadratic $\sigma$ we study, both with and without truncation, corresponds to the $k = 2$ setting: here, the population gradient flow starting from random initialization takes $T \gtrsim \log d$ time to "escape mediocrity" (Ben Arous et al., 2021; Braun et al., 2026), translating to a sample complexity of $n \gtrsim d \log d$ for online SGD. Beyond GD, the performance of spectral methods and approximate message passing in the proportional $n, d$ limit has been sharply characterized (Ma et al., 2019; Lu & Li, 2020; Maillard et al., 2020; Mondelli & Venkataramanan, 2021), along with optimal error rates and information-theoretic lower bounds (Barbier et al., 2019; Mondelli & Montanari, 2018). We note that the model (1.1) is a special case of a *multi-index model*; for this broader function class, the sample complexity of gradient-based training has also been investigated (Damian et al., 2022; Abbe et al., 2022; Dandi et al., 2024a; Collins-Woodfin et al., 2024; Oko et al., 2024; Zhang et al., 2025), as well as other algorithms for weak recovery (Zhang et al., 2022; Defilippis et al., 2025; Kovačević et al., 2025; Damian et al., 2025; Diakonikolas et al., 2025b;a). Finally, the concurrent work of Montanari & Wang (2026) employs dynamical mean-field theory to derive sharp thresholds for full-batch gradient descent to recover the first "nontrivial" subspace of the multi-index target function.

**Benefit of data reuse.** Prior work shows that in linear regression, multi-pass SGD can improve statistical efficiency when learning "hard" instances (Pillaud-Vivien et al., 2018; Carratino et al., 2018; Lin et al., 2025). The benefits of data repetition have also been investigated empirically in large language model pretraining (Muennighoff et al., 2023; Yan et al., 2025). In contrast, in gradient-based learning of single-index models, most existing works focus on one-pass (online) SGD (Ben Arous et al., 2021; Abbe et al., 2023; Damian et al., 2023), while analyses of full-batch GD often yield sample complexity bounds worse than those of one-pass SGD, due to the need for uniform concentration of the empirical gradient (Bietti et al., 2022; Mousavi-Hosseini et al., 2023). As previously mentioned, Dandi et al. (2024b); Lee et al. (2024); Arnaboldi et al. (2024) show that repeating each data point twice introduces a label preprocessing that lowers the information exponent and thereby improves the sample efficiency of SGD. However, the algorithms studied in these works can be viewed as online updates with a modified loss. Furthermore, in our setting of even target functions, the information exponent after arbitrary preprocessing remains at least 2 (in the language of Damian et al. (2024), the activation function we consider has *generative exponent* 2), and hence this loss-modification mechanism does not remove the $\log d$ factor in the sample complexity. Finally, Sarao Mannelli et al. (2020b) study the quadratic-link setting and show that the empirical risk landscape is benign at finite $n/d$ when the student model is *overparameterized*. While this suggests that GD on the empirical loss can succeed with linear-in-$d$ samples, it is unclear whether this advantage stems from data reuse or overparameterization; moreover, the landscape analysis does not characterize the iteration complexity of gradient-based learning.

## 2. Problem setup

**Notation.** We denote by $\mathcal{S}^{d-1}$ the unit sphere in $d$ dimensions. Given a vector $v$, $\|v\|$ denotes its $\ell_2$ norm. Given a matrix $A$, $\|A\|$ denotes its operator norm. Given a symmetric matrix $A$, $\lambda_i(A)$ denotes its $i$-th eigenvalue (in decreasing order) and $v_i(A)$ the corresponding eigenvector. We use $C, c > 0$ to denote numerical constants whose value may change from line to line.

**Weak and strong recovery.** Given $n$ samples $(x_i, y_i)_{i=1}^n$ drawn i.i.d. according to (1.1), the learning problem consists in recovering the unknown direction $\theta^\star$. Given an estimator $\hat{\theta}$ of $\theta^\star$, its performance is typically assessed either via the overlap (normalized correlation) between $\hat{\theta}$ and $\theta^\star$ or via their $\ell_2$ distance. In particular, $\hat{\theta}$ achieves weak/strong recovery if the following conditions hold:

$$\text{Weak recovery: } \lim_{n\to\infty} \frac{|\langle \hat{\theta}, \theta^\star \rangle|}{\|\hat{\theta}\| \cdot \|\theta^\star\|} = \epsilon > 0,$$

$$\text{Strong recovery: } \lim_{n\to\infty} \min_{s\in\{\pm1\}} \|\hat{\theta} - s\theta^\star\| = 0, \tag{2.1}$$

where the minimization over $s$ accounts for the sign ambiguity in the reconstruction of $\theta^\star$. Note that these performance measures can be readily related to the generalization error (Barbier et al., 2019; Maillard et al., 2020).

**Optimization algorithms.** In this paper, we study estimators $\hat{\theta}$ obtained via full-batch gradient descent methods. We will consider two choices of loss function: we start with the correlation loss $\ell(x, y; \theta) = -y\sigma(\langle x, \theta \rangle)$ in Section 3, and then move to the squared loss $\ell(x, y; \theta) = (y - \sigma(\langle x, \theta \rangle))^2$ in Section 4. Given a loss function $\ell : \mathbb{R}^d \times \mathbb{R} \times \mathbb{R}^d \to \mathbb{R}$, full-batch gradient methods compute the empirical gradient $\widehat{\mathcal{G}} : \mathbb{R}^d \to \mathbb{R}^d$ given by the following average over the data:

$$\widehat{\mathcal{G}}(\theta) := \nabla\widehat{\mathcal{L}}(\theta), \qquad \widehat{\mathcal{L}}(\theta) := \frac{1}{n}\sum_{i=1}^n \ell(x_i, y_i; \theta), \tag{2.2}$$

where the gradient is taken with respect to $\theta$ either on the sphere $\mathcal{S}^{d-1}$ (spherical gradient) or in $\mathbb{R}^d$ (Euclidean gradient). Then, gradient descent/flow is given by the following discrete/continuous iteration:

Gradient descent: $\theta_{t+1} = \theta_t - \eta\widehat{\mathcal{G}}(\theta_t),\ t \in \mathbb{N}$;

$$\text{Gradient flow: } \frac{d\theta(t)}{dt} = -\widehat{\mathcal{G}}(\theta(t)),\ t \geq 0. \tag{2.3}$$

Here, $\eta$ denotes the step size of gradient descent and gradient flow is obtained from gradient descent by taking the limit $\eta \to 0$. We remove the $\widehat{(\cdot)}$ when referring to the population counterparts of the quantities in (2.2):

$$\mathcal{G}(\theta) := \nabla\mathcal{L}(\theta), \qquad \mathcal{L}(\theta) := \mathbb{E}[\ell(x, y; \theta)], \tag{2.4}$$

where the expectation is over the input $x \sim \mathcal{N}(0, I_d)$.

We note that full-batch gradient methods reuse the same data in every iteration. In contrast, one pass SGD uses each data point only once, i.e., it corresponds to the iteration $\theta_{t+1} = \theta_t - \eta\nabla\ell(x_t, y_t; \theta_t)$, with $t$ being limited to $t \in \{1, \ldots, n\}$. For this online update, Ben Arous et al. (2021, Theorem 1.4) provides a sample complexity lower bound for the spherical dynamics on the correlation loss: when the activation function has information exponent 2 (which includes our problem setting), it is necessary to have $n \gtrsim d\log d$ samples for one-pass SGD to achieve weak recovery with any learning rate up to $\eta \lesssim 1/d$, which is the largest *stable* learning rate under the drift-plus-martingale decomposition in Ben Arous et al. (2021). In the ensuing section, we investigate whether full-batch GD can remove the $\log d$ factor in sample complexity and match the information theoretic limit in terms of dimension dependence.

## 3. Sample complexity separation in spherical gradient flow

In this section we consider the same algorithmic setup as Ben Arous et al. (2021); Damian et al. (2023) where the correlation loss is minimized on the sphere. Define the correlation loss for any $\theta \in \mathcal{S}^{d-1}$ as

$$\widehat{\mathcal{L}}(\theta) := -\frac{1}{n}\sum_{i=1}^{n} y_i\sigma(\langle x_i, \theta \rangle). \tag{3.1}$$

The Euclidean gradient can be written as

$$\nabla_{\mathbb{R}^d}\widehat{\mathcal{L}}(\theta) = -\frac{1}{n}\sum_{i=1}^{n} y_i\sigma'(\langle x_i, \theta \rangle)x_i = -A(\theta)\theta,$$

$$\text{where } A(\theta) := \frac{1}{n}\sum_{i=1}^{n} \frac{y_i\sigma'(\langle x_i, \theta \rangle)}{\langle x, \theta \rangle}x_i x_i^\top.$$

On unit sphere $\mathcal{S}^{d-1}$, the spherical gradient is the orthogonal projection of Euclidean gradient onto the tangent space at $\theta$:

$$\nabla_{\mathcal{S}^{d-1}}\widehat{\mathcal{L}}(\theta) = (I - \theta\theta^\top)\nabla_{\mathbb{R}^d}\widehat{\mathcal{L}}(\theta), \tag{3.2}$$

which gives the following gradient flow ODE,

$$\frac{d\theta(t)}{dt} = -\nabla_{\mathcal{S}^{d-1}}\widehat{\mathcal{L}}(\theta(t)) = (I - \theta(t)\theta(t)^\top)A(\theta(t))\theta(t). \tag{3.3}$$

Our goal is to characterize the necessary and sufficient sample size under which the spherical gradient flow (3.3) converges to a direction with nontrivial overlap with $\theta^\star$.

### 3.1. Quadratic activation requires $\Omega(d\log d)$ samples

We start by considering the quadratic activation $\sigma(z) = z^2$, which has been studied in Sarao Mannelli et al. (2020a); Cai et al. (2023) in the squared loss setting. We first calculate the Euclidean gradient of the correlation loss in $\theta$ as

$$\nabla_{\mathbb{R}^d}\widehat{\mathcal{L}}(\theta) = -\frac{2}{n}\sum_{i=1}^{n} y_i x_i \langle x_i, \theta \rangle = -A^\star\theta,$$

$$\text{where } A^\star := \frac{2}{n}\sum_{i=1}^{n} y_i x_i x_i^\top. \tag{3.4}$$

Hence the spherical gradient flow is implementing a power method on $A^\star$:

$$\frac{d\theta(t)}{dt} = (I - \theta(t)\theta(t)^\top)A^\star\theta(t). \tag{3.5}$$

Our first contribution is to show that, as long as the sample size $n \ll d\log d$, the performance of the estimator obtained from the spherical gradient flow in (3.5) is trivial, i.e., even weak recovery cannot be achieved. This implies that, for the quadratic activation $\sigma(z) = z^2$, full-batch spherical gradient flow with respect to the correlation loss is unable to provide a sample complexity improvement over one-pass SGD.

**Theorem 3.1.** *Consider the quadratic activation $\sigma(z) = z^2$. Let $\theta(t)$ be the solution at time $t$ of the spherical gradient flow ODE in (3.5) with initialization sampled uniformly from the sphere, i.e., $\theta(0) \sim \text{Unif}(\mathcal{S}^{d-1})$. Assume that $n = o(d\log d)$, i.e., $\lim_{n\to\infty} n/(d\log d) = 0$. Then, it holds that, almost surely,*

$$\lim_{n\to\infty}\lim_{t\to\infty}|\langle\theta(t), \theta^\star\rangle| = 0.$$

Theorem 3.1 suggests that, for the quadratic activation, minimizing the correlation loss on the sphere fails to weakly recovery $\theta^\star$ at finite $n/d$. This claim is empirically supported by Figure 1a, where we observe that as $d$ increases, achieving constant overlap requires larger $\delta = n/d$ and hence the weak recovery threshold is not finite as $d \to \infty$. Moreover, in Figure 1c we quantitatively verify the predicted $n/d \simeq \log d$ threshold.

*Proof sketch.* The ODE in (3.5) is a power iteration on the matrix $A^\star$. Thus, as the top eigenvalue of $A^\star$ can be shown to be simple with probability 1, $\theta(t)$ for large $t$ approaches

the principal eigenvector $v_1(A^\star)$. The spectrum of random matrices of the form in (3.4), as well as the overlap between $v_1(A^\star)$ and $\theta^\star$, has been computed precisely in (Lu & Li, 2020; Mondelli & Montanari, 2018) when *(i)* the activation $\sigma$ is bounded, and *(ii)* $n, d \to \infty$ with their ratio $n/d$ held fixed. Remarkably, following a similar strategy, we are able to show that $v_1(A^\star)$ is asymptotically uncorrelated with $\theta^\star$ when *(i)* the activation $\sigma(z) = z^2$ is unbounded, and *(ii)* $n, d \to \infty$ with $n = o(d \log d)$. We write $\sum_{i=1}^{n} y_i x_i x_i^\top$ into the block form $\begin{pmatrix} a & q^\top \\ q & P \end{pmatrix}$. Then, $|\langle v_1(A^\star), \theta^\star \rangle|$ can be expressed as $L'(\mu^*)/(L'(\mu^*) + 1/(\mu^*)^2)$, where $L(\mu)$ is the largest eigenvalue of $P + \mu q q^\top$, $L'(\mu)$ is its derivative (more precisely, (A.6) gives an upper/lower bound in terms of the right/left derivative, in case the derivative does not exist), and $\mu^*$ solves the fixed point equation $\mu^* = (L(\mu^*) - a)^{-1}$. Now, the largest eigenvalue of $P$ (and therefore of the rank-1 update $P + \mu q q^\top$) is at least $d \log n$, which implies $1/\mu^* = d \log n (1 + o(1))$. Further, $L'(\cdot)$ is upper bounded by a constant times $nd$. Plugging these bounds into the above expression for $|\langle v_1(A^\star), \theta^\star \rangle|$ gives the desired claim. The full proof is deferred to Appendix A. $\qquad\square$

### 3.2. $\Theta(d)$ samples suffice for truncated activation

As mentioned in the proof sketch of Theorem 3.1, for the quadratic activation, the spherical gradient flow in (3.5) is equivalent to a power iteration on the matrix $A^\star$, and its poor performance is due to the fact that the quadratic activation is unbounded. In fact, for bounded $\sigma$, existing results[1] give that $|\langle v_1(A^\star), \theta^\star \rangle|$ is bounded away from 0 when $n/d$ is sufficiently large. Thus, it is natural to study the performance of the spherical gradient flow in (3.5) with a *truncated* activation $\sigma$. For technical reasons (namely, to guarantee well-posedness and adequate regularity of the solution to the gradient flow), we opt for a smooth truncation as defined below. Let us fix constants $M, c > 0$, and introduce a $C^\infty$ cutoff function $\varphi : \mathbb{R} \to [0, 1]$ such that $\varphi(u) = 1$ for $|u| \leq M$, $\varphi(u) = 0$ for $|u| \geq 2M$ and $-c/M \leq \varphi'(u) \leq 0$ for $u \in \mathbb{R}$. This function can be constructed e.g. by *(i)* considering the standard bump function $\gamma(u) = e^{-1/u}$ for $u > 0$ and $\gamma(u) = 0$ for $u \leq 0$, *(ii)* defining $S(u) := \frac{\gamma(u)}{\gamma(u) + \gamma(1-u)}$, and *(iii)* setting $\varphi(u) = 1 - S\left(\frac{|u| - M}{M}\right)$. Next, we define

$$\sigma(z) := \int_0^{z^2} \varphi(u) du. \qquad (3.6)$$

This construction can be interpreted as a smooth truncation of the quadratic activation, since $\sigma(z) = z^2$ for $|z| \leq \sqrt{M}$, $\sigma(z) = h(z)$ for $|z| \in [\sqrt{M}, \sqrt{2M}]$ and $\sigma(z) = \bar{M}$ for

$|z| \geq \sqrt{2M}$, where $\bar{M} := \int_0^{2M} \varphi(u) du$ denotes the maximum value attained by $\sigma$, and $h$ is a smooth interpolation between the quadratic and constant part of $\sigma$. Then, $\sigma'(z) = 2z\, \varphi(z^2)$ and the Euclidean gradient of the correlation loss $\widehat{\mathcal{L}}(\theta)$ in (3.1) is given by

$$\nabla_{\mathbb{R}^d} \widehat{\mathcal{L}}(\theta) = -\frac{2}{n} \sum_{i=1}^{n} y_i x_i \langle x_i, \theta \rangle \varphi(\langle x_i, \theta \rangle^2) = -A(\theta)\theta,$$

where $A(\theta) := \frac{2}{n} \sum_{i=1}^{n} y_i x_i x_i^\top \varphi(\langle x_i, \theta \rangle^2)$.

$$(3.7)$$

Thus, the ODE defining the spherical gradient flow is

$$\frac{d\theta(t)}{dt} = (I - \theta(t)\theta(t)^\top) A(\theta(t))\theta(t). \qquad (3.8)$$

In contrast with the ODE in (3.5) which corresponds to a power iteration over the *fixed* matrix $A^\star$, the ODE in (3.8) corresponds to a power iteration over the *time-varying* matrix $A(\theta(t))$.

Our second contribution is to show that, when $n \gtrsim d$, the estimator obtained from the spherical gradient flow in (3.8) has overlap with $\theta^\star$ that approaches 1 as $M$ and the ratio $n/d$ grow and, therefore, it achieves weak recovery at finite $n/d$. As an immediate consequence, this result shows that full-batch GD is indeed able to achieve linear-in-$d$ sample complexity in exactly the same setting in which one-pass SGD requires $\gtrsim d \log d$ samples (recall that the truncated activation also has information exponent 2, so the lower bound for online SGD in Ben Arous et al. (2021) still applies).

**Theorem 3.2.** *Consider the smooth truncated activation $\sigma$ defined in (3.6), where $M$ is a large enough constant (independent of $n$, $d$). Let $\theta(t)$ be the solution at time $t$ of the spherical gradient flow ODE in (3.8), with initialization sampled uniformly from the sphere, i.e., $\theta(0) \sim \text{Unif}(\mathcal{S}^{d-1})$. Let $d$ be large enough and $n \geq CM^4 d$. Then, it holds that, with probability at least $1 - \exp(-cd^{1/5})$,*

$$\lim_{t \to \infty} |\langle \theta(t), \theta^\star \rangle| \geq 1 - C(e^{-M/2} + (d/n)^{1/5}), \quad (3.9)$$

*where $C, c > 0$ are constants (independent of $n, d, M$).*

We remark that the requirement $n \geq CM^4 d$ can be relaxed to $n \geq CM^{2.01} d$ using a more refined spectral analysis of the matrix $A^\star$, see Appendix C for details.

The result of Theorem 3.2 is illustrated in Figure 1b. We observe that, by simply truncating the quadratic link function, the learning curves at different $d$ mostly collapse (except at small values of $\delta = n/d$ which is likely due to non-asymptotic fluctuations), suggesting that the weak recovery threshold is constant: $\delta = \Theta(1)$. This contrasts the un-truncated setting (Figure 1a) where the required $\delta$ clearly increases with $d$.

---

[1] This can be seen from the sharp characterization of Lu & Li (2020); Mondelli & Montanari (2018) via random matrix theory or the concentration results in Chen & Candès (2017), and we will use similar ideas in our analysis (see Propositions B.1 and C.1).

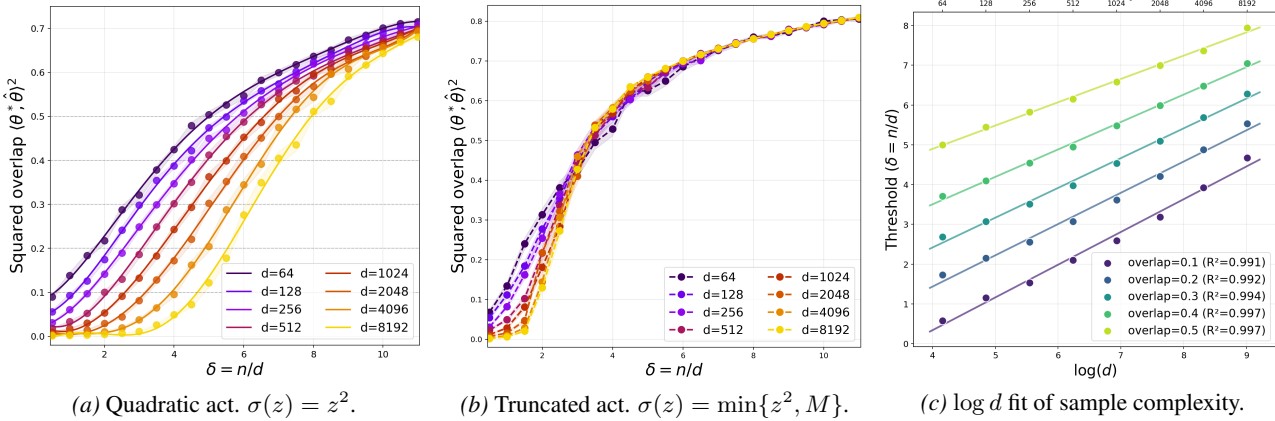

*(a)* Quadratic act. $\sigma(z) = z^2$.     *(b)* Truncated act. $\sigma(z) = \min\{z^2, M\}$.     *(c)* $\log d$ fit of sample complexity.

*Figure 1.* Overlap achieved by minimizing the empirical correlation loss on the sphere as a function of $\delta = n/d$. We run spherical GD with learning rate $\eta = 0.1$ for $T = 1000 \log^2 d$ steps. **Left:** for the unbounded quadratic activation, increasing $d$ yields a larger threshold $\delta$ for weak recovery; we include a spline fit (solid lines) to smooth out the fluctuations. **Middle:** for the truncated activation ($M = 8$) the overlap at fixed $\delta$ is almost $d$-independent. **Right:** threshold $\delta = n/d$ required to achieve target squared overlap values $\{0.1, 0.2, 0.3, 0.4, 0.5\}$ for $\sigma(z) = z^2$ (extracted from Figure 1a), where we observe a clear $\delta \simeq \log d$ fit.

*Proof sketch.* The proof has two main parts: *(i)* uniform spectral control of $A(\theta)$, and *(ii)* convergence of the gradient flow $\theta(t)$ to the principal eigenvector of $A(\theta)$.

As for part *(i)*, the key difficulty is that $A(\theta)$ changes with $\theta$. To remove this dependence, we express $A(\theta) = A^\star - B(\theta)$, where $A^\star$ is defined in (3.4) and $B(\theta) \succeq 0$ captures the truncation error. Note that the measurements $y_i$ are now bounded, so we can use the concentration of the empirical covariance matrix of sub-gaussian vectors to establish the spectral behavior of $A^\star$. More precisely, Proposition B.1 gives that, with probability $1 - e^{-d}$, the top eigenvector of $A^\star$ is almost aligned with the signal $\theta^\star$, the top-two eigenvalues of $A^\star$ are of constant order and there is a constant-order gap between them, i.e.,

$$|\langle v_1(A^\star), \theta^\star \rangle| \leq CM \sqrt{\frac{d}{n}},$$

$$|\lambda_1(A^\star) - 6| + |\lambda_2(A^\star) - 2| \leq C \left( e^{-M/3} + M \sqrt{\frac{d}{n}} \right).$$
$$(3.10)$$

We then show that $B(\theta)$ is small uniformly, since it is supported on the rare events that $\left\{ \langle x_i, \theta \rangle^2 > M \right\}$. More precisely, a VC dimension argument in Lemma B.2 gives uniform control of the empirical indicator mass, i.e.,

$$\frac{1}{n} \sum_{i=1}^{n} \mathbf{1} \left[ \langle x_i, \theta \rangle^2 > M \right] \leq C \left( e^{-M/2} + \sqrt{\frac{d}{n}} \log \left( \frac{n}{d} \right) \right)$$

holds with probability at least $1 - e^{-d}$ uniformly over $\theta \in \mathcal{S}^{d-1}$. This implies that the perturbation caused by $B(\theta)$ does not change much the spectral properties of $A(\theta)$, which

are therefore similar to those of $A^\star$ in (3.10) for any $\theta$ (see Theorem B.3).

As for part *(ii)*, Proposition B.4 shows that the empirical loss $\hat{\mathcal{L}}$ is a Lyapunov function along the flow, i.e., $\frac{d}{dt} \hat{\mathcal{L}}(\theta(t)) = -\|\nabla_{\mathcal{S}^{d-1}} \hat{\mathcal{L}}(\theta(t))\| \leq 0$. Thus, as $\hat{\mathcal{L}}$ is bounded from below, $\theta(t)$ converges to some stationary point $\theta_\infty$, which from (3.8) needs to be an eigenvector of $A(\theta)$. It remains to exclude convergence to the non-principal eigenvectors of $A(\theta)$. To do so, Proposition B.5 follows the approach developed in (Panageas & Piliouras, 2017) for standard gradient descent in the plane, with suitable adjustments stemming from working with gradient flow on the sphere. Denoting by $S$ the set of stationary points which are not principal eigenvectors, the center-stable manifold theorem implies that the basin of attraction of any $\theta_s \in S$ lies in a lower dimensional center-stable manifold, provided the linearization of the flow has an unstable direction. The linearization is governed by the spherical Hessian, expressed via $A(\theta) - \lambda_{\theta_s} I$ plus a reminder term due to the smoothening of the truncation. We use the uniform spectral gap and alignment of $A(\theta)$ from part *(i)* to get that unstable directions always exist at any non-principal stationary point, with the remainder terms being negligible for large enough $M$. Hence, the union of these basins has measure 0, and with random initialization the flow almost surely converges to the principal eigenvector of $A(\theta)$ (for some $\theta$). The latter, regardless of the value of $\theta$, is almost aligned with $\theta^\star$ due to the analysis of part *(i)*, which concludes the argument. The complete proof is contained in Appendix B.   □

We highlight that the argument of Theorem 3.2 crucially relies on showing a uniform-in-$\theta$ BBP transition for the matrix $A(\theta)$: with high probability, for all $\theta$ there is a spectral gap, in the sense that the top two eigenvalues of $A(\theta)$ are

separated by a constant (the largest is close to 6 and the second largest to 2, as in (3.10)), and the top eigenvector has an overlap with $\theta^\star$ which approaches 1 as the ratio $n/d$ grows. However, this type of landscape argument alone is unable to *(i)* bound the time it takes for the gradient flow to achieve weak recovery, since the matrix $A(\theta(t))$ is time-varying and the error in controlling the spectrum of $A(\theta(t))$ is an order $\sqrt{d}$ larger than the initial overlap $|\langle \theta(0), \theta^\star \rangle|$, and *(ii)* give strong recovery guarantees (see (2.1)), namely achieve arbitrarily small loss for a fixed $n/d$.

These two desiderata motivate a more refined understanding of the gradient descent dynamics, which we develop in Section 4 in the setting of minimizing the squared loss from small initialization.

# 4. Strong recovery and iteration complexity

We have previously shown that for a truncated quadratic activation, spherical gradient flow achieves weak recovery in the proportional $n, d$ regime. We now study the time complexity of full-batch GD and establish strong recovery guarantees for a truncated activation. To go beyond the landscape analysis in Section 3.2, we leverage the observation of Stöger & Soltanolkotabi (2021) that small initialization simplifies the analysis of the training dynamics: roughly speaking, with sufficiently small initialization, the "early phase" of GD is approximated by a power iteration on the matrix (3.4), which — thanks to the truncation — exhibits an informative BBP transition at finite $n/d$. On the other hand, because the parameters are initialized small (whereas $\|\theta^\star\| = 1$), the algorithm must also learn the correct norm. This is something that the correlation loss cannot capture, as the whole dynamics occurs on a sphere where the norm of the estimator does not change. Hence, we move to Euclidean gradient descent for minimizing the squared loss:

$$\widehat{\mathcal{L}}(\theta) = \frac{1}{2n} \sum_{i=1}^{n} \left( \sigma(\langle x_i, \theta \rangle) - \sigma(\langle x_i, \theta^\star \rangle) \right)^2. \qquad (4.1)$$

The gradient descent iteration is given by

$$\theta_{t+1} = \theta_t - \eta \widehat{\mathcal{G}}(\theta_t), \qquad (4.2)$$

where $\eta$ is the constant learning rate and $\widehat{\mathcal{G}}(\cdot) := \nabla \widehat{\mathcal{L}}(\cdot)$ denotes the empirical gradient given by

$$\widehat{\mathcal{G}}(\theta) = \frac{1}{n} \sum_{i=1}^{n} \left( \sigma(\langle x_i, \theta \rangle) - \sigma(\langle x_i, \theta^\star \rangle) \right) \sigma'(\langle x_i, \theta \rangle) x_i. \qquad (4.3)$$

We focus on the truncated quadratic activation $\sigma$ given by

$$\sigma(z) := \begin{cases} z^2, & \text{if } z^2 \le M, \\ M, & \text{if } z^2 > M. \end{cases} \qquad (4.4)$$

We note that choosing this truncation, as opposed to the smooth one in (3.6), is only for technical convenience, and we expect our results to hold more broadly.

Our third contribution is to show that, when $n \gtrsim d$, the estimator obtained from the gradient descent iteration in (4.2) achieves strong recovery in $\gtrsim \log d$ steps. More precisely, we prove that, after a "search phase" lasting at most an order of $\log d / \eta$ steps, the error $\|\theta_t - \theta^\star\|^2$ converges geometrically to 0 with $t$. This is the case for any fixed $n/d$ larger than a threshold depending only on $M$. We contrast the behavior proved in Theorem 4.1 below with the earlier result of Theorem 3.2 which *(i)* does not provide a time-complexity guarantee, and where *(ii)* the residual error gets arbitrarily small only as $n/d$ grows.

**Theorem 4.1.** *Consider the truncated quadratic activation in (4.4), where $M$ is a large enough constant (independent of $n$, $d$). Let $\theta_t$ be obtained after $t$ steps of the gradient descent iteration in (4.2), with learning rate $\eta \le c/M^2$. Assume that the initialization is sampled uniformly from the sphere of radius $r_0 = d^{-15}$, that is, $\theta_0 \sim \mathrm{Unif}(r_0 \mathcal{S}^{d-1})$. Let $d$ be large enough and $n \ge CM^4 d$. Then, for all $t \ge \bar{t}$, it holds that, with probability at least $1 - 2/d^2$,*

$$\|\theta_t - \theta^\star\|^2 \le C(1 - \eta\alpha)^{t-\bar{t}}, \qquad (4.5)$$

*where $\alpha, C$ are numerical constants (independent of $n, d, M$) and $\bar{t} \le C \log d / \eta$.*

To our knowledge, this is the first convergence rate and strong recovery guarantee for learning single-index models with information exponent 2 using standard full-batch GD in the proportional regime (i.e., empirical loss with finite $n/d$). We remark that the requirement $r_0 = d^{-15}$ can be relaxed to a lower power of $1/d$ at the cost of a more involved argument.

In Figures 2a and 2b, we track the growth of overlap and norm as a function of the number of GD steps (note that to guarantee small squared loss, we need to check the convergence of both quantities). We observe that, as the dimension gets larger (while keeping $n/d$ constant), the training dynamics takes longer to converge. In Figure 2c we quantitatively verify the required $T \simeq \log d$ runtime predicted by Theorem 4.1.

*Proof sketch.* The argument distinguishes two main phases in the gradient dynamics: *(i)* a *first search* phase, where the angle $\angle(\theta_t, \theta^\star)$ between $\theta_t$ and $\theta^\star$ shrinks to a small constant (independent of $n, d$) and the norm of $\theta_t$ grows up to a constant fraction (also independent of $n, d$) of the target norm $\|\theta^\star\|$, and *(ii)* a *second refinement* phase, which starts in the vicinity of the optimum and exhibits the *geometric convergence* of the distance to the optimum in (4.5).

We start by discussing the analysis of the first phase. Here,

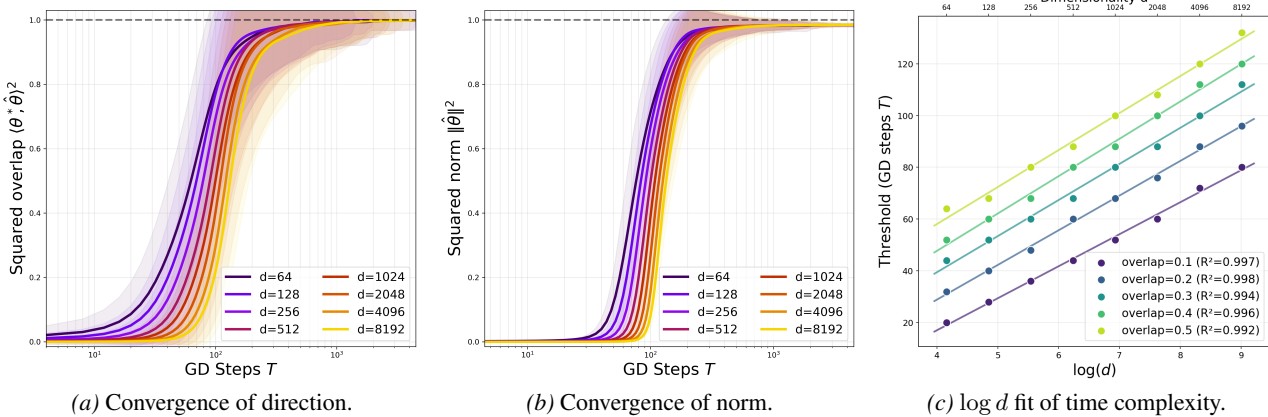

*(a)* Convergence of direction.      *(b)* Convergence of norm.      *(c)* $\log d$ fit of time complexity.

*Figure 2.* Overlap and parameter norm vs. number of GD steps. We use the truncated quadratic activation (4.4) with $M = 8$, run Euclidean gradient descent with learning rate $\eta = 0.1/M^2$ and initialization scale $1/d^2$, and fix $\delta = n/d = 10$. **Left, Middle:** Observe that the time required for non-trivial overlap and norm growth increases with $d$. **Right:** number of GD steps required to achieve target squared overlap values $\{0.1, 0.2, 0.3, 0.4, 0.5\}$ (extracted from Figure 2a), where we observe a clear $T \simeq \log d$ fit.

Proposition D.1 (see Appendix D.1) gives that

$$\angle(\theta_{t^*_{\measuredangle}}, \theta^\star) \leq C(e^{-M/2} + M\delta^{-1/2}), \qquad (4.6)$$

where $t^*_{\measuredangle} = \frac{3\log d}{\log(1+1.99\eta)}$. The idea is that, for $t \leq t^*_{\measuredangle}$, $\|\theta_t\|$ remains small and, therefore, $\langle x_i, \theta_t \rangle^2$ always stays below the threshold $M$. Thus, the dynamics is effectively a power iteration for the matrix $A^\star$, which exhibits the benign properties in (3.10). In particular, the choice of $t^*_{\measuredangle}$ is such that, at $t = t^*_{\measuredangle}$, *(i)* the norm $\|\theta_t\|$ has not grown enough for the truncation to have an effect, and *(ii)* the angle $\angle(\theta_t, \theta^\star)$ has become an arbitrarily small constant. Once the angle has reached a certain value, it can never exceed it again in the following iterations, see Proposition D.15 in Appendix D.4. At the same time, during this first phase, the norm $\|\theta_t\|$ grows geometrically in $t$, reaching the value of $1/4$ after an order of $\log d/\eta$ steps, see Proposition D.14 in Appendix D.3. These last two results (improvement in the angle and growth of the norm) rely on the uniform concentration of the empirical gradient $\widehat{\mathcal{G}}(\cdot)$ to its population counterpart $\mathcal{G}(\cdot) := \nabla\mathcal{L}(\cdot)$ given by

$$\mathcal{G}(\theta) = \mathbb{E}\Big[(\sigma(\langle x, \theta \rangle) - \sigma(\langle x, \theta^\star \rangle))\,\sigma'(\langle x, \theta \rangle)\,x\Big], \quad (4.7)$$

where the expectation is over $x \sim \mathcal{N}(0, I_d)$. The concentration is proved in Proposition D.8 (see Appendix D.2).

During the second phase, the iterate $\theta_t$ is already in a neighborhood of a global optimum: its norm is a constant factor away from $\|\theta^\star\| = 1$, and the angle $\angle(\theta_t, \theta^\star)$ is small (see (4.6)). Then, Proposition D.26 (see Appendix D.6) shows that the loss converges geometrically by establishing the one-point strong convexity

$$\Big\langle \widehat{\mathcal{G}}(\theta), \theta - \theta^\star \Big\rangle \geq \alpha \|\theta - \theta^\star\|^2, \qquad (4.8)$$

for a numerical constant $\alpha > 0$ (independent of $n, d$). To do so, we relate $\widehat{\mathcal{G}}(\theta)$ to the Gram matrix of the Jacobian

$$\widehat{H}(\theta, \widetilde{\theta}) := \frac{1}{n}\sum_{i=1}^{n} \sigma'(\langle x_i, \theta \rangle)\,\sigma'(\langle x_i, \widetilde{\theta} \rangle)\,x_i x_i^\top, \quad (4.9)$$

for a suitable choice of $\widetilde{\theta}$. Then, the key technical step consists in showing the uniform (in $\theta, \widetilde{\theta}$) concentration of $\widehat{H}(\theta, \widetilde{\theta})$ to the corresponding population quantity $H(\theta, \widetilde{\theta}) := \mathbb{E}[\widehat{H}(\theta, \widetilde{\theta})]$, see Proposition D.20 in Appendix D.5. By combining this concentration result with a lower bound on the population $H(\theta, \widetilde{\theta})$, we obtain (4.8), which leads to the desired geometric convergence of the error. The complete proof is contained in Appendix D. □

Our two-phase analysis is loosely inspired by the low-rank recovery literature (Stöger & Soltanolkotabi, 2021; Soltanolkotabi et al., 2023), which often proceeds through *(i)* subspace alignment, *(ii)* escape from saddle regions, and *(iii)* local refinement. This parallels our initial angle reduction and norm growth, followed by a local refinement phase. Beyond this high-level resemblance (and our use of a power-method–type initialization analysis, following that line of work), the approaches are fundamentally different. First, our non-linear forward model induces a different population loss and requires a specialized analysis. Second, and more importantly, each observation contains substantially less randomness (on the order of $d$ here, versus $d^2$ in low-rank recovery). Therefore, in contrast with this literature that utilizes well-established restricted isometry properties, we need to develop rather intricate and ad-hoc uniform concentration arguments. Finally, Chen et al. (2019) establish related convergence guarantees for quadratic activations, but still require $n \gtrsim d\,\mathrm{polylog}(d)$. Furthermore, their analysis

relies on very different techniques based on leave-one-out arguments, which require the additional logaritmic factors in the sample complexity.

## 5. Concluding remarks

In this work, we characterize the sample and iteration complexity of full-batch gradient descent in a canonical nonlinear single-index problem with information exponent 2, where one-pass SGD requires $n \gtrsim d \log d$ samples. In the spherical-gradient/correlation-loss setup, we prove a negative result for the unbounded quadratic activation: when $n \ll d \log d$, full-batch spherical gradient flow converges to an uninformative direction, implying that full-batch updates do not outperform one-pass SGD in sample complexity. At the same time, we establish a positive result via a minimal modification: with a truncated quadratic activation, full-batch spherical gradient flow achieves weak recovery with linear samples $n \gtrsim d$, matching the information-theoretically optimal dimension dependence (without the $\log d$ factor). Finally, moving beyond a landscape analysis, we study full-batch GD on the squared loss from small initialization, and we establish strong recovery together with a convergence rate guarantee: with $n \gtrsim d$ samples, GD reaches exact recovery after a logarithmic number of steps $T \gtrsim \log d$, yielding the first convergence rate results for plain full-batch GD in the proportional regime.

Several directions emerge naturally. First, our refined trajectory analysis is currently limited to squared loss dynamics from small initialization; an important next step is to characterize the runtime complexity of the spherical dynamics in Section 3. Second, it would be interesting to compute sharp constants for the phase transition in the sample complexity of full-batch GD (e.g., via dynamical mean-field theory (Celentano et al., 2021; Gerbelot et al., 2024; Han, 2025; Chen & Shen, 2025)). Finally, our current setting is restricted to single-index models with information exponent 2, where one-pass SGD and the information-theoretic limit exhibit a $\log d$ gap; a natural direction is to consider link functions with higher information exponent, or more generally multi-index models, and examine whether full-batch updates yield a similar (or potentially larger) statistical gain.

## Impact Statement

This paper presents work whose goal is to advance the field of Machine Learning. There are many potential societal consequences of our work, none which we feel must be specifically highlighted here.

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

## A. Proof of Theorem 3.1

We start by showing a crucial intermediate result: the principal eigenvector of $A^\star$ is asymptotically uncorrelated with $\theta^\star$.

**Theorem A.1.** *Assume $n = o(d \log d)$ and let $A^\star$ be defined in (3.4). Then, it holds that, almost surely,*

$$\lim_{n \to \infty} \langle \theta^\star, v_1(A^\star) \rangle^2 = 0, \tag{A.1}$$

*where $v_1(A^\star)$ is the principal eigenvector of $A^\star$.*

*Proof.* Since the claim is about the top eigenvector of $A^\star$, we can rescale $A^\star$ by a factor $n/2$. Furthermore, as $x_i \sim \mathcal{N}(0, I_d)$ is rotationally invariant, we can take $\theta^\star = e_1$ also without loss of generality, $e_1$ being the first element of the canonical basis in $\mathbb{R}^d$. Next, we write $X = (x_1, \ldots, x_n) \in \mathbb{R}^{d \times n}$ as

$$X = \begin{pmatrix} v^\top \\ U \end{pmatrix}, \qquad v \in \mathbb{R}^n,\ U \in \mathbb{R}^{(d-1) \times n}. \tag{A.2}$$

Let $Z = \operatorname{diag}(|\langle x_i, e_1 \rangle|^2)_{1 \le i \le n} \in \mathbb{R}^{n \times n}$ and use (A.2) to write

$$A^\star = \begin{pmatrix} v^\top \\ U \end{pmatrix} Z \begin{pmatrix} v & U^\top \end{pmatrix} = \begin{pmatrix} v^\top Z v & v^\top Z U^\top \\ U Z v & U Z U^\top \end{pmatrix} =: \begin{pmatrix} a & q^\top \\ q & P \end{pmatrix}. \tag{A.3}$$

Define the map

$$[0, \infty) \ni \mu \mapsto L(\mu) = \lambda_1(P + \mu q q^\top) > 0, \tag{A.4}$$

and let $\mu^* > 0$ solve

$$\mu = \frac{1}{L(\mu) - a}. \tag{A.5}$$

Then, by Proposition 2 of (Lu & Li, 2020), such $\mu^*$ is unique, $\lambda_1(A^\star) = L(\mu^*)$ and

$$\frac{\partial_- L(\mu^*)}{\partial_- L(\mu^*) + (1/\mu^*)^2} \le |\langle e_1, v_1(A^\star) \rangle|^2 \le \frac{\partial_+ L(\mu^*)}{\partial_+ L(\mu^*) + (1/\mu^*)^2}, \tag{A.6}$$

where $\partial_-$ and $\partial_+$ stand for the left- and right-derivatives.

Next, we claim that there exists a constant $c > 0$ such that for all small enough $\epsilon > 0$

$$\mathbb{P}[\lambda_1(P) < (2 - \epsilon) d \log n] \le e^{-cd\epsilon^2} + e^{-n^{\epsilon/3}/2}. \tag{A.7}$$

**Proof of the claim in (A.7).** Let $\hat{\imath} = \arg\max_i |\langle x_i, e_1 \rangle|^2$. Then,

$$\lambda_1(P) \ge \frac{e_{\hat{\imath}}^\top U^\top P U e_{\hat{\imath}}}{\|U e_{\hat{\imath}}\|^2} = \sum_{i=1}^n |\langle x_i, e_1 \rangle|^2 \frac{|\langle U e_i, U e_{\hat{\imath}} \rangle|^2}{\|U e_{\hat{\imath}}\|^2} \ge |\langle x_{\hat{\imath}}, e_1 \rangle|^2 \|U e_{\hat{\imath}}\|^2. \tag{A.8}$$

Since $U$ is independent of the first row of $X$, hence of $\hat{\imath}$ and $\langle x_{\hat{\imath}}, e_1 \rangle$, we have the distributional identity

$$|\langle x_{\hat{\imath}}, e_1 \rangle|^2 \|U e_{\hat{\imath}}\|^2 \overset{d}{=} \left( \max_{1 \le i \le n} T_i^2 \right) \left( \sum_{j=1}^{d-1} Y_j^2 \right) =: TY, \tag{A.9}$$

where $T_1, \ldots, T_n, Y_1, \ldots, Y_{n-1}$ are i.i.d. standard real Gaussian random variables. By a standard concentration estimate, e.g. Chernoff inequality, we have

$$\mathbb{P}[Y \le d(1 - \epsilon/3)] \le e^{-cd\epsilon^2}, \tag{A.10}$$

for some constant $c > 0$. We thus have

$$\mathbb{P}[\lambda_1(P) < (2 - \epsilon) d \log n] \le \mathbb{P}[TY \le (2 - \epsilon) d \log n] \le e^{-cd\epsilon^2} + \mathbb{P}\left[ T \le \frac{2 - \epsilon}{1 - \epsilon/3} \log n \right]$$

$$\le e^{-cd\epsilon^2} + \mathbb{P}[T \le 2(1 - \epsilon/3) \log n]. \tag{A.11}$$

We then conclude using that, for another constant $c > 0$,

$$\mathbb{P}[T \leq 2(1 - \epsilon/3) \log n] = \mathbb{P}[T_1^2 \leq 2(1 - \epsilon/3) \log n]^n \leq \left(1 - e^{-(1-\epsilon/3) \log n}\right)^{n/2}$$

$$= \left(1 - n^{-(1-\epsilon/3)}\right)^{n/2} \leq e^{-n^{\epsilon/3}/2}. \tag{A.12}$$

$\square$

The claim in (A.7) implies that $L(\mu) \geq L(0) = \lambda_1(P) \geq d \log n$ with high probability. Furthermore, by Chebyshev's inequality, with probability $1 - O(1/n)$, we have that $a \leq 4n$. We thus have that, with high probability,

$$\frac{1}{\mu^*} = L(\mu^*) - a \geq d \log n - 4n = d \log n(1 + o(1)), \tag{A.13}$$

by the assumption that $n = o(d \log d)$. Finally, notice that the map $L(\mu)$ is Lipschitz with the Lipschitz constant given by

$$\|qq^\top\| = v^\top Z U^\top U Z v \leq \|U\|^2 \|Zv\|^2. \tag{A.14}$$

By Theorem 4.4.5 of (Vershynin, 2018), we have that

$$\mathbb{P}[\|U\|^2 \geq K(n+d)(1+\epsilon)] \leq e^{-(n+d)\epsilon^2}, \tag{A.15}$$

for some constant $K > 0$. Furthermore, by Chebyshev's inequality, with probability $1 - O(1/n)$, we have that $\|Zv\|^2 \leq 16n$. Hence,

$$\partial_{\pm} L(\mu^*) \leq \|qq^\top\| \leq 17Knd \tag{A.16}$$

with high probability. Therefore by (A.6) we have that, with high probability,

$$|\langle e_1, v_1(A^\star) \rangle|^2 \leq \frac{17Knd}{(d \log n)^2(1 + o(1))} = o(1), \tag{A.17}$$

which concludes the argument. $\square$

We are now ready to prove Theorem 3.1.

*Proof of Theorem 3.1.* We first show that the top eigenvalue of $A^\star$ is simple with probability 1. If $n = 1$, $A^\star = 2y_1 x_1 x_1^\top$. This matrix has rank 1 whenever the vector $\sqrt{y_1} x_1$ is not 0, which happens with probability 1. Thus, we can assume $n \geq 2$. Let

$$A_{n-1} := \frac{2}{n} \sum_{i=1}^{n-1} y_i x_i x_i^\top, \qquad \text{so that} \qquad A^\star = A_{n-1} + \frac{2}{n} y_n x_n x_n^\top.$$

We write $u_n := x_n \sqrt{2y_n/n}$, so $A^\star = A_{n-1} + u_n u_n^\top$.

Let $A \in \mathbb{R}^{d \times d}$ be symmetric and let $E := \ker(A - \lambda_1(A)I)$ be its top eigenspace. If $u \not\perp E$, we claim that the largest eigenvalue of $B := A + uu^\top$ is simple.

**Proof of the claim.** Choose a unit vector $v \in E$ with $u^\top v \neq 0$. Then

$$v^\top B v = v^\top A v + (u^\top v)^2 = \lambda_1(A) + (u^\top v)^2 > \lambda_1(A),$$

hence $\lambda_1(B) \geq v^\top B v > \lambda_1(A)$. Next, by the Courant–Fischer characterization,

$$\lambda_2(B) = \min_{\substack{T \subset \mathbb{R}^d \\ \dim T = d-1}} \max_{\substack{w \in T \\ \|w\|=1}} w^\top B w.$$

Taking $T$ to be the orthogonal subspace to $u$, for any unit $w \in T$ we have $w^\top B w = w^\top A w + (u^\top w)^2 = w^\top A w \leq \lambda_1(A)$, so

$$\lambda_2(B) \leq \max_{\substack{w \in u^\perp \\ \|w\|=1}} w^\top B w \leq \lambda_1(A).$$

Therefore $\lambda_1(B) > \lambda_1(A) \geq \lambda_2(B)$, which implies $\lambda_1(B)$ has multiplicity 1. $\qquad\square$

Now, let $E_{n-1}$ be the top eigenspace of $A_{n-1}$. By the claim,

$$\mathbb{P}(\lambda_1(A^\star) \text{ is not simple}) \leq \mathbb{P}(u_n \perp E_{n-1}) \leq \mathbb{P}(y_n = 0) + \mathbb{P}(x_n \perp E_{n-1}).$$

On the other hand, notice that $y_n = |\langle x_n, \theta^* \rangle|^2$ is $\chi^2$-distributed and that $x_n \in \mathbb{R}^d$ is rotationally invariant and independent of $x_1, \ldots, x_{n-1}$ hence of the non-empty subspace $E_{n-1}$ of $\mathbb{R}^d$. Thus the last quantity is zero, proving that $\lambda_1(A^*)$ is simple with probability 1. As an immediate consequence, with probability 1, as $t \to \infty$, $\theta(t)$ converges to the principal eigenvector of $A^\star$. Hence, the desired result follows from Theorem A.1 and an application of the Borel-Cantelli lemma. $\qquad\square$

# B. Proof of Theorem 3.2

We start by showing spectral properties of the matrix $A(\theta)$ defined in (3.7). Next, in Appendix B.2 we prove general convergence results for the spherical gradient flow, relying on parts of the analysis from Appendix B.1. Lastly, by combining these results, the desired convergence of gradient flow with random initialization follows and the argument is presented in Appendix B.3.

## B.1. Spectral properties of $A(\theta)$

**Proposition B.1.** *Let $\{x_i\}_{i=1}^n \overset{\text{i.i.d.}}{\sim} \mathcal{N}(0, I_d)$ and $y_i = \sigma(\langle x_i, \theta^\star \rangle)$, where $\sigma$ is a truncation of the quadratic function, given by either (3.6) or (4.4), with $M$ large enough constant (independent of $n$, $d$). Let $\delta = n/d$ and assume that $\delta \geq CM^4$. Let $D_n \in \mathbb{R}^{d \times d}$ and $E_d \in \mathbb{R}^{d \times d}$ be defined as*

$$D_n := \frac{1}{n} \sum_{i=1}^n y_i x_i x_i^\top, \qquad E_d := \begin{pmatrix} c_\sigma^1 & 0 & 0 & \ldots & 0 \\ 0 & c_\sigma^2 & 0 & \ldots & 0 \\ 0 & 0 & c_\sigma^2 & \ldots & 0 \\ \vdots & \vdots & \vdots & \ddots & \vdots \\ 0 & 0 & 0 & 0 & c_\sigma^2 \end{pmatrix} = \begin{pmatrix} c_\sigma^1 & 0 \\ 0 & c_\sigma^2 I_{d-1} \end{pmatrix},$$

*where $c_\sigma^1, c_\sigma^2 \in \mathbb{R}$ are constants dependent on $\sigma$, such that $|c_\sigma^1 - 3| \leq Ce^{-M/3}$ and $|c_\sigma^2 - 1| \leq Ce^{-M/3}$. Moreover, let $U$ be an orthogonal matrix such that $U^\top \theta^\star = e_1$, where $e_1$ is the first element of the canonical basis of $\mathbb{R}^d$. Then it holds that*

$$\|D_n - U E_d U^\top\| \leq C_1 M \delta^{-1/2} \leq C_2 \delta^{-1/4}, \tag{B.1}$$

*with probability at least $1 - e^{-d}$. Here, $C_1, C_2 > 0$ are numerical constants independent of $n, d, M, \delta$.*

*Proof.* Let $z_i = \sqrt{y_i} x_i$, which is well defined as $y_i \geq 0$. We denote by $\|\cdot\|_{\psi_2}$ the sub-gaussian norm of a random vector, defined as in (Vershynin, 2018, Definition 3.4.1). Then, $z_i$ is a sub-gaussian vector with its sub-gaussian norm $\|z_i\|_{\psi_2}$ bounded by $C_2 \sqrt{M}$, as

$$\|z_i\|_{\psi_2} = \sup_{u \in \mathcal{S}^{d-1}} \|\langle z_i, u \rangle\|_{\psi_2} = \sup_{u \in \mathcal{S}^{d-1}} \|\langle \sqrt{y_i} x_i, u \rangle\|_{\psi_2} \leq C_1 \sqrt{M} \sup_{u \in \mathcal{S}^{d-1}} \|\langle x_i, u \rangle\|_{\psi_2} = C_2 \sqrt{M},$$

due to the fact that $\sqrt{\sigma(z)} \leq C_1 \sqrt{M}$ for all $z$, and each $x_i$ has constant sub-gaussian norm. Then, a direct application of (Vershynin, 2018, Exercise 4.7.3) gives that, with probability at least $1 - e^{-d}$,

$$\|D_n - \mathbb{E}[D_n]\| \leq C_3 M \delta^{-1/2} \|\mathbb{E}[D_n]\| \leq C_4 \delta^{-1/4} \|\mathbb{E}D_n\|,$$

where the last inequality follows from the fact that $\delta \geq CM^4$. We now move to calculating $\mathbb{E}[D_n]$. To that aim, we write

$$\mathbb{E}[D_n] = \mathbb{E}[\sigma(\langle x, \theta^\star \rangle) x x^\top] = \mathbb{E}[\sigma(\langle Ux, \theta^\star \rangle) Ux (Ux)^\top] = U \left( \mathbb{E}[\sigma(\langle x, e_1 \rangle) x x^\top] \right) U^\top,$$

where we have used the rotational invariance of the Gaussian distribution and that $U$ is an orthogonal matrix such that $U^\top \theta^\star = e_1$. As $\sigma$ is even, all the off-diagonal entries of $\mathbb{E}[\sigma(\langle x, e_1 \rangle) x x^\top]$ are 0. Moreover, we can set

$$c_\sigma^1 := \left( \mathbb{E}[\sigma(\langle x, e_1 \rangle) x x^\top] \right)_{1,1} = \mathbb{E}[\sigma(z) z^2], \text{ and } c_\sigma^2 := \left( \mathbb{E}[\sigma(\langle x, e_1 \rangle) x x^\top] \right)_{i,i} = \mathbb{E}[\sigma(z) w^2], \quad \forall i \in \{2, \ldots, d\},$$

where $z \sim \mathcal{N}(0, 1)$ and $w \sim \mathcal{N}(0, 1)$ are independent random variables. Assuming that the bounds on $c_\sigma^1$ and $c_\sigma^2$ from the statement of this proposition hold, we get that $\|\mathbb{E}[D_n]\| \leq c_\sigma^1 = 3 + Ce^{-M/3}$, from which the claim in (B.1) follows. Thus, it is only left to prove the bounds on $c_\sigma^1$ and $c_\sigma^2$, and we will do these calculations for $\sigma$ defined in (3.6) and $\sigma$ defined in (4.4) separately.

**Bounding $c_\sigma^1$ and $c_\sigma^2$, for $\sigma$ defined in** (3.6). We first bound $c_\sigma^1$. By definition it holds that

$$c_\sigma^1 = \mathbb{E}[\sigma(z)z^2] = \mathbb{E}\left[z^2 \int_0^{z^2} \varphi(u)du\right] = \int_0^\infty \varphi(u)\,\mathbb{E}\left[z^2 \mathbf{1}\left\{u < z^2\right\}\right]du = \int_0^{2M} \varphi(u)\,\mathbb{E}\left[z^2 \mathbf{1}\left\{u < z^2\right\}\right]du,$$

where we have used the Fubini-Tonelli theorem and the fact that $\varphi(u) = 0$ for $u > 2M$. We can express the $\mathbb{E}\left[z^2 \mathbf{1}\left\{u < z^2\right\}\right]$ term via the erfc function, as

$$\mathbb{E}\left[z^2 \mathbf{1}\left\{u < z^2\right\}\right] = \operatorname{erfc}\left(\sqrt{u/2}\right) + \sqrt{2u/\pi}e^{-u/2} =: g(u),$$

which follows from an integration by parts. Also, a direct calculation gives

$$\int_0^a g(u) = 3\operatorname{erf}\left(\sqrt{a/2}\right) + a\operatorname{erfc}\left(\sqrt{a/2}\right) - \frac{6}{\sqrt{\pi}}e^{-a/2}\sqrt{a/2}. \tag{B.2}$$

Next, as $0 \le \varphi \le 1$, $\varphi(u) \equiv 1$ on $[0, M]$, and $g(u)$ is positive for $u \ge 0$, we get the following bounds

$$\int_0^M g(u)du \le \int_0^{2M} \varphi(u)\,g(u)du \le \int_0^{2M} g(u)du. \tag{B.3}$$

By plugging this in (B.2) and using that $\operatorname{erfc}(\sqrt{M/2}) \le e^{-M/2}$, we get

$$|c_\sigma^1 - 3| \le Ce^{-M/3}.$$

Turning to $c_\sigma^2$, by definition it holds

$$c_\sigma^2 = \mathbb{E}[\sigma(z)w^2] = \mathbb{E}\left[w^2 \int_0^{z^2} \varphi(u)du\right] = \int_0^\infty \phi(u)\,\mathbb{E}\left[w^2 \mathbf{1}\left\{u < z^2\right\}\right]du = \int_0^{2M} \varphi(u)\,\mathbb{E}\left[\mathbf{1}\left\{u < z^2\right\}\right]du,$$

where we have used the Fubini-Tonelli theorem, the fact that $\varphi(u) = 0$ for $u > 2M$, and that $z$ and $w$ are independent random variables with $\mathbb{E}[w^2] = 1$. As before, we relate $\mathbb{E}\left[z^2 \mathbf{1}\left\{u < z^2\right\}\right]$ to the erfc function:

$$\mathbb{E}\left[\mathbf{1}\left\{u < z^2\right\}\right] = \operatorname{erfc}\left(\sqrt{u/2}\right).$$

Plugging this back into the integral, doing the bounds as in (B.3) and using that $\operatorname{erfc}(\sqrt{M/2}) \le e^{-M/2}$, we conclude that

$$|c_\sigma^2 - 1| \le Ce^{-M/3}.$$

**Bounding $c_\sigma^1$ and $c_\sigma^2$, for $\sigma$ defined in** (4.4). We first calculate $c_\sigma^1$. A direct calculation gives

$$
\begin{aligned}
|c_\sigma^1 - 3| = \left|\mathbb{E}[\sigma(z)z^2] - 3\right| &= \left|\mathbb{E}\left[z^4 \mathbf{1}\left\{z^2 \le M\right\}\right] + M\mathbb{E}\left[z^2 \mathbf{1}\left\{z^2 > M\right\}\right] - 3\right| \\
&= \left|-\mathbb{E}\left[z^4 \mathbf{1}\left\{z^2 > M\right\}\right] + M\mathbb{E}\left[z^2 \mathbf{1}\left\{z^2 > M\right\}\right]\right| \\
&= \left|(M - 3)\operatorname{erfc}\left(\sqrt{M/2}\right) - \frac{6}{\sqrt{\pi}}\sqrt{\frac{M}{2}}e^{-M/2}\right| \\
&\le Ce^{-M/3}.
\end{aligned}
$$

Similarly, for $c_\sigma^2$, we have that

$$
\begin{aligned}
|c_\sigma^2 - 1| = |\mathbb{E}[\sigma(z)w^2] - 1| &= \left|\mathbb{E}\left[z^2 \mathbf{1}\left\{z^2 \le M\right\}\right] + M\mathbb{E}\left[\mathbf{1}\left\{z^2 > M\right\}\right] - 1\right| \\
&= \left|-\mathbb{E}\left[z^2 \mathbf{1}\left\{z^2 > M\right\}\right] + M\mathbb{E}\left[\mathbf{1}\left\{z^2 > M\right\}\right]\right| \\
&= \left|(M - 1)\operatorname{erfc}\left(\sqrt{M/2}\right) - \sqrt{\frac{2M}{\pi}}e^{-M/2}\right| \\
&\le Ce^{-M/3}.
\end{aligned}
$$

$\square$

**Lemma B.2.** *It holds that*

$$\sup_{\theta \in \mathcal{S}^{d-1}} \frac{1}{n} \sum_{i=1}^{n} \mathbf{1}\{\langle x_i, \theta \rangle^2 > M\} \leq C(e^{-M/2} + \delta^{-1/2} \log \delta),$$

*with probability at least $1 - e^{-d}$, where $C$ is a numerical constant independent of $n, d, M, \delta$.*

*Proof.* Given $\theta \in \mathcal{S}^{d-1}$, let us introduce the labeling function $f_\theta : \mathbb{R}^d \to \{0, 1\}$ as

$$f_\theta(x) := \mathbf{1}\{\langle x, \theta \rangle^2 > M\},$$

and let $\mathcal{F}$ be the set of all possible labeling functions, i.e., $\mathcal{F} := \{f_\theta : \theta \in \mathcal{S}^{d-1}\}$. We denote by $P$ the expectation under $\mathcal{N}(0, I_d)$ and by $P_n$ the empirical measure, that is,

$$P f_\theta := \mathbb{E}\left[\mathbf{1}\{\langle x_i, \theta \rangle^2 > M\}\right], \qquad P_n f_\theta := \frac{1}{n} \sum_{i=1}^{n} \mathbf{1}\{\langle x_i, \theta \rangle^2 > M\}.$$

For any fixed $\theta \in S^{d-1}$, it holds that $\langle x, \theta \rangle \sim \mathcal{N}(0, 1)$, hence

$$P f_\theta = \mathbb{P}\left[|\langle x, \theta \rangle| > \sqrt{M}\right] \leq C e^{-M/2}. \tag{B.4}$$

Therefore,

$$\sup_\theta P_n f_\theta \leq \sup_\theta \left(P f_\theta + |P_n f_\theta - P f_\theta|\right) \leq C e^{-M/2} + \sup_\theta |P_n f_\theta - P f_\theta|. \tag{B.5}$$

We denote the uniform deviation as

$$\Delta_n := \sup_\theta |P_n f_\theta - P f_\theta| = \sup_{f \in \mathcal{F}} |P_n f - P f|.$$

For each $f \in \mathcal{F}$, the random variables $f(x_i) \in \{0, 1\}$ are i.i.d., so Hoeffding's inequality gives, for any $\varepsilon > 0$,

$$\mathbb{P}[|P_n f - P f| > \varepsilon] \leq 2 e^{-2n\varepsilon^2}. \tag{B.6}$$

Let $\Pi_\mathcal{F}(n)$ be the number of distinct labelings $\{(f(x_1), \ldots, f(x_n)) : f \in \mathcal{F}\}$ realized on the sample. By the Sauer–Shelah lemma (Vershynin, 2018, Theorem 8.3.16), if $v := \mathrm{VC}(\mathcal{F})$,

$$\Pi_\mathcal{F}(n) \leq \left(\frac{en}{v}\right)^v, \tag{B.7}$$

where $\mathrm{VC}(\mathcal{F})$ denotes the $VC$ dimension of the set $\mathcal{F}$.

We proceed to give a lower and upper bound on $v = \mathrm{VC}(\mathcal{F})$. First, note that the set $\left\{\sqrt{M}de_i\right\}_{i=1}^{d}$, where $e_i$ is the $i$-th element of the canonical orthonormal basis of $\mathbb{R}^d$, is shattered by $\mathcal{F}$. Thus, a direct lower bound is $d$. For the upper bound, observe that each set $\{x : f_\theta(x) = 1\}$ can be expressed as the union of two parallel halfspaces orthogonal to $\theta$. This is a subset of the set of unions of any two halfspaces in $\mathbb{R}^d$, which we denote by $\mathcal{H}_2$. Then, by monotonicity of the $VC$ dimension, we have

$$\mathrm{VC}(\mathcal{F}) \leq \mathrm{VC}(\mathcal{H}_2).$$

If we denote by $\mathcal{H}$ the set of all halfspaces in $\mathbb{R}^d$, then by (Goldberg & Jerrum, 1993, Theorem 2.2) it holds for some constant $c > 1$

$$\mathrm{VC}(\mathcal{H}_2) \leq c \cdot d,$$

since each element of $\mathcal{H}_2$ can be expressed via two polynomial inequalities of degree 1. Thus, $d \leq \mathrm{VC}(\mathcal{F}) \leq c \cdot d$. By plugging this bound into (B.7), we get

$$\Pi_\mathcal{F}(n) \leq \left(\frac{en}{d}\right)^{cd} = (e\delta)^{cd}.$$

Taking a union bound over these labelings into (B.6) implies

$$\mathbb{P}[\Delta_n > \varepsilon] \leq 2\,\Pi_{\mathcal{F}}(n)\,e^{-2n\varepsilon^2} \leq 2(e\delta)^{cd}e^{-2n\varepsilon^2}.$$

Solving for $\varepsilon$ at confidence level $1 - \eta$ yields

$$2(e\delta)^{cd}e^{-2n\varepsilon^2} \leq \eta, \quad \text{then} \quad \varepsilon \geq \sqrt{\frac{c\log\delta}{2\delta} + \frac{\log(2/\eta)}{2n}}.$$

Thus, with probability at least $1 - \eta$,

$$\Delta_n \leq \sqrt{\frac{c\log\delta}{2\delta} + \frac{\log(2/\eta)}{2n}}.$$

Combining with (B.5) yields

$$\sup_{\theta \in S^{d-1}} P_n f_\theta \leq Ce^{-M/2} + \sqrt{\frac{c\log\delta}{2\delta} + \frac{\log(2/\eta)}{2n}} \quad \text{w.p.} \geq 1 - \eta.$$

Taking $\eta = e^{-d}$, we get that

$$\sup_{\theta \in S^{d-1}} P_n f_\theta \leq C(e^{-M/2} + \delta^{-1/2}\log\delta),$$

with probability $1 - e^{-d}$. $\qquad\qquad\qquad\qquad\qquad\qquad\qquad\qquad\qquad\qquad\qquad\qquad\qquad\qquad\qquad$ □

We can now state the main spectral convergence result of this section.

**Theorem B.3.** *For M large enough constant (independent of n, d) and $\delta \geq CM^4$, it holds with probability at least $1 - \exp(-cd^{1/5})$ that*

$$\inf_\theta |\langle v_1(A(\theta)), \theta^\star\rangle| \geq 1 - C(e^{-M/2} + \delta^{-1/5}), \tag{B.8}$$

*where $c, C \geq 0$ are numerical constant independent of $n, d, M, \delta$. Moreover,*

$$\inf_{\theta \in \mathcal{S}^{d-1}} \lambda_1(A(\theta)) \geq 2c_\sigma^1 - C(e^{-M/2} + \delta^{-1/5}), \qquad \sup_{\theta \in \mathcal{S}^{d-1}} \lambda_2(A(\theta)) \leq 2c_\sigma^2 + C\delta^{-1/4},$$

*for $c_\sigma^1$ and $c_\sigma^2$ constants from Proposition B.1.*

*Proof.* By definition we have

$$A(\theta) := \frac{2}{n}\sum_{i=1}^n y_i x_i x_i^\top \varphi(\langle x_i, \theta\rangle^2).$$

Let us now introduce matrices $A^\star$ and $B(\theta)$ as

$$A^\star := \frac{2}{n}\sum_{i=1}^n y_i x_i x_i^\top, \qquad B(\theta) := \frac{2}{n}\sum_{i=1}^n y_i x_i x_i^\top \left(1 - \varphi(\langle x_i, \theta\rangle^2)\right).$$

Then, it holds that

$$A^\star - B(\theta) = A(\theta) \preceq A^\star. \tag{B.9}$$

By definition, we have

$$\theta^{\star\top} A(\theta)\theta^\star = \theta^{\star\top} A^\star \theta^\star - \theta^{\star\top} B(\theta)\theta^\star.$$

From Proposition B.1, we have that

$$|\theta^{\star\top} A^\star \theta^\star - 2c_\sigma^1| \leq C\delta^{-1/4}.$$

On the other hand, from Cauchy-Schwartz, it follows that

$$\theta^{\star\top} B(\theta)\theta^{\star} = \frac{2}{n} \sum_{i=1}^{n} y_i \langle x_i, \theta^{\star} \rangle^2 \left(1 - \varphi(\langle x_i, \theta \rangle^2)\right)$$

$$\leq \sqrt{\frac{2}{n} \sum_{i=1}^{n} y_i^2 \langle x_i, \theta^{\star} \rangle^4} \cdot \sqrt{\frac{2}{n} \sum_{i=1}^{n} \left(1 - \varphi(\langle x_i, \theta \rangle^2)\right)} \qquad (B.10)$$

$$\leq \sqrt{\frac{2}{n} \sum_{i=1}^{n} y_i^2 \langle x_i, \theta^{\star} \rangle^4} \cdot \sqrt{\frac{2}{n} \sum_{i=1}^{n} \mathbf{1}\{\langle x_i, \theta \rangle^2 > M\}},$$

The final inequality follows from the fact that $\varphi(u) \in [0, 1]$ for any $u \in \mathbb{R}$, and specifically $\varphi(u) = 1$ for $|u| \leq M$. Let us now give bounds on both of these terms. First, note that by definition of $y_i = \sigma(\langle x_i, \theta^{\star} \rangle)$ it holds

$$\frac{2}{n} \sum_{i=1}^{n} y_i^2 \langle x_i, \theta^{\star} \rangle^4 \leq \frac{2}{n} \sum_{i=1}^{n} \langle x_i, \theta^{\star} \rangle^8 .$$

Moreover, since $\theta^{\star}$ is sampled from a unit sphere independent from all $x_i$, $\langle x_i, \theta^{\star} \rangle$ can be viewed as a random variable $q_i \sim \mathcal{N}(0, 1)$. Even though each $q_i^8$ is not sub-gaussian, we will manage to use known concentration results by splitting it into a bounded part and tail part. Namely, for all $i \in [n]$, let

$$w_i := q_i^8 \mathbf{1}\left\{q_i^8 \leq n^{4/5}\right\}, \text{ and } z_i := q_i^8 \mathbf{1}\left\{q_i^8 > n^{4/5}\right\}.$$

Then, by triangle inequality and union bound, it holds

$$\mathbb{P}\left(\left|\frac{1}{n} \sum_{i=1}^{n}(q_i^8 - \mathbb{E}\left[q_1^8\right])\right| \geq 106\right) \leq \mathbb{P}\left(\left|\frac{1}{n} \sum_{i=1}^{n}(w_i - \mathbb{E}\left[w_i\right])\right| \geq 106\right) + \mathbb{P}\left(\left|\frac{1}{n} \sum_{i=1}^{n}(z_i - \mathbb{E}\left[z_i\right])\right| \geq 106\right). \quad (B.11)$$

Since $w_i$ is bounded, it is subgaussian with the sub-gaussian norm $\|w_i\|_{\psi_2} \leq c n^{4/5}$, as per (Vershynin, 2018, Example 2.5.8). Then by Hoeffding's inequality (Vershynin, 2018, Theorem 2.6.2), we have

$$\mathbb{P}\left(\left|\frac{1}{n} \sum_{i=1}^{n} w_i - \mathbb{E}\left[w_i\right]\right| \geq 106\right) \leq \exp\left(-c n^{1/5}\right). \qquad (B.12)$$

Turning to the term $z_i$, by triangle inequality and union bound, for arbitrary $j \in [n]$, it holds that

$$\mathbb{P}\left(\left|\frac{1}{n} \sum_{i=1}^{n} z_i - \mathbb{E}\left[z_i\right]\right| \geq 106\right) \leq \sum_{i=1}^{n} \mathbb{P}\left(|z_i - \mathbb{E}\left[z_i\right]| \geq 106\right) = n \mathbb{P}\left(|z_j - \mathbb{E}\left[z_j\right]| \geq 106\right) \leq \exp\left(-c n^{1/5}\right) \quad (B.13)$$

The last bound follows from the chain of inequalities below,

$$\mathbb{P}\left(|z_j - \mathbb{E}\left[z_j\right]| \geq 106\right) \leq \mathbb{P}\left(|z_j| \geq 106 - \mathbb{E}\left[z_j\right]\right) \leq \mathbb{P}\left(|z_j| \geq 1\right) \leq \mathbb{P}\left(q_j \geq n^{1/10}\right) \leq \exp\left(-c n^{1/5}\right),$$

where we have used that $\mathbb{E}\left[z_j\right] \leq \mathbb{E}\left[q_j^8\right] = 105$ and the bound on the tail of a normal distribution (Vershynin, 2018, Proposition 2.1.2). Finally, by plugging in (B.12) and (B.13) into (B.11), for some constant $c$, we have

$$\frac{2}{n} \sum_{i=1}^{n} y_i^2 \langle x_i, \theta^{\star} \rangle^4 \leq 106,$$

with probability $1 - \exp(-c n^{1/5})$. Note that for $\delta > 1$, $1 - \exp(-c n^{1/5}) \leq 1 - \exp(-c d^{1/5})$
Let us turn our attention to the term $\frac{2}{n} \sum_{i=1}^{n} \mathbf{1}\{\langle x_i, \theta \rangle^2 > M\}$. An application of Lemma B.2 gives that, uniformly over $\theta$,

$$\frac{2}{n} \sum_{i=1}^{n} \mathbf{1}\{\langle x_i, \theta \rangle^2 > M\} \leq C(e^{-M/2} + \delta^{-1/2} \log \delta),$$

with probability at least $1 - e^{-d}$. Using the overwhelming probability bounds on these two terms, alongside the union bound, we get that for any $\epsilon > 0$, uniformly over $\theta$,

$$\theta^{\star\top} B(\theta)\theta^{\star} \leq C(e^{-M/2} + \delta^{-1/4+\epsilon}),$$

with probability $1 - \exp(-cd^{1/5})$. This means that

$$\theta^{\star\top} A(\theta)\theta^{\star} = \theta^{\star\top}(A^{\star} - B(\theta))\theta^{\star} \geq 2c_{\sigma}^1 - C(e^{-M/2} + \delta^{-1/5}), \tag{B.14}$$

with probability $1 - \exp(-cd^{1/5})$. Let us now denote $w = |\langle\theta^{\star}, v_1(A(\theta))\rangle|$. Then, it holds

$$\lambda_1(A(\theta))w^2 + \lambda_2(A(\theta))(1 - w^2) \geq \theta^{\star\top}(A(\theta))\theta^{\star}.$$

Thus,

$$w^2(\lambda_1(A(\theta)) - \lambda_2(A(\theta))) \geq \theta^{\star\top}(A(\theta))\theta^{\star} - \lambda_2(A(\theta)). \tag{B.15}$$

Due to (B.9) and Proposition B.1, it holds uniformly over $\theta$

$$\lambda_2(A(\theta)) \leq \lambda_2(A^{\star}) = 2c_{\sigma}^2 + C\delta^{-1/4}.$$

Taking $\sup_{\theta\in\mathcal{S}^{d-1}}$ on both sides gives that, with probability $1 - \exp(-cd^{1/5})$,

$$\sup_{\theta\in\mathcal{S}^{d-1}} \lambda_2(A(\theta)) \leq 2c_{\sigma}^2 + C\delta^{-1/4}.$$

From Proposition B.1,(B.9) and (B.14), with probability at least $1 - \exp(-cd^{1/5})$, we have that, for large enough $n$ and $d$,

$$2c_{\sigma}^1 + C(e^{-M/2} + \delta^{-1/4}) = \lambda_1(A^{\star}) \geq \lambda_1(A(\theta)) \geq \theta^{\star\top}A(\theta)\theta^{\star} = 2c_{\sigma}^1 - C(e^{-M/2} + \delta^{-1/5}), \tag{B.16}$$

which implies that $\lambda_1(A(\theta)) \geq 2c_{\sigma}^1 - C(e^{-M/2} + \delta^{-1/5})$. Taking $\inf_{\theta\in\mathcal{S}^{d-1}}$ on both sides gives that, with probability $1 - \exp(-cd^{1/5})$,

$$\inf_{\theta\in\mathcal{S}^{d-1}} \lambda_1(A(\theta)) \geq 2c_{\sigma}^1 - C(e^{-M/2} + \delta^{-1/5}).$$

Putting all of this back into (B.15), it follows that, for $n$ and $d$ large enough,

$$w \geq \frac{2(c_{\sigma}^1 - c_{\sigma}^2) - C(e^{-M/2} + \delta^{-1/5})}{2(c_{\sigma}^1 - c_{\sigma}^2) + C(e^{-M/2} + \delta^{-1/5})} = 1 - C(e^{-M/2} + \delta^{-1/5}), \tag{B.17}$$

where the last equality follows from the bounds $|c_{\sigma}^1 - 3| \leq Ce^{-M/3}$ and $|c_{\sigma}^2 - 1| \leq Ce^{-M/3}$ from Proposition B.1. As bound in (B.17) holds uniformly for all $\theta$, we can take the infimum to conclude that, for large enough $n$ and $d$,

$$\inf_{\theta} |\langle\theta^{\star}, v_1(A(\theta))\rangle| \geq 1 - C(e^{-M/2} + \delta^{-1/5}),$$

with probability at least $1 - \exp(-cd^{1/5})$. $\qquad\qquad\qquad\qquad\qquad\qquad\qquad\qquad\qquad\qquad\qquad\qquad\square$

## B.2. Convergence results

Firstly, for any initial condition $\theta(0) = \theta_0 \in \mathcal{S}^{d-1}$ and any $t \in \mathbb{R}$, the point $\theta(t) \in \mathcal{S}^{d-1}$ indeed exists and is unique. This is due to the fact that the RHS of (3.8) is a smooth $C^{\infty}$ vector field, so uniqueness and existence follows from (Lee, 2003)[Chapter 9]. Furthermore, there exists a smooth $C^{\infty}$ flow map $\Phi : \mathbb{R} \times \mathcal{S}^{d-1} \to \mathcal{S}^{d-1}$ such that, for any $\theta(0) = \theta_0 \in \mathcal{S}^{d-1}$,

$$\Phi(t, \theta_0) = \theta(t). \tag{B.18}$$

Due to uniqueness of the solution of the ODE in (3.8), it holds that $\Phi(t_2, \theta(t_1)) = \theta(t_1 + t_2)$, for arbitrary $t_1, t_2 \in \mathbb{R}$ and initial condition $\theta(0) \in \mathcal{S}^{d-1}$. For an arbitrary $t \in \mathbb{R}$ and $\theta \in \mathcal{S}^{d-1}$ we will also use the notation $\Phi_t(\theta)$, interchangeably with $\Phi(t, \theta)$.

Note that the stationary points of the flow will be exactly the ones for which $\nabla_{\mathcal{S}^{d-1}} \widehat{\mathcal{L}}(\theta(t)) = 0$. Writing this out, we get that $\theta_s$ is a stationary point of the flow if and only if

$$(I - \theta_s \theta_s^\top) A(\theta_s) \theta_s = 0,$$

which happens if and only if $\theta_s$ is an eigenvector of $A(\theta_s)$. Furthermore, the flow converges to a unique stationary point $\theta_\infty$ that, due to the fact that it is a local minimizer, must also be the principal eigenvector of $A(\theta_\infty)$. We show this result in the next two propositions.

**Proposition B.4** (Convergence of the spherical gradient flow). *Consider the flow as defined in* (3.8), *i.e.,*

$$\frac{d\theta(t)}{dt} = (I - \theta(t)\theta(t)^\top) A(\theta(t))\theta(t), \qquad A(\theta) = \frac{2}{n} \sum_{i=1}^n y_i x_i x_i^\top \varphi(\langle x_i, \theta \rangle^2),$$

*initialized at $\theta_0 \in \mathcal{S}^{d-1}$. Then:*

*a) The loss $\widehat{\mathcal{L}}(\theta) = -\frac{1}{n} \sum_i y_i \sigma(\langle x_i, \theta \rangle)$ is a Lyapunov function, i.e.*

$$\frac{d}{dt} \widehat{\mathcal{L}}(\theta(t)) = -\left\| (I - \theta\theta^\top) A(\theta)\theta \right\|^2 \le 0.$$

*b) The trajectory converges to a stationary point $\theta_\infty$, satisfying $\nabla_{\mathcal{S}^{d-1}} \widehat{\mathcal{L}}(\theta_\infty) = 0$, that is,*

$$(I - \theta_\infty \theta_\infty^\top) A(\theta_\infty)\theta_\infty = 0, \quad \text{or equivalently,} \quad A(\theta_\infty)\theta_\infty = \lambda_\infty \theta_\infty, \text{ for some } \lambda_\infty.$$

*Proof.* We first compute the time derivative of the loss,

$$\frac{d}{dt} \widehat{\mathcal{L}}(\theta(t)) = \nabla_{\mathcal{S}^{d-1}} \widehat{\mathcal{L}}(\theta(t))^\top \frac{d\theta(t)}{dt} = -\|\nabla_{\mathcal{S}^{d-1}} \widehat{\mathcal{L}}(\theta(t))\|^2 \le 0,$$

which proves the statement of *a)*.

From *a)*, it directly follows that $\widehat{\mathcal{L}}(\theta(t))$ is non-increasing in $t$. Note that $\widehat{\mathcal{L}}(\theta(t))$ is also bounded below, due to the fact that the function $\sigma(z)$, as defined in (3.6), is bounded. Thus, $\widehat{\mathcal{L}}(\theta(t))$ has a limit at infinity, which we define as

$$\widehat{\mathcal{L}}_\infty := \lim_{t \to \infty} \widehat{\mathcal{L}}(\theta(t)). \tag{B.19}$$

We now prove that $\theta(t)$ has a unique limit for $t \to \infty$, by proving that $\theta(t)$ is Cauchy at infinity. Towards that end, let $\epsilon > 0$ be arbitrary. Then we choose $C > 0$ such that for all $t \ge C$ it holds $\left| \widehat{\mathcal{L}}(\theta(t)) - \widehat{\mathcal{L}}_\infty \right| \le \epsilon^2/2$. Note that this is possible due to (B.19). Let $\dot{\theta}(t) = \frac{d\theta(t)}{dt}$. Then, for any $t_1, t_2 \ge C$, it holds

$$\begin{aligned}
\|\theta(t_1) - \theta(t_2)\|^2 &= \left\| \int_{t_1}^{t_2} \dot{\theta}(t) dt \right\|^2 \\
&\le \left( \int_{t_1}^{t_2} \|\dot{\theta}(t)\| dt \right)^2 \\
&\le \int_{t_1}^{t_2} \|\dot{\theta}(t)\|^2 dt \\
&= \int_{t_1}^{t_2} \|\nabla_{\mathcal{S}^{d-1}} \widehat{\mathcal{L}}(\theta(t))\|^2 \, dt \\
&= \int_{t_1}^{t_2} -\frac{d}{dt} \widehat{\mathcal{L}}(\theta(t)) \, dt \\
&= \widehat{\mathcal{L}}(\theta(t_1)) - \widehat{\mathcal{L}}(\theta(t_2)) = (\widehat{\mathcal{L}}(\theta(t_1)) - \widehat{\mathcal{L}}_\infty) + (\widehat{\mathcal{L}}_\infty - \widehat{\mathcal{L}}(\theta(t_2))) \le \epsilon^2,
\end{aligned}$$

proving that $\theta(t)$ is indeed Cauchy at infinity. Since the sphere is complete, as a closed subset in $\mathbb{R}^d$, every Cauchy sequence has a limit on the sphere. This implies that the function $\theta(t)$ has a limit at infinity that lies on the sphere, which we will denote by $\theta_\infty$. We now prove that $\|\nabla_{\mathcal{S}^{d-1}}\widehat{\mathcal{L}}(\theta_\infty)\| = 0$. Integrating $\|\nabla_{\mathcal{S}^{d-1}}\widehat{\mathcal{L}}(\theta(t))\|$ over $[0, \infty)$ yields

$$\int_0^\infty \|\nabla_{\mathcal{S}^{d-1}}\widehat{\mathcal{L}}(\theta(t))\|^2 \, dt = \int_0^\infty -\frac{d}{dt}\widehat{\mathcal{L}}(\theta(t)) \, dt = \widehat{\mathcal{L}}(\theta(0)) - \widehat{\mathcal{L}}_\infty < \infty.$$

By continuity we have that $\lim_{t\to\infty} \nabla_{\mathcal{S}^{d-1}}\widehat{\mathcal{L}}(\theta(t)) = \nabla_{\mathcal{S}^{d-1}}\widehat{\mathcal{L}}(\theta_\infty)$, so it must be $\|\nabla_{\mathcal{S}^{d-1}}\widehat{\mathcal{L}}(\theta_\infty)\| = 0$ or otherwise the integral would diverge. This concludes the proof of *b)*. □

**Proposition B.5** (Convergence to principal eigenvector)**.** *Let $\theta_\infty$ be a stationary point from Proposition B.4 to which the flow from (3.8) converges. Then, $\theta_\infty$ is the principal eigenvector of $A(\theta_\infty)$, with probability at least $1 - \exp(-cd^{1/5})$ over sampling of $x_i$, and almost surely over the sampling of $\theta_0$.*

*Proof.* We follow the approach developed in (Panageas & Piliouras, 2017) for standard gradient descent in the plane, with suitable adjustments stemming from our analysis of the gradient flow on the sphere. Towards that end, let us first define the set of all stationary points $\theta_s$ that are not principal eigenvectors of their corresponding matrices $A(\theta_s)$ as

$$S := \left\{\theta_s \in \mathcal{S}^{d-1} : \exists \lambda_s \text{ s.t. } A(\theta_s) = \lambda_s \theta_s \text{ and } \lambda_s < \lambda_1(A(\theta_s))\right\}. \tag{B.20}$$

We will prove that the probability that appropriate $\theta_\infty \in S$ is 0, over the sampling of $\theta_0$. Thus, we introduce the set of all initial conditions whose corresponding trajectories converge to a point in $S$ as

$$C_S := \left\{\hat{\theta} \in \mathcal{S}^{d-1} : \lim_{t\to\infty} \theta(t) = \theta_\infty \in S, \text{ for the flow (3.8) with } \theta(0) = \hat{\theta}\right\}.$$

We will use the center and stable manifold theorem, which characterizes the dimension of the manifold of points that could eventually converge to the stationary point of interest by looking at the linearized flow around this point. For convenience we restate the theorem here (p. 65 of (Shub, 1987)) adjusted for our use as in (Panageas & Piliouras, 2017).

**Theorem B.6** (Center and Stable manifold, Theorem 9 of (Panageas & Piliouras, 2017))**.** *Let $\tilde{\theta}_s$ be a fixed point of the $C^r$ local diffeomorphism $\tilde{\Phi}_T : U \to \mathbb{R}^n$, where $U$ is an open neighborhood of the point $\tilde{\theta}_s$ in $\mathbb{R}^n$, and $r \geq 1$. Let $E^s \oplus E^c \oplus E^u$ be the invariant splitting of $\mathbb{R}^n$ into the generalized eigenspaces of the differential $d_\theta\tilde{\Phi}_T(\tilde{\theta}_s)$ corresponding to eigenvalues of absolute value less than one, equal to one, and larger than one, respectively. To the $d_\theta\tilde{\Phi}_T(\tilde{\theta}_s)$ invariant space $E^u \oplus E^c$ there is associated a local $\tilde{\Phi}_T$ invariant $C^r$ embedded disc $W_{loc}^{sc\,2}$ of dimension $\dim(E^u \oplus E^c)$, and a ball $B$ around $\tilde{\theta}_s$ such that*

$$\tilde{\Phi}_T(W_{loc}^{sc}) \cap B \subseteq W_{loc}^{sc}. \text{ If } \tilde{\Phi}_T^n(x) \in B \text{ for all } n \geq 0, \text{ then } x \in W_{loc}^{sc}.$$

We now explain how this theorem applies to dynamical systems defined on the sphere. Let $\Phi_T$ be a local diffeomorphism on the sphere, and $\theta_s$ one of its fixed points. Let $U_{\theta_s} \subseteq \mathcal{S}^{d-1}$ be an open neighborhood of $\theta_s$. Since $\mathcal{S}^{d-1}$ is a smooth manifold, there exist local diffeomorphic charts

$$\phi : U \to U_{\theta_s}, \qquad \Psi : U_{\theta_s} \to \mathbb{R}^{d-1},$$

where $U \subseteq \mathbb{R}^{d-1}$ is an open neighborhood of $\tilde{\theta}_s := \phi^{-1}(\theta_s)$. Moreover, the charts may be chosen such that

$$D\phi(\theta_s) = I_{d-1}, \qquad D\psi(\theta_s) = I_{d-1},$$

after appropriate identification of $T_{\theta_s}$ with $\mathbb{R}^{d-1}$. Define the local representation of $\Phi_T$ in coordinates as

$$\tilde{\Phi}_T(\tilde{\theta}) = \Psi \circ \Phi_T \circ \Psi(\tilde{\theta}_s).$$

Then $\tilde{\theta}_s$ is the fixed point of $\tilde{\Phi}_T$, and under the above identification of tangent spaces,

$$d_\theta\tilde{\Phi}_T(\tilde{\theta}_s) = d_\theta\Phi_T(\theta_s).$$

Consequently, the hypothesis and conclusion of the Euclidean Center and Stable manifold theorem apply verbatim to sphere. We thus obtain the following result.

---

[2] $W_{loc}^{sc}$ denotes the local center stable manifold of the point $\tilde{\theta}_s$.

**Theorem B.7** (Center and Stable manifold, adjusted to sphere). *Let $\theta_s$ be a fixed point of the $C^r$ local diffeomorphism $\Phi_T : U \to V$, where $U$ and $V$ are open neighborhood of the point $\theta_s$ in $\mathcal{S}^{d-1}$, and $r \geq 1$. Let $E^s \oplus E^c \oplus E^u$ be the invariant splitting of $\mathbb{R}^n$ into the generalized eigenspaces of the differential $d_\theta \Phi_T(\theta_s)$ corresponding to eigenvalues of absolute value less than one, equal to one, and larger than one, respectively. To the $d_\theta \Phi_T(\theta_s)$ invariant space $E^u \oplus E^c$ there is associated a local $\Phi_T$ invariant $C^r$ embedded disc $W_{loc}^{sc} \subseteq \mathcal{S}^{d-1}$ of dimension $\dim(E^u \oplus E^c)$, and an open neighborhood ball $U_{\theta_s}$ around $\theta_s$ such that*

$$\Phi_T(W_{loc}^{sc}) \cap U_{\theta_s} \subseteq W_{loc}^{sc}. \text{ If } \Phi_T^n(x) \in U_{\theta_s} \text{ for all } n \geq 0, \text{ then } x \in W_{loc}^{sc}. \tag{B.21}$$

Let us fix $T = 1$ (any positive constant would suffice), and consider the local diffeomorphism $\Phi_1(\theta)$. By definition, any $\theta_s \in S$ is a fixed point of $\Phi_1$. Thus, for each $\theta_s$ there exists an appropriate open neighborhood $U_{\theta_s}$ and a locally invariant center-stable manifold $W_{loc}^{sc}(\theta_s)$ of dimension $\dim(E^u(\theta_s) \oplus E^c(\theta_s))$, such that (B.21) holds on $U_{\theta_s}$. We denote by $U_S$ the union of all open spaces

$$U_S := \bigcup_{\theta_s \in S} U_{\theta_s}.$$

Take an arbitrary element $\hat\theta \in C_S$. By definition of $C_S$, there exists $\theta_s \in S$ such that

$$\lim_{t \to \infty} \Phi(t, \hat\theta) = \theta_s.$$

It follows that for some $i \in \mathbb{N}$ it holds $\Phi_1^n(\Phi(i, \hat\theta)) \in U_{\theta_s}$, for all $n \geq 0$. Then, applying Theorem B.7 we conclude that $\Phi(i, \hat\theta) \in W_{loc}^{sc}(\theta_s)$. Therefore,

$$C_s \subset \bigcup_{\theta_s \in S} \bigcup_{t=1}^{\infty} \Phi_i^{-1}(W_{loc}^{sc}(\theta_s)).$$

At this stage we have not proved that $S$ is countable, therefore it is not immediate that $U_s$ is a countable union of open sets. However, the sphere $\mathcal{S}^{d-1}$ is hereditarily Lindelöf (Willard, 2012). This means that every open cover of an open subset has a countable subcover. It follows that the open cover $U_s$ admits a countable subcover, i.e., there exists a sequence $(\theta_m)_{m=1}^\infty \subset S$ such that

$$U_s = \bigcup_{j=1}^{\infty} U_{\theta_j}.$$

Since each $\theta_s \in S$ belongs to some $U_{\theta_j}, j \in \mathbb{N}$, the above argument applies with $\theta_j$ in place of $\theta_s$, yielding

$$C_s \subset \bigcup_{j=1}^{\infty} \bigcup_{i=1}^{\infty} \Phi_i^{-1}(W_{loc}^{sc}(\theta_j)). \tag{B.22}$$

It remains to show that each center-stable manifold $W_{loc}^{sc}(\theta_j)$ has measure 0 on the sphere. This follows once we establish that for each $j \in \mathbb{N}$,

$$\dim(E^u(\theta_j) \oplus E^c(\theta_j)) < d - 1,$$

which will be proved in the rest of the argument. Towards this end, we fix the coordinates to the canonical basis induced by the standard embedding $\mathcal{S}^{d-1}$ into $\mathbb{R}^d$, and refer interchangeably to the differential and its Jacobian matrix representation in these coordinates as $d\Phi_1(\theta_j)$. We then state a claim relating the Jacobian of the flow to the Hessian of the loss function at stationary points, i.e.,

$$d\Phi_1(\theta_j) = e^{-H_{\hat{\mathcal{L}}}^{\mathcal{S}^{d-1}}(\theta_j)}, \tag{B.23}$$

where $H_{\hat{\mathcal{L}}}^{\mathcal{S}^{d-1}}(\theta_j)$ is the spherical Hessian of $\hat{\mathcal{L}}$ evaluated at $\theta_j$. Note that the spherical Hessian $H_{\hat{\mathcal{L}}}^{\mathcal{S}^{d-1}}(\theta)$ connects to the Euclidean Hessian $H_{\hat{\mathcal{L}}}^{\mathbb{R}^d}(\theta)$ via the formula

$$H_{\hat{\mathcal{L}}}^{\mathcal{S}^{d-1}}(\theta) = P_\theta H_{\hat{\mathcal{L}}}^{\mathbb{R}^d}(\theta) P_\theta - \left\langle \nabla_{\mathbb{R}^d} \hat{\mathcal{L}}(\theta), \theta \right\rangle \cdot P_\theta, \tag{B.24}$$

where $P_\theta := (I - \theta\theta^\top)$ denotes the orthogonal projection onto $T_\theta \mathcal{S}^{d-1}$. This identity follows from direct calculation, analogous to the derivation of the spherical gradient used to define the gradient flow on the sphere in (3.2).

**Proof of claim in** (B.23). By definition, for $\theta(0) = \theta_j$,

$$\frac{d}{dt}\Phi_t(\theta_j) = \frac{d}{dt}\theta(t) = -\nabla_{\mathcal{S}^{d-1}}\hat{\mathcal{L}}(\theta(t)) = -\nabla_{\mathcal{S}^{d-1}}\hat{\mathcal{L}}(\Phi_t(\theta_j)).$$

Since the flow map $\Phi_t(\theta) = \Phi(t,\theta)$ is smooth on its domain, as noted when defining it in (B.18), differentiation with respect to parameters $t$ and $\theta$ commutes, yielding

$$\frac{d}{dt}d_\theta\Phi_t(\theta_j) = d_\theta\left[-\nabla_{\mathcal{S}^{d-1}}\hat{\mathcal{L}}(\Phi_t(\theta_j))\right].$$

Applying the chain rule gives

$$d_\theta\left[-\nabla_{\mathcal{S}^{d-1}}\hat{\mathcal{L}}(\Phi_t(\theta_j))\right] = -H_{\hat{\mathcal{L}}}^{\mathcal{S}^{d-1}}(\Phi_t(\theta_j))d_\theta\Phi_t(\theta_j),$$

where $H_{\hat{\mathcal{L}}}^{\mathcal{S}^{d-1}}(\Phi_t(\theta_j))$ is the mentioned spherical Hessian of $\hat{\mathcal{L}}$ at the point $\Phi_t(\theta_j)$. Consequently,

$$\frac{d}{dt}d_\theta\Phi_t(\theta_j) = -H_{\hat{\mathcal{L}}}^{\mathcal{S}^{d-1}}(\Phi_t(\theta_j))d_\theta\Phi_t(\theta_j). \tag{B.25}$$

As $\Phi_0$ is the identity map on the sphere, we have the initial condition $d_\theta\Phi_0(\theta_j) = I$. Moreover, because $\theta_j$ is a fixed point of $\Phi_t$ for any $t$, it holds that $H_{\hat{\mathcal{L}}}(\Phi_t(\theta_j)) = H_{\hat{\mathcal{L}}}(\theta_j)$. Solving the ODE defined by (B.25) yields the unique solution

$$d_\theta\Phi_t(\theta_j) = e^{-tH_{\hat{\mathcal{L}}}(\theta_j)},$$

proving the claim. $\qquad\square$

By direct calculations, for any $\theta \in \mathcal{S}^{d-1}$, it holds

$$H_{\hat{\mathcal{L}}}^{\mathbb{R}^d}(\theta) = -A(\theta) - R(\theta), \tag{B.26}$$

where $R(\theta) := \frac{4}{n}\sum_{i=1}^n y_i \langle x_i, \theta\rangle^2 \varphi'(\langle x_i, \theta\rangle^2)x_i x_i^\top$. Taking $\theta = \theta_j$ and substituting (B.26) into (B.24) yields

$$\begin{aligned}
H_{\hat{\mathcal{L}}}^{\mathcal{S}^{d-1}}(\theta_j) &= (I - \theta_j\theta_j^\top)(-A(\theta_j) - R(\theta_j))(I - \theta_j\theta_j^\top) + \theta_j^\top A(\theta_j)\theta_j \cdot (I - \theta_j\theta_j^\top) \\
&= -A(\theta_j) + \lambda_{\theta_j}I - P_{\theta_j}R(\theta_j)P_{\theta_j},
\end{aligned}$$

where we have used the fact that $\theta_j$ is an eigenvector of $A(\theta_j)$, i.e. $A(\theta_j)\theta_j = \lambda_{\theta_j}\theta_j$. Let us denote by $\tilde{R}(\theta_j) = P_{\theta_j}R(\theta_j)P_{\theta_j}$. Substituting the derived expression for $H_{\hat{\mathcal{L}}}^{\mathcal{S}^{d-1}}(\theta_j)$ into (B.23) gives

$$d\Phi_1(\theta_j) = \exp\left(A(\theta_j) - \lambda_{\theta_j}I + \tilde{R}(\theta_j)\right).$$

By definition from Theorem B.7, the dimension of the center-stable manifold satisfies

$$\dim(E^u(\theta_j) \oplus E^c(\theta_j)) = \#\{\lambda_i : \lambda_i(d\Phi_1(\theta_j)) \leq 1\} = \#\left\{\lambda_i : \lambda_i\left(A(\theta_j) - \lambda_{\theta_j}I + \tilde{R}(\theta_j)\right) \leq 0\right\}.$$

Next, we show that this matrix has a strictly positive eigenvalue for every $\theta_j$. Using the lower bound on $\theta^{\star\top}A(\theta_j)\theta^\star$ from (B.16), one gets, uniformly over $\theta_j \in S$, with probability $1 - \exp(-cd^{1/5})$, for some constant $C$,

$$\begin{aligned}
\theta^{\star\top}\left(A(\theta_j) - \lambda_{\theta_j}I + \tilde{R}(\theta_j)\right)\theta^\star &\geq 2c_\sigma^1 - C(e^{-M/2} + \delta^{-1/5}) - \lambda_{\theta_j} + \theta^{\star\top}\tilde{R}(\theta_j)\theta^\star \\
&\geq 2(c_\sigma^1 - c_\sigma^2) - C(e^{-M/2} + \delta^{-1/5}) + \theta^{\star\top}\tilde{R}(\theta_j)\theta^\star,
\end{aligned}$$

where the last inequality comes from Theorem B.3, since $\theta_j \in S$ which by (B.20) directly implies $\lambda_{\theta_j} \leq \lambda_2(A(\theta_j))$.

We claim that, for some constant $C$ independent of $n, d, M, \delta$, with probability $1 - \exp(-cd^{1/5})$, it holds

$$\sup_{\theta_j \in S}\left|\theta^{\star\top}\tilde{R}(\theta_j)\theta^\star\right| \leq C(\sqrt{M}e^{-M/2} + \delta^{-1/8}). \tag{B.27}$$

**Proof of claim in** (B.27). First, notice $R(\theta)$ is a negative semi-definite matrix, since $\varphi'(\cdot) \leq 0$ and $y_i \geq 0$. As it holds $\tilde{R}(\theta_j) = P_{\theta_j} R(\theta_j) P_{\theta_j}$ we have

$$
\begin{aligned}
\left| \theta^{\star\top} \tilde{R}(\theta_j) \theta^\star \right| &\leq \left| \theta^{\star\top} P_{\theta_j} R(\theta_j) P_{\theta_j}(\theta_j) \theta^\star \right| \\
&\leq \left| \theta^{\star\top} (I - \theta_j \theta_j^\top) R(\theta_j)(I - \theta_j \theta_j^\top) \theta^\star \right| \\
&\leq \left| \theta^{\star\top} R(\theta_j) \theta^\star \right| + |2R_1| + |R_2|,
\end{aligned}
$$

where $R_1 = \theta^{\star\top} R(\theta_j) \theta_j \langle \theta_j, \theta^\star \rangle$ and $R_2 = \langle \theta_j, \theta^\star \rangle^2 \theta_j^\top R(\theta_j) \theta_j$. We bound each term separately. Firstly, by definition

$$
\begin{aligned}
\left| \theta^{\star\top} R(\theta_j) \theta^\star \right| &= \frac{4}{n} \sum_{i=1}^n y_i \langle x_i, \theta^\star \rangle^2 \langle x_i, \theta_j \rangle^2 \left| \varphi'(\langle x_i, \theta_j \rangle^2) \right| \\
&\leq \frac{c}{M} 2M \frac{1}{n} \sum_{i=1}^n y_i \langle x_i, \theta^\star \rangle^2 \mathbf{1}\{M < \langle x_i, \theta_j \rangle^2 < 2M\},
\end{aligned}
$$

where the last inequality follows from the properties of $\varphi$. Proceeding as in the proof of Theorem B.3, we use Cauchy-Schwartz as in (B.10), followed by Chebyshev inequality and Lemma B.2 to get

$$
\begin{aligned}
\sup_{\theta_j \in S} \frac{c}{n} \sum_{i=1}^n y_i \langle x_i, \theta^\star \rangle^2 \mathbf{1}\{M < \langle x_i, \theta_j \rangle^2 < 2M\} &\leq \sqrt{\frac{c}{n} \sum_{i=1}^n y_i^2 \langle x_i, \theta^\star \rangle^4} \cdot \sup_{\theta_j \in S} \sqrt{\frac{c}{n} \sum_{i=1}^n \mathbf{1}\{M < \langle x_i, \theta_j \rangle^2 < 2M\}} \\
&\leq c \cdot \sup_{\theta \in \mathcal{S}^{d-1}} \sqrt{\frac{c}{n} \sum_{i=1}^n \mathbf{1}\{\langle x_i, \theta \rangle^2 > M\}} \\
&\leq C(\delta^{-1/5} + e^{-M/2}), \tag{B.28}
\end{aligned}
$$

uniformly over $\theta_j \in S$, with probability $1 - \exp(-cd^{1/5})$. Hence,

$$
\sup_{\theta_j \in S} \left| \theta^{\star\top} \tilde{R}(\theta_j) \theta^\star \right| \leq C(\delta^{-1/5} + e^{-M/2}),
$$

uniformly over $\theta_j \in S$, with probability $1 - \exp(-cd^{1/5})$.

Similarly, for the term $R_1$, it holds that

$$
\begin{aligned}
\sup_{\theta_j \in S} |R_1| &= \sup_{\theta_j \in S} \left| \frac{4}{n} \sum_{i=1}^n y_i \langle x_i, \theta^\star \rangle \langle x_i, \theta_j \rangle^3 \varphi'(\langle x_i, \theta_j \rangle^2) \langle \theta_j, \theta^\star \rangle \right| \\
&\leq C(e^{-M/2} + \delta^{-1/4}) \frac{c}{M} (2M)^{3/2} \frac{1}{n} \sum_{i=1}^n y_i |\langle x_i, \theta^\star \rangle|,
\end{aligned}
$$

where we have used Proposition B.1 to obtain the bound $|\langle \theta_j, \theta^\star \rangle| = C(e^{-M/2} + \delta^{-1/4})$, due to the fact that $\theta_j$ is not the principal eigenvector. Thus, by the assumption that $\delta > CM^4$ made in the Theorem 3.2, we obtain that

$$
\sup_{\theta_j \in S} |R_1| \leq C(\sqrt{M} e^{-M/2} + \delta^{-1/8}).
$$

An analogous calculation gives

$$
\sup_{\theta_j \in S} |R_2| \leq C(\sqrt{M} e^{-M/2} + \delta^{-1/8}).
$$

Finally, combining the above bounds, we conclude that for $M$ large enough

$$
\sup_{\theta_j \in S} \left| \theta^{\star\top} \tilde{R}(\theta_j) \theta^\star \right| \leq C(\sqrt{M} e^{-M/2} + \delta^{-1/8}),
$$

proving the claim. □

As discussed above, a direct consequence of the bound in (B.27) is that

$$\theta^{\star\top}(A(\theta_j) - \lambda_{\theta_j} I + \theta^{\star\top}\tilde{R}(\theta_j)\theta^\star > 0,$$

with probability $1 - \exp(-cd^{1/5})$. In particular, this implies that the matrix $A(\theta_j) - \lambda_{\theta_j} I + \tilde{R}(\theta_j)$ has at least one positive eigenvalue. Consequently, for every $\theta_j \in S$,

$$\dim(E^u(\theta_j) \oplus E^c(\theta_j)) < d - 1.$$

It follows that the stable-center manifold $W_{loc}^{sc}(\theta_j)$ has codimension at least one on $\mathcal{S}^{d-1}$, and hence

$$\mu(W_{loc}^{sc}(\theta_j)) = 0,$$

for $\mu$ the uniform probability measure on the sphere. Next, since $\Phi_t$ is smooth for any $t$ and so is locally Lipschitz, we have that, for any set $T \subseteq \mathcal{S}^{d-1}$,

$$\mu(T) = 0 \quad \Longrightarrow \quad \mu\left(\Phi_t^{-1}(T)\right) = 0.$$

In particular, for $T = W_{loc}^{sc}(\theta_j)$, it holds that $\mu(\Phi_1^{-1}(W_{loc}^{sc}(\theta_j))) = \mu(W_{loc}^{sc}(\theta_j)) = 0$, uniformly over $\theta_j$, with probability $1 - O(\frac{1}{n})$ over the sampling of $[x_i]_{i=1}^n$. Finally, by (B.22), the set $C_S$ is contained in a countable union of sets that have measure 0. Hence,

$$\mu(C_s) = \mathbb{P}_{\theta_0}(\theta_\infty \text{ is not a principal eigenvector of } A(\theta_\infty)) = 0,$$

where the probability is taken with respect to the uniform sampling of $\theta_0 \in \mathcal{S}^{d-1}$. Therefore, with probability $1 - O(\frac{1}{n})$ over the sampling of the data $[x_i]_{i=1}^n$, and almost surely with respect to random initialization $\theta_0$, the limit point $\theta_\infty$ is the principal eigenvector of $A(\theta_\infty)$. This completes the proof of the proposition. □

### B.3. Concluding the argument

*Proof of Theorem 3.2.* By Proposition B.4, it holds that there exists a convergence point $\theta_\infty$ such that

$$\lim_{t \to \infty} \theta(t) = \theta_\infty,$$

which is a stationary point of $\widehat{\mathcal{L}}$. By using Theorem B.3, we obtain that, for any matrix $A(\theta)$, it holds that it has a simple principal eigenvalue and the spectral gap satisfies $\lambda_1(A(\theta)) - \lambda_2(A(\theta)) \geq 2(c_\sigma^1 - c_\sigma^2) - C(e^{-M/2} + \delta^{-1/5})$. As the assumptions of Proposition B.5 are satisfied, we get $\theta_\infty$ is a principal eigenvector of the matrix $A(\theta_\infty)$ with probability at least $1 - \exp(-cd^{1/5})$ over sampling of $x_i$, and almost surely over the sampling of $\theta_0$. Moreover, by (B.8) the corresponding top eigenvector achieves recovery, which implies that

$$|\langle \theta_\infty, \theta^\star \rangle| \geq 1 - C(e^{-M/2} + \delta^{-1/5}).$$

Finally, this allows us to conclude that

$$\lim_{t \to \infty} |\langle \theta(t), \theta^\star \rangle| \geq 1 - C(e^{-M/2} + \delta^{-1/5}),$$

with probability at least $1 - \exp(-cd^{1/5})$ □

## C. Relaxing the lower bound on $n/d$

Recall that Proposition B.1 gives that, with probability at least $1 - e^{-d}$, the matrix $A^\star$ concentrates around a certain deterministic matrix. As a consequence, the leading eigenspace of $A^\star$ is well-behaved, that is, the top eigenvector of $A^\star$ is almost aligned with the signal $\theta^\star$, the top-two eigenvalues of $A^\star$ are of constant order and there is a constant-order gap between them. More precisely, it holds that

$$|\langle v_1(A^\star), \theta^\star \rangle| \leq CM\sqrt{\frac{d}{n}}, \quad |\lambda_1(A^\star) - 6| + |\lambda_2(A^\star) - 2| \leq C\left(e^{-M/3} + M\sqrt{\frac{d}{n}}\right).$$

Thus, in order to get an approximation error of the top eigenvector alignment and the top two eigenvalues of order at most $(\frac{n}{d})^{1/4}$, used for the final bound in Theorem 3.2, the assumption $n \geq CM^4d$ is needed. This assumption can be relaxed to $n \geq CM^{2+\epsilon}d$ for any $\epsilon > 0$, by getting a sharper concentration rate on the top eigenvector alignment and top two eigenvalues. The trade-off is that the resulting concentration holds with probability at least $1 - 1/n$ instead of at least $1 - e^{-d}$, and we additionally require an upper bound on the ratio $n/d$ of order $e^{M/30}$.

Below, we present such a result that relies on (Mondelli & Montanari, 2018), and can be used in place of Proposition B.1 .

**Proposition C.1** (Spectral properties of the matrix $A^\star$). *Let us be in the setting of (1.1), i.e., let $\theta^\star \in \mathcal{S}^{d-1}$, $\{x_i\}_{i=1}^n \overset{\text{i.i.d.}}{\sim} \mathcal{N}(0, I_d)$, and $y_i = \sigma(\langle x_i, \theta^\star \rangle)$, for $\sigma$ defined in (3.6), with $M$ a large enough constant (independent of $n$ and $d$). Let $\delta = n/d$ and assume $\delta = \Delta(M)$, where $\Delta : \mathbb{R}_+ \to \mathbb{R}_+$ is any function such that*

$$\lim_{M \to \infty} \frac{\Delta(M)}{M^2} = \infty, \qquad \lim_{M \to \infty} \frac{\Delta(M)}{e^{M/30}} = 0.$$

*Let $D_n \in \mathbb{R}^{d \times d}$ be a matrix defined as*

$$D_n := \frac{1}{n} \sum_{i=1}^n y_i x_i x_i^\top.$$

*Then, for $n$ and $d$ large enough, it holds with probability at least $1 - \frac{1}{n}$ that*

$$|\langle v_1(D_n), \theta^\star \rangle| \leq 1 - C\delta^{-1},$$
$$|\lambda_1(D_n) - 3| \leq C\delta^{-1},$$
$$|\lambda_2(D_n) - 1| \leq C\delta^{-1/2},$$

*where $C$ is a numerical constants independent of $n, d, M, \delta$.*

Before turning to the proof of the result, we note that the arguments of Section B remain valid, if Proposition B.1 is replaced with Proposition C.1. This substitution amounts to replacing the constants $c_\sigma^1$ by 3, $c_\sigma^2$ by 1, and adjusting the probability statements accordingly.

*Proof.* Note that throughout the proof, to simplify exposition, we will rely on the big $O$ notation for $M$ and $\delta$. This is justified by both $M$ and $\delta$ being taken large enough (although still independent of $n$ and $d$).

As we rely on (Mondelli & Montanari, 2018, Lemma 2), we start by reintroducing some notation therein. First, since $x_i$ is isotropic Gaussian, without loss of generality we can assume that $\theta^\star = e_1$, $e_1$ being the first element of the canonical basis. Then, we consider random variables that follow the same distribution as the data from (1.1) in the direction of $e_1$, that is,

$$X \sim \mathcal{N}(0, 1), \quad Y \sim \sigma(X).$$

Note that the pre-processing function $\mathcal{T}$ introduced in (Mondelli & Montanari, 2018), is the identity in our setting. We therefore omit further explicit reference to it and work directly with $Y = \sigma(X)$. Let $\tau$ be the supremum of the support of $Y$, i.e.,

$$\tau = \inf\{y : \mathbb{P}(Y \leq y) = 1\}. \tag{C.1}$$

For $\lambda \in (\tau, \infty)$ and $\delta \in (0, \infty)$, define

$$\phi(\lambda) = \lambda \cdot \mathbb{E}\left\{\frac{Y \cdot X^2}{\lambda - Y}\right\}, \tag{C.2}$$

and

$$\psi_\delta(\lambda) = \lambda \left(\frac{1}{\delta} + \mathbb{E}\left\{\frac{Y}{\lambda - Y}\right\}\right). \tag{C.3}$$

Note that $\phi(\lambda)$ is a monotone non-increasing function and that $\psi_\delta(\lambda)$ is a convex function. Let $\bar{\lambda}_\delta$ be the point at which $\psi_\delta$ attains its minimum, i.e.,

$$\bar{\lambda}_\delta = \arg\min_{\lambda \geq \tau} \psi_\delta(\lambda). \tag{C.4}$$

For $\lambda \in (\tau, \infty)$, define also

$$\zeta_\delta(\lambda) = \psi_\delta(\max(\lambda, \bar{\lambda}_\delta)). \tag{C.5}$$

Directly from the properties of $\varphi$, one gets

$$\tau = \sup_{z \in \mathbb{R}} \sigma(z) = \bar{M} = \int_0^{2M} \varphi(u)du,$$

from which it follows that $M < \tau < 2M$. First, we prove that the assumptions of (Mondelli & Montanari, 2018, Lemma 2) are satisfied, that is,

$$\lim_{\lambda \to \tau^+} \mathbb{E}\left[\frac{Y}{(\lambda - Y)^2}\right] = \lim_{\lambda \to \tau^+} \mathbb{E}\left[\frac{YX^2}{\lambda - Y}\right] = +\infty.$$

Since $\sigma(x) \geq 0$, for $\lambda > \tau$, it holds that

$$\begin{aligned}
\mathbb{E}\left[\frac{YX^2}{\lambda - Y}\right] = \int_{\mathbb{R}} \frac{\sigma(x)x^2}{\lambda - \sigma(x)} \frac{e^{-x^2/2}}{\sqrt{2\pi}} dx &\geq \int_{\sqrt{2M}}^{\infty} \frac{\sigma(x)x^2}{\lambda - \sigma(x)} \frac{e^{-x^2/2}}{\sqrt{2\pi}} dx \\
&= \frac{\tau}{\lambda - \tau} \int_{\sqrt{2M}}^{\infty} x^2 \frac{e^{-x^2/2}}{\sqrt{2\pi}} dx \\
&= \frac{\tau}{\lambda - \tau} \cdot c_+,
\end{aligned}$$

where in this case $c_+ = \int_{\sqrt{2M}}^{\infty} x^2 \frac{e^{-x^2/2}}{\sqrt{2\pi}} dx$ is a positive constant. From this directly follows

$$\lim_{\lambda \to \tau^+} \mathbb{E}\left[\frac{YX^2}{\lambda - Y}\right] \geq \lim_{\lambda \to \tau^+} \frac{\tau}{\lambda - \tau} \cdot c_+ = +\infty,$$

as $\tau > M > 0$. Similarly, one gets for some $c_+ > 0$

$$\mathbb{E}\left[\frac{Y}{(\lambda - Y)^2}\right] \geq \frac{\tau}{(\lambda - \tau)^2} \cdot c_+,$$

and hence,

$$\lim_{\lambda \to \tau^+} \mathbb{E}\left[\frac{Y}{(\lambda - Y)^2}\right] \geq \lim_{\lambda \to \tau^+} \frac{\tau}{(\lambda - \tau)^2} \cdot c_+ = +\infty.$$

**Calculating $\lambda_\delta^*$.** Let $\lambda_\delta^*$ be defined as in (Mondelli & Montanari, 2018, Lemma 2), i.e., the unique solution $\lambda > \tau$ to

$$\zeta_\delta(\lambda) = \phi(\lambda).$$

We will then find it by looking at the solution to the equation

$$\phi(\lambda) = \psi_\delta(\lambda), \tag{C.6}$$

for $\lambda \geq \bar{\lambda}_\delta \geq \tau$. This is the same as finding the largest $\lambda$ for which it holds

$$\mathbb{E}\left\{\frac{Y \cdot (X^2 - 1)}{\lambda - Y}\right\} = \frac{1}{\delta}.$$

Let us denote the LHS of the previous equation as $f(\lambda)$. We have that

$$\begin{aligned}
f(\lambda) &= \int_{\mathbb{R}} \frac{\sigma(x)(x^2 - 1)}{\lambda - \sigma(x)} \frac{e^{-x^2/2}}{\sqrt{2\pi}} dx \\
&= \sqrt{\frac{2}{\pi}} \int_0^{\sqrt{M}} \frac{x^2(x^2 - 1)}{\lambda - x^2} e^{-x^2/2} dx + \sqrt{\frac{2}{\pi}} \int_{\sqrt{M}}^{\infty} \frac{\sigma(x)(x^2 - 1)}{\lambda - \sigma(x)} e^{-x^2/2} dx.
\end{aligned}$$

We restrict the domain of search of $\lambda$ by assuming that $e^{M/8} > \lambda > 3M$, and we will verify that there is a solution to (C.6) in this interval later. Then, for $x \geq \sqrt{M}$, the definition of $\sigma$ implies $M \leq \sigma(x) \leq 2M$, hence

$$\frac{M}{\lambda - M} \leq \frac{\sigma(x)}{\lambda - \sigma(x)} \leq \frac{2M}{\lambda - 2M}. \tag{C.7}$$

Using this bound, we obtain

$$\frac{M}{\lambda - M} \cdot \sqrt{\frac{2}{\pi}} \int_{\sqrt{M}}^{\infty} (x^2 - 1)e^{-x^2/2}dx \leq \sqrt{\frac{2}{\pi}} \cdot \int_{\sqrt{M}}^{\infty} \frac{\sigma(x)(x^2-1)}{\lambda - \sigma(x)}e^{-x^2/2}dx \leq \frac{2M}{\lambda - 2M} \cdot \sqrt{\frac{2}{\pi}} \int_{\sqrt{M}}^{\infty}(x^2-1)e^{-x^2/2}dx.$$

Consequently, we may write

$$f(\lambda) = \sqrt{\frac{2}{\pi}} \int_0^{\sqrt{M}} \frac{x^2(x^2-1)}{\lambda - x^2}e^{-x^2/2}dx + r(\lambda, M) \cdot \sqrt{\frac{2}{\pi}} \int_{\sqrt{M}}^{\infty}(x^2-1)e^{-x^2/2}dx,$$

where the coefficient $r(\lambda, M)$ satisfies $\frac{M}{\lambda - M} \leq r(\lambda, M) \leq \frac{2M}{\lambda - 2M}$. Next, note that

$$\frac{x^2(x^2-1)}{\lambda - x^2} = (1 - \lambda - x^2) + \frac{\lambda(\lambda - 1)}{\lambda - x^2},$$

so we can use this to simplify integrals showing up in $f(\lambda)$. Namely, we use the additivity of the integral to write

$$f(\lambda) = \sqrt{\frac{2}{\pi}} [(1 - \lambda)B_0(M) - B_2(M) + \lambda(\lambda - 1)H(\lambda, M)] + r(\lambda, M) \cdot \sqrt{\frac{2}{\pi}} [T_2(M) - T_0(M)],$$

where the "easy" integrals are

$$B_0(M) = \int_0^{\sqrt{M}} e^{-x^2/2}dx = \sqrt{\frac{\pi}{2}} \,\text{erf}(\sqrt{M/2}),$$

$$B_2(M) = \int_0^{\sqrt{M}} x^2 e^{-x^2/2}dx = \sqrt{\frac{\pi}{2}} \,\text{erf}(\sqrt{M/2}) - \sqrt{M}e^{-M/2},$$

$$T_0(M) = \int_{\sqrt{M}}^{\infty} e^{-x^2/2}dx = \sqrt{\frac{\pi}{2}} \,\text{erfc}(\sqrt{M/2}),$$

$$T_2(M) = \int_{\sqrt{M}}^{\infty} x^2 e^{-x^2/2}dx = \sqrt{M}e^{-M/2} + \sqrt{\frac{\pi}{2}} \,\text{erfc}(\sqrt{M/2}),$$

and the "hard" one is

$$H(\lambda, M) = \int_0^{\sqrt{M}} \frac{e^{-x^2/2}}{\lambda - x^2}dx.$$

As we assume that $M$ is large enough, we can use the asymptotic notation $\Omega(1), O(1), \Theta(1)$ (intended for large $M$). Combining the above calculations, bounds, and the estimate $\text{erfc}(\sqrt{M/2}) = \Theta\left(e^{-M/2}/\sqrt{M}\right)$, we get

$$f(\lambda) = -\lambda + \sqrt{\frac{2}{\pi}}\lambda(\lambda - 1)H(\lambda, M) + r(\lambda, M)\Theta\left(\sqrt{M}e^{-M/2}\right) + \Theta\left(e^{-M/2}/\sqrt{M}\right)$$

$$= -\lambda + \sqrt{\frac{2}{\pi}}\lambda(\lambda - 1)H(\lambda, M) + O(e^{-M/4}) \tag{C.8}$$

Let us turn our attention now to $H(\lambda, M)$. Since $|x| \leq \sqrt{M} < \sqrt{\lambda}$, we can uniformly expand

$$\frac{1}{\lambda - x^2} = \frac{1}{\lambda} \sum_{k=0}^{\infty} \left(\frac{x^2}{\lambda}\right)^k.$$

Integrating term-wise gives

$$H(\lambda, M) = \sum_{k=0}^{\infty} \frac{1}{\lambda^{k+1}} \int_0^{\sqrt{M}} x^{2k} e^{-x^2/2}\, dx.$$

Note that the function $x \to x^{2k}e^{-x^2/2}$ is increasing on $[0, \sqrt{2k}]$ and decreasing on $[\sqrt{2k}, \infty]$. Thus, if $k \geq M/2$, it holds that

$$\int_0^{\sqrt{M}} x^{2k}e^{-x^2/2}\, dx < \sqrt{M}e^{-M/2}M^k.$$

From this, it follows that

$$\sum_{k=M/2}^{\infty} \frac{1}{\lambda^{k+1}} \int_0^{\sqrt{M}} x^{2k} e^{-x^2/2} \, dx = O(e^{-M/2}).$$

We treat the case $k < M/2$ by completing the integral to $\infty$, as it has a closed form that we can more easily bound. Namely, it holds

$$\int_0^{\sqrt{M}} x^{2k} e^{-x^2/2} \, dx \leq \int_0^{\infty} x^{2k} e^{-x^2/2} \, dx,$$

and

$$\int_0^{\infty} x^{2k} e^{-x^2/2} \, dx = 2^{k-\frac{1}{2}} \Gamma\left(k + \tfrac{1}{2}\right) = (2k-1)!! \sqrt{\frac{\pi}{2}}. \tag{C.9}$$

Next, using Stirling's upper bound, we have

$$(2k-1)!! \sqrt{\frac{\pi}{2}} \leq 10 \left(\frac{2k}{e}\right)^k.$$

Therefore,

$$\sum_{k=4}^{M/2-1} \frac{1}{\lambda^{k+1}} \int_0^{\sqrt{M}} x^{2k} e^{-x^2/2} \, dx \leq c \sum_{k=4}^{M/2-1} \frac{1}{\lambda^{k+1}} \left(\frac{2k}{e}\right)^k = c \sum_{k=4}^{M/2-1} \frac{1}{\lambda} \left(\frac{2k}{e\lambda}\right)^k = O\left(\frac{1}{\lambda^4}\right).$$

Lastly, via integration by parts, one can get that

$$\int_{\sqrt{M}}^{\infty} x^{2k} e^{-x^2/2} \, dx \leq c M^{k-1/2} e^{-M/2},$$

from which it follows that, for $k \leq 3$,

$$\frac{1}{\lambda^{k+1}} \int_0^{\sqrt{M}} x^{2k} e^{-x^2/2} \, dx = \frac{1}{\lambda^{k+1}} (2k-1)!! \sqrt{\frac{\pi}{2}} - O(e^{-M/2}).$$

Thus, by putting all of this together and calculating $(2k-1)!!$ explicitly for $k \leq 3$, we get

$$H(\lambda, M) = \sqrt{\frac{\pi}{2}} \left(\frac{1}{\lambda} + \frac{1}{\lambda^2} + \frac{3}{\lambda^3}\right) + O\left(\frac{1}{\lambda^4}\right) + O\left(e^{-M/2}\right).$$

Plugging this back into the expression (C.8) for $f(\lambda)$, we have that

$$f(\lambda) = -\lambda + \lambda(\lambda - 1)\left(\frac{1}{\lambda} + \frac{1}{\lambda^2} + \frac{3}{\lambda^3}\right) + O\left(\frac{1}{\lambda^2}\right) + O(e^{-M/4})$$

$$= \frac{2}{\lambda} + O\left(\frac{1}{\lambda^2}\right) + O(e^{-M/4}).$$

As we restrict our domain of search to $\lambda < e^{M/8}$, we get that

$$f(\lambda) = \frac{2}{\lambda} + O\left(\frac{1}{\lambda^2}\right).$$

Solving $f(\lambda) = \frac{1}{\delta}$ we get that there is a unique solution for $\lambda \in (3M, e^{M/8})$, which is

$$\lambda_\delta^* = 2\delta + O_\delta(1). \tag{C.10}$$

Note that this solution is in the interval $(3M, e^{M/8})$, as assumed during the argument above.

**Calculating $\bar{\lambda}_\delta$.** Let us turn our attention to $\bar{\lambda}_\delta$. Since $\psi_\delta(\lambda)$ is a convex function, $\bar{\lambda}_\delta$ is the unique solution to

$$\psi'_\delta(\lambda) = 0.$$

Writing this condition out, we have that it is equivalent to solving

$$\int_\mathbb{R} \frac{(\sigma(x))^2}{(\lambda - \sigma(x))^2} \frac{e^{-x^2/2}}{\sqrt{2\pi}} dx = \frac{1}{\delta}. \tag{C.11}$$

The LHS can be further expressed as

$$\int_\mathbb{R} \frac{(\sigma(x))^2}{(\lambda - \sigma(x))^2} \frac{e^{-x^2/2}}{\sqrt{2\pi}} dx = \sqrt{\frac{2}{\pi}} \cdot \int_0^{\sqrt{M}} \frac{x^4}{(\lambda - x^2)^2} e^{-x^2/2} dx + \sqrt{\frac{2}{\pi}} \cdot \int_{\sqrt{M}}^\infty \frac{\sigma(x)^2}{(\lambda - \sigma(x))^2} e^{-x^2/2} dx.$$

We assume $e^{M/10} > \lambda > 3M$ and we will verify later that the desired solution lies in this interval. Using the bound (C.7), the tail integral satisfies

$$\sqrt{\frac{2}{\pi}} \int_{\sqrt{M}}^\infty \frac{\sigma(x)^2}{(\lambda - \sigma(x))^2} e^{-x^2/2} dx = \tilde{r}(\lambda, M) \operatorname{erfc}\left(\sqrt{\frac{M}{2}}\right),$$

where the coefficient $\tilde{r}(\lambda, M)$ satisfies $\frac{M^2}{(\lambda - M)^2} \leq \tilde{r}(\lambda, M) \leq \frac{4M^2}{(\lambda - 2M)^2}$. Turning to the other integral, as $|x| \leq \sqrt{M} < \sqrt{\lambda}$, we can expand

$$\frac{1}{(\lambda - x^2)^2} = \frac{1}{\lambda^2} \frac{1}{(1 - x^2/\lambda)^2} = \frac{1}{\lambda^2} \sum_{k=0}^\infty (k+1) \left(\frac{x^2}{\lambda}\right)^k.$$

Integrating term wise yields

$$\sqrt{\frac{2}{\pi}} \int_0^{\sqrt{M}} \frac{x^4}{(\lambda - x^2)^2} e^{-x^2/2} dx = \sqrt{\frac{2}{\pi}} \sum_{k=0}^\infty \frac{k+1}{\lambda^{k+2}} \int_0^{\sqrt{M}} x^{2k+4} e^{-x^2/2} dx.$$

As before, the function $x \to x^{2k+4} e^{-x^2/2}$ is increasing on $[0, \sqrt{2k+4}]$, so if $k \geq M/2 - 2$, then it holds

$$\int_0^{\sqrt{M}} x^{2k+4} e^{-x^2/2} dx < \sqrt{M} e^{-M/2} M^{k+2}.$$

We assume $\lambda > 3M$ and we will verify later that the desired solution lies in this interval. Then,

$$\sum_{k=M/2-2}^\infty \frac{k+1}{\lambda^{k+2}} \int_0^{\sqrt{M}} x^{2k+4} e^{-x^2/2} dx = O(e^{-M/2}).$$

For $k < M/2 - 2$, we write

$$\sqrt{\frac{2}{\pi}} \int_0^{\sqrt{M}} x^{2k+4} e^{-x^2/2} dx \leq \sqrt{\frac{2}{\pi}} \int_0^\infty x^{2k+4} e^{-x^2/2} dx = (2k+3)!!$$

Then, by using the bound from Sterling formula, for $k \geq 2$ we have

$$\sum_{k=2}^{M/2-3} \frac{k+1}{\lambda^{k+2}} \int_0^{\sqrt{M}} x^{2k+4} e^{-x^2/2} dx = O\left(\frac{1}{\lambda^4}\right).$$

Finally, for large $M$ and $k < 2$, one can get

$$\frac{1}{\lambda^{k+2}} \int_{\sqrt{M}}^\infty x^{2k+4} e^{-x^2/2} dx = O(M e^{-M/2}),$$

from which it follows

$$\frac{1}{\lambda^{k+2}}\sqrt{\frac{2}{\pi}}\int_0^{\sqrt{M}}x^{2k+4}e^{-x^2/2}\,dx = \frac{1}{\lambda^{k+2}}(2k+3)!! - O(Me^{-M/2}).$$

Calculating explicitly the double factorial for $k < 2$ and combing the bounds, one gets

$$\sqrt{\frac{2}{\pi}}\int_0^{\sqrt{M}}\frac{x^4}{(\lambda - x^2)^2}e^{-x^2/2}dx = \left(\frac{3}{\lambda^2} + \frac{30}{\lambda^3}\right) + O\left(\frac{1}{\lambda^4}\right) + O\left(Me^{-M/2}\right).$$

If we restrict our domain of search to $\lambda \le e^{M/10}$, we get that $\frac{1}{\lambda^4} \ge Me^{-M/2}$, so

$$\sqrt{\frac{2}{\pi}}\int_0^{\sqrt{M}}\frac{x^4}{(\lambda - x^2)^2}e^{-x^2/2}dx = \left(\frac{3}{\lambda^2} + \frac{30}{\lambda^3}\right) + O\left(\frac{1}{\lambda^4}\right).$$

Then, plugging all of this into (C.11) gives

$$\frac{3}{\lambda^2} + \frac{30}{\lambda^3} + O\left(\frac{1}{\lambda^4}\right) + \tilde{r}(\lambda, M)\operatorname{erfc}\left(\sqrt{\frac{M}{2}}\right) = \frac{1}{\delta}. \tag{C.12}$$

Recall that $\operatorname{erfc}(\sqrt{M/2}) = \Theta\left(e^{-M/2}/\sqrt{M}\right)$ and that $e^{-M/10} > \lambda > 3M$, hence

$$\tilde{r}(\lambda, M)\operatorname{erfc}\left(\sqrt{\frac{M}{2}}\right) = O\left(\frac{1}{\lambda^3}\right).$$

Thus, the equation we want to solve is

$$\frac{3}{\lambda^2} + O\left(\frac{1}{\lambda^3}\right) = \frac{1}{\delta},$$

which gives

$$\bar{\lambda}_\delta = \sqrt{3}\delta^{1/2} + O_\delta(1). \tag{C.13}$$

Note that this solution is in the interval $(3M, e^{M/10})$, as assumed during the argument above.

**Calculating overlap.** From the formulas for $\lambda_\delta^*$ and $\bar{\lambda}_\delta$, i.e. (C.13) and (C.10), we can see that

$$\lambda_\delta^* > \bar{\lambda}_\delta.$$

Therefore, directly from (Mondelli & Montanari, 2018, Lemma 2) we get that there is weak recovery. Let us further explicitly calculate the overlap, which according to the previous Lemma is

$$|\langle v_1(D_n), \theta^\star\rangle| \xrightarrow{\text{a.s.}} \frac{\psi_\delta'(\lambda_\delta^*)}{\psi_\delta'(\lambda_\delta^*) - \phi'(\lambda_\delta^*)}. \tag{C.14}$$

Plugging in, we have

$$\psi_\delta'(\lambda_\delta^*) = \frac{1}{\delta} - \left(\sqrt{\frac{2}{\pi}} \cdot \int_0^{\sqrt{M}}\frac{x^4}{(\lambda_\delta^* - x^2)^2}e^{-x^2/2}dx + \sqrt{\frac{2}{\pi}} \cdot \int_{\sqrt{M}}^\infty\frac{\sigma(x)^2}{(\lambda_\delta^* - \sigma(x))^2}e^{-x^2/2}dx\right).$$

Recall that from (C.10) it holds $\lambda_\delta^* = 2\delta + O_\delta(1) \gg 2M^2$. Thus, we can do an expansion for $x \in \left[0, \sqrt{M}\right]$ as

$$\frac{x^4}{(\lambda_\delta^* - x^2)^2} = \frac{1}{\lambda_\delta^{*2}}\frac{x^4}{(1 - x^2/\lambda_\delta^*)^2} = \frac{x^4}{\lambda_\delta^{*2}}\Theta_{\lambda_\delta^*}(1).$$

Plugging this in, using calculations of Gaussian moments in (C.9), and the bound in (C.7) for $x \geq \sqrt{M}$, yields

$$
\begin{aligned}
\psi_\delta'(\lambda_\delta^*) &= \frac{1}{\delta} - \left( \frac{1}{\lambda_\delta^{*2}} \cdot \sqrt{\frac{2}{\pi}} \int_0^{\sqrt{M}} \frac{x^4}{(1 - x^2/\lambda_\delta^*)^2} e^{-x^2/2} dx + \sqrt{\frac{2}{\pi}} \cdot \int_{\sqrt{M}}^\infty \frac{\sigma(x)^2}{(\lambda_\delta^* - \sigma(x))^2} e^{-x^2/2} dx \right) \\
&= \frac{1}{\delta} - \left( \frac{1}{\lambda_\delta^{*2}} \cdot O_{\lambda_\delta^*}(1) + O(e^{-M/2}) \right) \\
&= \frac{1}{\delta} - \left( O(\delta^{-2}) + O(e^{-M/2}) \right) \\
&= \frac{1}{\delta} - O(\delta^{-2}).
\end{aligned}
$$

Similarly,

$$
\begin{aligned}
-\phi'(\lambda_\delta^*) &= \sqrt{\frac{2}{\pi}} \cdot \int_0^{\sqrt{M}} \frac{x^6}{(\lambda_\delta^* - x^2)^2} e^{-x^2/2} dx + \sqrt{\frac{2}{\pi}} \cdot \int_{\sqrt{M}}^\infty \frac{\sigma(x)^2 x^2}{(\lambda_\delta^* - \sigma(x))^2} e^{-x^2/2} dx \\
&= \left( \frac{1}{\lambda_\delta^{*2}} \cdot \sqrt{\frac{2}{\pi}} \int_0^{\sqrt{M}} \frac{x^6}{(1 - x^2/\lambda_\delta^*)^2} e^{-x^2/2} dx + \sqrt{\frac{2}{\pi}} \cdot \int_{\sqrt{M}}^\infty \frac{\sigma(x)^2 x^2}{(\lambda_\delta^* - \sigma(x))^2} e^{-x^2/2} dx \right) \\
&= O(\delta^{-2}) + O(e^{-M/2}) \\
&= O(\delta^{-2}).
\end{aligned}
$$

Therefore, it holds that

$$
\begin{aligned}
\frac{\psi_\delta'(\lambda_\delta^*)}{\psi_\delta'(\lambda_\delta^*) - \phi'(\lambda_\delta^*)} &= \frac{\delta^{-1} - O(\delta^{-2})}{\delta^{-1} + O(\delta^{-2})} \\
&= 1 - O\left(\delta^{-1}\right).
\end{aligned}
$$

Plugging this into (C.14) proves that

$$
|\langle v_1(D_n), \theta^\star \rangle| \xrightarrow{\text{a.s.}} 1 - O(\delta^{-1}).
$$

By Borel-Cantelli lemma we get that for $n$ large enough

$$
|\langle v_1(D_n), \theta^\star \rangle| \leq 1 - C\delta^{-1},
$$

with probability at least $1 - \frac{1}{n}$, and some constant $C$.

**Calculating top eigenvalues.** Let us turn our attention to the top eigenvalues of $D_n$. Again, by (Mondelli & Montanari, 2018, Lemma 2), we have that

$$
\begin{aligned}
\lambda_1(D_n) &\xrightarrow{\text{a.s.}} \zeta_\delta(\lambda_\delta^*) = \psi_\delta(\lambda_\delta^*), \\
\lambda_2(D_n) &\xrightarrow{\text{a.s.}} \zeta_\delta(\bar{\lambda}_\delta) = \psi_\delta(\bar{\lambda}_\delta).
\end{aligned} \tag{C.15}
$$

As before, recall it holds $\lambda_\delta^* = 2\delta + O_\delta(1) \gg 2M^2$. Thus, we can do an expansion argument similar to the one done following (C.8), which yields

$$
\begin{aligned}
\psi_\delta(\lambda_\delta^*) &= \lambda_\delta^* \left( \frac{1}{\delta} + \sqrt{\frac{2}{\pi}} \cdot \int_0^{\sqrt{M}} \frac{x^2}{\lambda_\delta^* - x^2} e^{-x^2/2} dx + \sqrt{\frac{2}{\pi}} \cdot \int_{\sqrt{M}}^\infty \frac{\sigma(x)}{\lambda_\delta^* - \sigma(x)} e^{-x^2/2} dx \right) \\
&= \lambda_\delta^* \left( \frac{1}{\delta} + \frac{1}{\lambda_\delta^*} \cdot \sqrt{\frac{2}{\pi}} \int_0^{\sqrt{M}} \frac{x^2}{1 - x^2/\lambda_\delta^*} e^{-x^2/2} dx + \sqrt{\frac{2}{\pi}} \cdot \int_{\sqrt{M}}^\infty \frac{\sigma(x)}{\lambda_\delta^* - \sigma(x)} e^{-x^2/2} dx \right) \\
&= \lambda_\delta^* \left( \frac{1}{\delta} + \frac{1}{\lambda_\delta^*} + O\left( \frac{1}{\lambda_\delta^{*2}} \right) + O(e^{-M/2}) \right) \\
&= 2\delta \left( \frac{1}{\delta} + \frac{1}{2\delta} + O(\delta^{-2}) + O(e^{-M/2}) \right) + O(\delta^{-1}) \\
&= 3 + O(\delta^{-1}),
\end{aligned}
$$

where we have again used (C.7) to get a bound on the tail integral. Similarly, recalling $\bar{\lambda}_\delta = \sqrt{3}\delta^{1/2} + O_\delta(1)$, yields

$$
\begin{aligned}
\psi_\delta(\bar{\lambda}_\delta) &= \bar{\lambda}_\delta \left( \frac{1}{\delta} + \sqrt{\frac{2}{\pi}} \cdot \int_0^{\sqrt{M}} \frac{x^2}{\bar{\lambda}_\delta - x^2} e^{-x^2/2} dx + \sqrt{\frac{2}{\pi}} \cdot \int_{\sqrt{M}}^\infty \frac{\sigma(x)}{\bar{\lambda}_\delta - \sigma(x)} e^{-x^2/2} dx \right) \\
&= \bar{\lambda}_\delta \left( \frac{1}{\delta} + \frac{1}{\bar{\lambda}_\delta} + O\left( \frac{1}{\bar{\lambda}_\delta^2} \right) + O\left( e^{-M/2} \right) \right) \\
&= \sqrt{3}\delta^{1/2} \left( \frac{1}{\delta} + \frac{1}{\sqrt{3}\delta^{1/2}} + O(\delta^{-1}) \right) + O(\delta^{-1/2}) \\
&= 1 + O(\delta^{-1/2}).
\end{aligned}
$$

Plugging this into (C.15) yields

$$
\begin{aligned}
\lambda_1(D_n) &\xrightarrow{\text{a.s.}} 3 + O(\delta^{-1}), \\
\lambda_2(D_n) &\xrightarrow{\text{a.s.}} 1 + O(\delta^{-1/2}).
\end{aligned}
$$

By Borel-Cantelli lemma we get that for $n$ large enough

$$
|\lambda_1(D_n) - 3| \le C\delta^{-1}, \quad |\lambda_2(D_n)| \le 1 + C\delta^{-1/2}.
$$

with probability at least $1 - \frac{1}{n}$, and some constant $C$. $\qquad\square$

# D. Proof of Theorem 4.1

## D.1. Angle reduction during the first phase

In the first phase of gradient descent, we have that $\theta_t$ gets high angular overlap with $\theta^\star$. We formalize the claim in the following main proposition of this subsection.

**Proposition D.1.** *Consider the truncated quadratic activation in (4.4), with large enough $M$. Let $\theta_t$ be obtained from the gradient descent iteration in (4.2), with learning rate $\eta < 1/100$. Assume that the initialization is sampled uniformly from the sphere of radius $r_0 = d^{-15}$, that is, $\theta_0 \sim \mathrm{Unif}(r_0 \mathcal{S}^{d-1})$. Denote by $\phi_t := \angle(\theta_t, \theta^\star) = \arccos\left( \frac{|\langle \theta_t, \theta^\star \rangle|}{\|\theta_t\|_2 \|\theta^\star\|_2} \right)$. Let $n, d$ be large enough, and fix $\delta = n/d$ such that $\delta \ge CM^2$. Then, with probability at least $1 - \frac{1}{d^2}$,*

$$
\phi_{t^*_\angle} \le C(e^{-M/2} + M\delta^{-1/2}), \tag{D.1}
$$

*where $t^*_\angle = \frac{3\log d}{\log(1+1.99\eta)}$ and $C, c > 0$ are universal constants (independent of $n, d, M, \delta$).*

In this first phase of gradient descent, the norm will be small enough and, therefore, $\langle x_i, \theta_t \rangle^2$ stays below the threshold $M$ across all iterates. We start by showing that this indeed is the case for any $\theta_t$ whose norm is bounded by $r(d) := \sqrt{M/2d}$.

**Lemma D.2.** *For $i \in [n]$, let $x_i \overset{\text{i.i.d.}}{\sim} \mathcal{N}(0, I_d)$, where $n/d = \delta$ is a constant (independent of $n, d$). Furthermore, we denote by $B_{r(d)}$ the Euclidean ball around $0$ of radius $r(d)$. Then, for large enough $d$ and $n$, it holds*

$$
\sup_{\theta \in B_{r(d)}} \sup_{i \in [n]} \langle x_i, \theta \rangle^2 \le M,
$$

*with probability at least $1 - \exp(-n^{1/2})$.*

*Proof.* Note that the index sets $[n]$ and $B_{r(d)}$ are independent, so we can swap the two supremum operators that show up in the expression of lemma. Furthermore, we have

$$
\sup_{\theta \in B_{r(d)}} \langle x_i, \theta \rangle^2 = r(d)^2 \|x_i\|^2,
$$

which yields

$$
\sup_{\theta \in B_{r(d)}} \sup_{i \in [n]} \langle x_i, \theta \rangle^2 = r(d)^2 \sup_{i \in [n]} \|x_i\|^2. \tag{D.2}
$$

Since each $x_i \sim \mathcal{N}(0, I_d)$, we have $\|x_i\|^2 \sim \chi_d^2$. Then, from the Laurent-Massart bound (Laurent & Massart, 2000, Lemma 1), it follows that

$$\mathbb{P}\left(\|x_i\|^2 \geq d + 2\sqrt{dw} + 2w\right) \leq e^{-w}.$$

Applying a union bound gives

$$\mathbb{P}\left(\sup_{i \in [n]} \|x_i\|^2 \geq d + 2\sqrt{dw} + 2w\right) \leq ne^{-w}.$$

By setting $w = n^{2/3}$, and for large enough $d$ and $n$, we get that

$$\sup_{i \in [n]} \|x_i\|^2 < 2d, \tag{D.3}$$

with probability at least $1 - \exp(-n^{1/2})$. Plugging this back into (D.2) yields

$$\sup_{\theta \in B_{r(d)}} \sup_{i \in [n]} \langle x_i, \theta \rangle^2 < r(d)^2 \, 2d,$$

with probability at least $1 - \exp(-n^{1/2})$. Taking $r(d) = \sqrt{\frac{M}{2d}}$ finishes the proof. $\qquad\square$

By Lemma D.2, for $\theta \in B_{r(d)}$ it holds $\sigma(\langle x_i, \theta \rangle) = \langle x_i, \theta \rangle^2$, so the expression for the empirical gradient in (4.3) simplifies. Namely, the empirical gradient at arbitrary $\theta \in \mathbb{R}^d$ can be expressed as

$$\widehat{\mathcal{G}}(\theta) = -A^\star \theta + \hat{R}(\theta), \tag{D.4}$$

where

$$A^\star = \frac{2}{n} \sum_{i=1}^n y_i x_i x_i^\top = \frac{2}{n} \sum_{i=1}^n \sigma(\langle x_i, \theta^\star \rangle) x_i x_i^\top, \qquad \hat{R}(\theta) = \frac{2}{n} \sum_{i=1}^n \langle x_i, \theta \rangle^3 x_i.$$

Then, for all $\theta_t \in B_{r(d)}$, the gradient update from (4.2) can be written as

$$\theta_{t+1} = \theta_t - \eta \nabla \widehat{\mathcal{G}}(\theta_t) = (I + \eta A^\star)\theta_t - \eta \hat{R}(\theta_t). \tag{D.5}$$

Let $\tilde{\theta}_t := (I + \eta A^\star)^t \theta_0$ be the power method iterate, and let $\xi_t := \theta_t - \tilde{\theta}_t$ be the error vector. Throughout the first phase of iterations, the error term $\xi_t$ will be bounded in terms of $\tilde{\theta}_t$. It will be convenient for the subsequent proof to explicitly identify the time until which this relationship holds. To this end, define

$$t^*_{r(d)} = \max\left\{\bar{t} \in \mathbb{N} \,:\, \forall t \leq \bar{t}, \quad \theta_{t+1} \in B_{r(d)} \text{ and } \|\xi_t\| < \|\tilde{\theta}_t\|\right\},$$

as the time step until which this simplification in (D.4) holds, and the error term $\xi_t$ is bounded in norm by $\tilde{\theta}_t$. We will relate later $t^*_{\not\perp}$ from Proposition D.1 to $t^*_{r(d)}$. Working towards this, we first state one auxiliary lemma that controls how much impact can $\hat{R}(\theta_t)$ have on the evolution of $\theta_t$.

**Lemma D.3.** *Uniformly for all $\theta \in \mathbb{R}^d$, with probability at least $1 - \exp(-n^{1/2})$, it holds*

$$\|\hat{R}(\theta)\| \leq 8 \, d^2 \|\theta\|^3$$

*Proof.* Writing out the definition of $\hat{R}(\theta)$ and using the triangle inequality and Cauchy-Schwarz, we get

$$\|\hat{R}(\theta)\| \leq \frac{2}{n} \sum_{i=1}^n \|\langle x_i, \theta \rangle^3 x_i\| \leq 2\|\theta\|^3 \frac{1}{n} \sum_{i=1}^n \|x_i\|^4. \tag{D.6}$$

From (D.3), we readily have that, with probability at least $1 - \exp(-n^{1/2})$,

$$\sup_{i \in [n]} \|x_i\|^4 < 4d^2,$$

which implies that

$$\frac{1}{n} \sum_{i=1}^n \|x_i\|^4 \leq \sup_{i \in [n]} \|x_i\|^4 < 4d^2.$$

Plugging this back into (D.6) concludes the proof. $\qquad\square$

Parts of the analysis, notably Lemma D.4, Lemma D.5 and Lemma D.6, follow the arguments in (Stöger & Soltanolkotabi, 2021, Section 8). The next lemma gives a bound on the approximation error term $\xi_t$, which will be useful to relate $\theta_t$ with $\tilde{\theta}_t$. Note that we will simplify further references to the leading eigenvalues and eigenvector of $A^\star$ by writing $\lambda_1 := \lambda_1(A^\star)$, $\lambda_2 := \lambda_2(A^\star)$ and $v_1 := v_1(A^\star)$, as these quantities will appear frequently in this part.

**Lemma D.4.** *Assume that, for all $\theta$, $\|\hat{R}(\theta)\| \leq C_{\hat{R}}\|\theta\|^3$ for some constant $C_{\hat{R}}$, and let $\lambda_1$ be the largest eigenvalue of $A^\star$. Then, for all $t \leq t^*_{r(d)} + 1$, it holds that*

$$\|\xi_t\| \leq \frac{4C_{\hat{R}}r_0^3}{\lambda_1}(1 + \eta\lambda_1)^{3t}.$$

*Proof.* Since $t \leq t^*_{r(d)} + 1$ implies that $\theta_t \in B_{r(d)}$, so identity $\xi_t = \theta_t - \tilde{\theta}_t$ holds. We derive a recursion for $\xi_t$

$$\begin{aligned}
\xi_t &= \theta_t - \tilde{\theta}_t \\
&= \left[(I + \eta A^\star)\theta_{t-1} - \eta\hat{R}(\theta_{t-1})\right] - \left[(I + \eta A^\star)\tilde{\theta}_{t-1}\right] \\
&= (I + \eta A^\star)(\theta_{t-1} - \tilde{\theta}_{t-1}) - \eta\hat{R}(\theta_{t-1}) = (I + \eta A^\star)\xi_t - \eta\hat{R}(\theta_{t-1}).
\end{aligned}$$

Unrolling this gives $\xi_t = -\sum_{i=1}^{t}(I + \eta A^\star)^{t-i}\eta\hat{R}(\theta_{i-1})$. Taking norms, we have

$$\|\xi_t\| \leq \eta\sum_{i=1}^{t}\|(I + \eta A^\star)\|^{t-i}\|\hat{R}(\theta_{i-1})\| \leq \eta\sum_{i=1}^{t}(1 + \eta\lambda_1)^{t-i}C_{\hat{R}}\|\theta_{i-1}\|^3$$

For $i - 1 < t^*_{r(d)}$, we have

$$\|\theta_{i-1}\| \leq 2\|\tilde{\theta}_{i-1}\| = 2\|(I + \eta A^\star)^{i-1}\theta_0\| \leq 2r_0(1 + \eta\lambda_1)^{i-1}.$$

Thus, we conclude that

$$\begin{aligned}
\|\xi_t\| &\leq 8C_{\hat{R}}\eta\, r_0^3 \sum_{i=1}^{t}(1 + \eta\lambda_1)^{t-i}(1 + \eta\lambda_1)^{3(i-1)} \\
&\leq 8C_{\hat{R}}\eta\, r_0^3(1 + \eta\lambda_1)^{t-1}\sum_{j=0}^{t-1}(1 + \eta\lambda_1)^{2j} \\
&\leq 8C_{\hat{R}}\eta\, r_0^3(1 + \eta\lambda_1)^{t-1}\frac{(1 + \eta\lambda_1)^{2t}}{(1 + \eta\lambda_1)^2 - 1} \\
&\leq \frac{4C_{\hat{R}}r_0^3}{\lambda_1}(1 + \eta\lambda_1)^{3t}.
\end{aligned}$$

$\square$

The next lemma gives a lower bound on $t^*_{r(d)}$, which will be useful when later on bounding $t^*_{\not\perp}$ in Proposition D.1.

**Lemma D.5.** *Assume that, for all $\theta$, $\|\hat{R}(\theta)\| \leq C_{\hat{R}}\|\theta\|^3$ for some constant $C_{\hat{R}}$, and let $v_1$ denote the top eigenvector of $A^\star$. Then, we have*

$$t^*_{r(d)} \geq \min\left\{\left\lfloor\frac{\log\left(\frac{\lambda_1|\langle v_1, \theta_0\rangle|}{4C_{\hat{R}}r_0^3}\right)}{2\log(1 + \eta\lambda_1)}\right\rfloor, \left\lfloor\frac{\log\left(\frac{r(d)}{4r_0}\right)}{\log(1 + \eta\lambda_1))}\right\rfloor - 1, \left\lfloor\frac{\log\left(\frac{r(d)}{16\eta C_{\hat{R}}r_0^3}\right)}{3\log(1 + \eta\lambda_1))}\right\rfloor\right\} =: t^*_{\text{lb}}.$$

*Proof.* We will prove this by induction. Let us define the set

$$\mathcal{T} := \left\{t \in \mathbb{N} : \theta_{t+1} \in B_{r(d)} \text{ and } \|\xi_t\| < \|\tilde{\theta}_t\|\right\}.$$

It is immediate to see that $0 \in \mathcal{T}$. We will next prove that if, for arbitrary $t \in \mathbb{N}$, it holds that $[t] \subseteq \mathcal{T}$ and $t + 1 \leq t_{\text{lb}}^*$, then it must be that $(t + 1) \in \mathcal{T}$. By induction, this would directly imply that

$$[t_{\text{lb}}^*] \subset \mathcal{T},$$

from which the claim of the lemma follows.

Thus, let us take an arbitrary $t \in \mathbb{N}$ such that $[t] \subseteq \mathcal{T}$ and $t + 1 \leq t_{\text{lb}}^*$. We can lower bound $\|\tilde{\theta}_{t+1}\|$ as

$$\|\tilde{\theta}_{t+1}\|_2 \geq \left| v_1^\top \tilde{\theta}_{t+1} \right| = \left| v_1^\top (I + \eta A^\star)^{t+1} \theta_0 \right| = (1 + \eta\lambda_1)^{t+1} \left| \langle v_1, \theta_0 \rangle \right|.$$

As $[t] \in \mathcal{T}$ implies $t + 1 \leq t_{r(d)}^* + 1$, we can use Lemma D.4 to get

$$\|\xi_{t+1}\| \leq \frac{4C_{\hat{R}} r_0^3}{\lambda_1} (1 + \eta\lambda_1)^{3(t+1)}.$$

Thus, it will hold that $\|\xi_{t+1}\| \leq \|\tilde{\theta}_{t+1}\|$ when

$$\frac{4C_{\hat{R}} r_0^3}{\lambda_1} (1 + \eta\lambda_1)^{3(t+1)} \leq (1 + \eta\lambda_1)^{t+1} \left| \langle v_1, \theta_0 \rangle \right|.$$

Rearranging for $t$, we get $\|\xi_{t+1}\| \leq \|\tilde{\theta}_{t+1}\|$ when the following inequality holds

$$(1 + \eta\lambda_1)^{2(t+1)} \leq \frac{\lambda_1 \left| \langle v_1, \theta_0 \rangle \right|}{4C_{\hat{R}} r_0^3} \iff t + 1 \leq \frac{\log\left( \frac{\lambda_1 |\langle v_1, \theta_0 \rangle|}{4C_{\hat{R}} r_0^3} \right)}{2\log(1 + \eta\lambda_1)}.$$

As the RHS of the previous equation holds for $t + 1$, we conclude that $\|\xi_{t+1}\| \leq \|\tilde{\theta}_{t+1}\|$. It is left to prove that $\theta_{t+2} \in B_{r(d)}$. By (D.5), we have $\theta_{t+2} = (I + \eta A^\star)\theta_{t+1} - \eta\hat{R}(\theta_{t+1})$. By using Lemma D.3 and the fact that $\|\xi_{t+1}\| \leq \|\tilde{\theta}_{t+1}\|$, we get

$$\begin{aligned}
\|\theta_{t+2}\| &\leq (1 + \eta\lambda_1)\|\theta_{t+1}\| + \|\eta\hat{R}(\theta_{t+1})\| \\
&\leq (1 + \eta\lambda_1)\|\theta_{t+1}\| + \eta C_{\hat{R}}\|\theta_{t+1}\|^3 \\
&\leq 2(1 + \eta\lambda_1)\|\tilde{\theta}_{t+1}\| + 8\eta C_{\hat{R}}\|\tilde{\theta}_{t+1}\|^3 \\
&\leq 2r_0(1 + \eta\lambda_1)^{t+2} + 8\eta C_{\hat{R}} r_0^3 (1 + \eta\lambda_1)^{3(t+1)} \\
&\leq r(d),
\end{aligned}$$

where the last inequality follows from the fact that $t + 1 \leq t_{\text{lb}}^*$. $\qquad \square$

The next lemma will be useful to track the evolution of $\theta_t$ in the direction of $v_1$, as $v_1$ is correlated with $\theta^\star$ by Proposition C.1.

**Lemma D.6.** *Let $V_1 = \text{span}\{v_1\}$ and $V_1^\perp$ be its orthogonal complement. Let $\lambda_2$ denote the second largest eigenvalue of $A^\star$, or equivalently the largest eigenvalue of $A^\star$ on $V_1^\perp$. We define $\theta_t^{v_1} := (v_1 v_1^\top)\theta_t$ and $\theta_t^{v_1^\perp} := (I - v_1 v_1^\top)\theta_t$. Then, for $t \leq t_{r(d)}^*$, it holds that*

$$\frac{\|\theta_t^{v_1^\perp}\|}{\|\theta_t^{v_1}\|} \leq \frac{\sqrt{r_0^2 - \langle\theta_0, v_1\rangle^2}(1 + \eta\lambda_2)^t + \|\xi_t\|}{|\langle\theta_0, v_1\rangle| (1 + \eta\lambda_1)^t - \|\xi_t\|}.$$

*Proof.* As $\theta_t = \tilde{\theta}_t + \xi_t$, we have

$$\|\theta_t^{v_1}\| = \|(v_1 v_1^\top)\tilde{\theta}_t + (v_1 v_1^\top)\xi_t\|_2 \geq \|(v_1 v_1^\top)\tilde{\theta}_t\| - \|\xi_t\| \geq \|\theta_t^{v_1}\|_2 = |\langle v_1, \theta_0\rangle| (1 + \eta\lambda_1)^t - \|\xi_t\|_2,$$

where the last passage uses that $(v_1 v_1^\top)\tilde{\theta}_t = (1 + \eta\lambda_1)^t (v_1 v_1^\top \theta_0)$. Furthermore, in the directions orthogonal to $v_1$, it holds that

$$\|\theta_t^{v_1^\perp}\| = \|(I - v_1 v_1^\top)\tilde{\theta}_t + (I - v_1 v_1^\top)\xi_t\| \leq \|(I - v_1 v_1^\top)\tilde{\theta}_t\| + \|\xi_t\|.$$

The operator $(I + \eta A^\star)^t$ restricted to $V_1^\perp$ has norm $(1 + \eta\lambda_2)^t$, so

$$\|(I - v_1 v_1^\top)\tilde{\theta}_t\|_2 = \|(I + \eta A^\star)^t (I - v_1 v_1^\top)\theta_0\| \le (1 + \eta\lambda_2)^t \|(I - v_1 v_1^\top)\theta_0\|.$$

This gives

$$\|\theta_t^{v_1^\perp}\|_2 \le \|(I - v_1 v_1^\top)\theta_0\|(1 + \eta\lambda_2)^t + \|\xi_t\|$$
$$= \sqrt{r_0^2 - \langle\theta_0, v_1\rangle^2}(1 + \eta\lambda_2)^t + \|\xi_t\|.$$

Forming the ratio $\|\theta_t^{v_1^\perp}\|/\|\theta_t^{v_1}\|$ gives the desired bound. □

The next lemma gives an upper and lower bound on $|\langle\theta_0, v_1\rangle|$.

**Lemma D.7.** *For large enough $n$ and $d$, it holds that*

$$\frac{r_0}{\sqrt[4]{d}} \ge |\langle\theta_0, v_1\rangle| \ge \frac{r_0}{8d^{2.6}}\sqrt{\frac{\pi}{2}}, \tag{D.7}$$

*with probability at least $1 - \frac{1}{d^{2.1}}$.*

*Proof.* As $\theta_0$ is independently sampled from $v_1$, it holds, by symmetry, that $\langle\theta_0/\|\theta_0\|_2, v_1\rangle$ has the density of the first coordinate of the random vector uniformly sampled from the unit sphere. This random variable has density in $[-1, 1]$ (Vershynin, 2026, Exercise 3.27) as

$$f(u) = c_d(1 - t^2)^{\frac{d-3}{2}}, \quad c_d := \frac{\Gamma(d/2)}{\sqrt{\pi}\Gamma((d-1)/2)}.$$

As $f(u) \le c_d$, for any $\epsilon \in [0, 1]$, it holds that

$$\mathbb{P}\left(|\langle\theta_0/\|\theta_0\|_2, v_1\rangle| \le \epsilon\right) = 2\int_0^\epsilon f(u)du \le 2\epsilon c_d.$$

By Gautchi's inequality (Gautschi, 1959), it holds that

$$c_d = \frac{\Gamma(d/2)}{\sqrt{\pi}\Gamma((d-1)/2)} \le \sqrt{\frac{d}{2\pi}},$$

which gives

$$\mathbb{P}\left(|\langle\theta_0/\|\theta_0\|_2, v_1\rangle| \le \epsilon\right) \le 2\epsilon\sqrt{\frac{d}{2\pi}}.$$

Taking $\epsilon = \frac{1}{8d^{2.6}}\sqrt{\frac{\pi}{2}}$ gives the desired lower bound.

For the upper bound we again rely on the independence of $\theta_0$ and $v_1$. Note that $\theta_0/\|\theta_0\|_2 \to \langle\theta_0/\|\theta_0\|_2, v_1\rangle$ is a 1-Lipschitz function on the sphere. Note that $\mathbb{E}\langle\theta_0/\|\theta_0\|_2, v_1\rangle = 0$ by symmetry, so we can use the concentration of Lipschitz functions on the sphere (Vershynin, 2018, Theorem 5.1.4) to get that

$$\mathbb{P}\left(|\langle\theta_0/\|\theta_0\|_2, v_1\rangle| \ge t\right) \le 2\exp(-c_1 dt^2),$$

for some constant $c_1$. Taking $t = d^{1/4}$ gives the desired upper bound and a union bound between the two events giving upper and lower bound concludes the proof. □

We now have all the ingredients to prove Proposition D.1.

*Proof of Proposition D.1.* Note that, as $\theta^\star \in \mathcal{S}^{d-1}$, it holds

$$\frac{|\langle \theta_t, \theta^\star \rangle|}{\|\theta_t\|\|\theta^\star\|} = |\langle \theta_t/\|\theta_t\|, \theta^\star \rangle|.$$

Recall that $v_1$ denotes the top eigenvector of the matrix $A^\star$, so that

$$
\begin{aligned}
|\langle \theta_t/\|\theta_t\|, \theta^\star \rangle| &= |\langle \theta_t/\|\theta_t\| - v_1, \theta^\star \rangle + \langle v_1, \theta^\star \rangle| \\
&\geq |\langle v_1, \theta^\star \rangle| - |\langle \theta_t/\|\theta_t\| - v_1, \theta^\star \rangle| \\
&\geq 1 - C(e^{-M/2} + M\delta^{-1/2}) - |\langle \theta_t/\|\theta_t\| - v_1, \theta^\star \rangle|,
\end{aligned}
$$

where the last inequality holds with probability $1 - e^{-cd}$ from Proposition B.1, without assumption $\delta > CM^4$. Note that, for each fixed $\theta_t$, we can always choose the sign of $v_1$ such that $\langle \theta_t, v_1 \rangle \geq 0$. Using that fact, we bound $|\langle \theta_t/\|\theta_t\| - v_1, \theta^\star \rangle|$ as

$$
\begin{aligned}
|\langle \theta_t/\|\theta_t\| - v_1, \theta^\star \rangle| &\leq \|\theta_t/\|\theta_t\|_2 - v_1\| \\
&= \sqrt{2 - 2|\langle \theta_t/\|\theta_t\|, v_1 \rangle|} \\
&= \sqrt{2 - \frac{2}{\sqrt{1 + \|\theta_t^{v_1^\perp}\|^2/\|\theta_t^{v_1}\|^2}}}.
\end{aligned}
\tag{D.8}
$$

Next, Lemma D.4 and Lemma D.6 give that, for any $t \leq t^*_{r(d)}$,

$$\frac{\|\theta_t^{v_1^\perp}\|}{\|\theta_t^{v_1}\|} \leq \frac{\sqrt{r_0^2 - \langle \theta_0, v_1 \rangle^2}(1 + \eta\lambda_2)^t + \|\xi_t\|}{|\langle \theta_0, v_1 \rangle|(1 + \eta\lambda_1)^t - \|\xi_t\|} \leq \frac{\sqrt{r_0^2 - \langle \theta_0, v_1 \rangle^2}(1 + \eta\lambda_2)^t + \frac{4C_{\hat{R}}r_0^3}{\lambda_1}(1 + \eta\lambda_1)^{3t}}{|\langle \theta_0, v_1 \rangle|(1 + \eta\lambda_1)^t - \frac{4C_{\hat{R}}r_0^3}{\lambda_1}(1 + \eta\lambda_1)^{3t}}.$$

Let $t^*_\xi$ be such that, for any $t \leq t^*_\xi$,

$$\frac{4C_{\hat{R}}r_0^3}{\lambda_1}(1 + \eta\lambda_1)^{3t} \leq \min\left\{\frac{1}{2}|\langle \theta_0, v_1 \rangle|(1 + \eta\lambda_1)^t, \ \frac{1}{2}\sqrt{r_0^2 - \langle \theta_0, v_1 \rangle^2}(1 + \eta\lambda_2)^t\right\}.$$

A direct calculation gives that

$$t^*_\xi = \min\left\{\frac{\log\left(\frac{\lambda_1}{8C_{\hat{R}}r_0^3}|\langle \theta_0, v_1 \rangle|\right)}{2\log(1 + \eta\lambda_1)}, \ \frac{\log\left(\frac{\lambda_1}{8C_{\hat{R}}r_0^3}\sqrt{r_0^2 - \langle \theta_0, v_1 \rangle^2}\right)}{\log\left(\frac{(1 + \eta\lambda_1)^3}{1 + \eta\lambda_2}\right)}\right\}.$$

Then, for any $t \leq t^*_\xi$, and large enough $d$,

$$\frac{\|\theta_t^{v_1^\perp}\|}{\|\theta_t^{v_1}\|} \leq 3\sqrt{\frac{r_0^2}{\langle \theta_0, v_1 \rangle^2} - 1} \cdot \left(\frac{1 + \eta\lambda_2}{1 + \eta\lambda_1}\right)^t \leq Cd^{5/2} \cdot \left(\frac{1 + \eta\lambda_2}{1 + \eta\lambda_1}\right)^t,$$

where the last passage follows from (D.7). We now want to find $t$ such that the RHS of the previous bound is smaller than $M\delta^{-1/2}$. To do so, we note that

$$Cd^{5/2} \cdot \left(\frac{1 + \eta\lambda_2}{1 + \eta\lambda_1}\right)^t \leq M^2\delta^{-1}$$

is implied by taking

$$t \geq t^*_{\not\leq} := \frac{3\log d}{\log(1 + 1.99\eta)},$$

as

$$t \geq \frac{3\log d}{\log(1 + 1.99\eta)} \geq \frac{3\log d}{\log(1 + (\lambda_1 - \lambda_2)\eta)} \geq \frac{3\log d}{\log\left(\frac{1 + \eta\lambda_1}{1 + \eta\lambda_2}\right)} \implies Cd^{5/2} \cdot \left(\frac{1 + \eta\lambda_2}{1 + \eta\lambda_1}\right)^t \leq M^2\delta^{-1},$$

due to the bounds on $\lambda_1$ and $\lambda_2$ from Proposition B.1, the concavity of $\log$ and the inequality $d^3 \geq Cd^{5/2}M^{-2}\delta$ for large enough $d$. Let us set $t_u^* := \min\left\{t_{r(d)}^*, t_\xi^*\right\}$. Then, for any $t \in [t_{\measuredangle}^*, t_u^*]$ it holds

$$\frac{\|\theta_t^{v_1^\perp}\|}{\|\theta_t^{v_1}\|} \leq M^2\delta^{-1}.$$

Plugging this back into (D.8) yields

$$|\langle\theta_t/\|\theta_t\|_2 - v_1, \theta^\star\rangle| \leq CM\delta^{-1/2},$$

which implies that almost surely

$$|\langle\theta_t/\|\theta_t\|_2, \theta^\star\rangle| \geq 1 - C(e^{-M/2} + M\delta^{-1/2}).$$

Doing a Taylor expansion of $\arccos(1 - C(e^{-M/2} + M\delta^{-1/2}))$ gives (D.1), for large enough $M$ and $\delta \geq CM^2$.

It is only left to prove that $\min\left\{t_{r(d)}^*, t_\xi^*\right\} = t_u^* > t_{\measuredangle}^*$, which we do by proving separately that $t_{\measuredangle}^* < t_{r(d)}^*$ and $t_{\measuredangle}^* < t_\xi^*$.

**Proof of the claim that $t_{\measuredangle}^* < t_{r(d)}^*$.** By Lemma D.5, we have that $t_{r(d)}^* \geq t_{\text{lb}}^*$. Thus, it suffices to show that $t_{\measuredangle}^*$ is strictly smaller than each of the three terms in the minimum defining $t_{\text{lb}}^*$. For the first term, we use the upper and lower bound on $\lambda_1$ from Proposition B.1 (which hold with probability $1 - e^{-d}$), the upper bound on $C_{\hat{R}}$ from Lemma D.3 (which holds with probability $1 - \exp(-n^{1/2})$) and the lower bound on $|\langle v_1, \theta_0\rangle|$ from Lemma D.7 (which holds with probability $1 - \frac{1}{d^{2.1}}$) to obtain for large enough $d$

$$\left\lfloor\frac{\log\left(\frac{\lambda_1|\langle v_1, \theta_0\rangle|}{4C_{\hat{R}}r_0^3}\right)}{2\log(1 + \eta\lambda_1)}\right\rfloor \geq \frac{\log(d^{-5}r_0^{-2})}{2\log(1 + \eta\lambda_1)} = \frac{\log d^{25}}{2\log(1 + \eta\lambda_1)} \geq \frac{10\log(d)}{\log(1 + 3.01\eta)}.$$

Note, that by the union bound on probabilities, the previous inequality holds with probability at least $1 - \frac{2}{d^{2.1}}$, for large enough $d$. On the other hand, from upper and lower bound on $\lambda_1$ and $\lambda_2$ from Proposition B.1 (which hold with probability $1 - e^{-d}$) we get

$$t_{\measuredangle}^* = \frac{3\log d}{\log(1 + 1.99\eta)} \leq \frac{10\log(d)}{\log(1 + 3.01\eta)},$$

where since for $\eta < 1/100$, inequality $\frac{10}{\log(1+3.01\eta)} > \frac{3}{\log(1+1.99\eta)}$ holds. Similarly we can get that, with probability at least $1 - \frac{1}{d^{2.1}}$,

$$\frac{\log\left(\frac{r(d)}{4r_0}\right)}{\log(1 + \eta\lambda_1)} \geq \frac{14\log(d)}{\log(1 + 3.01\eta)}, \quad \text{and} \quad \frac{\log\left(\frac{r(d)}{16\eta C_{\hat{R}}r_0^3}\right)}{3\log(1 + \eta\lambda_1)} \geq \frac{10\log(d)}{\log(1 + 3.01\eta)},$$

from which follows that $t_{\measuredangle}^* < t_{r(d)}^*$.

**Proof of claim that $t_{\measuredangle}^* < t_\xi^*$.** By same argument as before we can get that

$$t_{\measuredangle}^* \leq \frac{10\log d}{\log(1 + 3.01\eta)} < \frac{\log\left(\frac{\lambda_1}{8C_{\hat{R}}r_0^3}|\langle\theta_0, v_1\rangle|\right)}{2\log(1 + \eta\lambda_1)},$$

with probability at least $1 - \frac{1}{d^{2.1}}$. Moreover, by upper and lower bound on $\lambda_1$ from Proposition C.1, the upper bound on $C_{\hat{R}}$ from Lemma D.3, and the upper bound on $|\langle v_1, \theta_0\rangle|$ from Lemma D.7, it holds with probability at least $1 - 2/d^{2.1}$,

$$\frac{\log\left(\frac{\lambda_1}{8C_{\hat{R}}r_0^3}(r_0 - |\langle\theta_0, v_1\rangle|)\right)}{3\log(1 + \eta\lambda_1) - \log(1 + \eta\lambda_2)} \geq \frac{13\log d}{3\log(1 + 2.99\eta) - \log(1 + 1.01\eta)}.$$

As for $\eta < 1/100$, inequality $\frac{13}{3\log(1+2.99\eta)-\log(1+1.01\eta)} > \frac{3}{\log(1+1.99\eta)}$ holds, we have that $t_{\measuredangle}^* < t_\xi^*$. $\qquad\square$

## D.2. Uniform concentration of the empirical gradient

Fix $R \geq 1$ and $r \in (0, R]$, and define the annulus

$$\Theta := \{\theta \in \mathbb{R}^d : r \leq \|\theta\| \leq R\}. \tag{D.9}$$

**Lemma D.8.** *Let $\widehat{\mathcal{G}}(\theta)$ and $\mathcal{G}(\theta)$ be the empirical and population gradient, respectively, defined in (4.3) and (4.7). Assume $M > 1$ and that $\sqrt{M}/R$ is lower bounded by a sufficiently large constant. Fix $\epsilon \in (0, 1/2)$. Then, there exist constants $c, C > 0$, such that if*

$$n \geq C \frac{M^2}{\epsilon^2} \left( d \log \left( \frac{M}{\epsilon} \right) + \log \left( \log \left( \frac{R}{r} \right) + 2 \right) \right), \tag{D.10}$$

*then with probability at least $1 - 6e^{-cd}$,*

$$\sup_{\theta \in \Theta} \frac{\|\widehat{\mathcal{G}}(\theta) - \mathcal{G}(\theta)\|}{\|\theta\|} \leq \epsilon.$$

### D.2.1. PRELIMINARY RESULTS

**Lemma D.9** (Lemma 2.7.7 in (Vershynin, 2018)). *Let $X, Y$ be real random variables (no independence assumed) with $\|X\|_{\psi_2} \leq K$ and $\|Y\|_{\psi_2} \leq L$. Then, $\|XY\|_{\psi_1} \leq KL$.*

**Lemma D.10** (Theorem 2.8.1 in (Vershynin, 2018)). *Let $Y_1, \ldots, Y_n$ be i.i.d. mean-zero with $\|Y_1\|_{\psi_1} \leq K$. Then for all $t > 0$,*

$$\mathbb{P}\left( \left| \frac{1}{n} \sum_{i=1}^{n} Y_i \right| > t \right) \leq 2 \exp\left( -cn \min\left\{ \frac{t^2}{K^2}, \frac{t}{K} \right\} \right),$$

*for a universal constant $c > 0$.*

**Lemma D.11** (Annulus net with multiplicative radii). *Fix $\varepsilon \in (0, 1/10)$. There exists a finite set $\mathcal{N}_\varepsilon \subset \Theta$ such that:*

- *For every $\theta \in \Theta$ there exists $\hat{\theta} \in \mathcal{N}_\varepsilon$ with*

$$\|\hat{\theta}\| \leq \|\theta\| \leq (1 + \varepsilon)\|\hat{\theta}\|, \qquad \|\theta - \hat{\theta}\| \leq 2\varepsilon\|\theta\|. \tag{D.11}$$

- *The cardinality satisfies*

$$|\mathcal{N}_\varepsilon| \leq (J + 1) \left( \frac{3}{\varepsilon} \right)^d, \qquad J := \left\lceil \frac{\log(R/r)}{\log(1 + \varepsilon)} \right\rceil. \tag{D.12}$$

*Proof.* Let $\mathcal{U}_\varepsilon$ be an $\varepsilon$-net of the unit sphere in $d$ dimensions, namely for any $u \in \mathcal{S}^{d-1}$, there exists $\hat{u} \in \mathcal{U}_\varepsilon$ such that $\|u - \hat{u}\| \leq \varepsilon$. Then, by Corollary 4.2.13 of (Vershynin, 2018), we have that

$$|\mathcal{U}_\varepsilon| \leq \left( \frac{3}{\varepsilon} \right)^d.$$

Next, define radii $r_j = r(1 + \varepsilon)^j$ for $j \in \{0, \ldots, J\}$, where $J$ is the smallest integer such that $r_J \geq R$ (so that $J := \left\lceil \frac{\log(R/r)}{\log(1+\varepsilon)} \right\rceil$ is a valid choice). We now define the set $\mathcal{N}_\varepsilon := \{r_j \hat{\theta} : \hat{\theta} \in \mathcal{U}_\varepsilon, j \in \{0, \ldots, J\}\}$. Then, (D.12) holds. It remains to show (D.11). Let $\theta = \bar{r}u$, with $\bar{r} = \|\theta\|$ and $u \in \mathcal{S}^{d-1}$. Pick $j$ such that $r_j \leq \bar{r} \leq r_j(1 + \varepsilon)$, $\hat{u} \in \mathcal{U}_\varepsilon$ such that $\|u - \hat{u}\| \leq \varepsilon$ and $\hat{\theta} = r_j \hat{u}$. Then, the inequality on $\|\theta\|$ in (D.11) follows immediately and the other inequality is obtained below:

$$\|\hat{\theta} - \theta\| = \|\bar{r}u - r_j\hat{u}\| \leq |\bar{r} - r_j| \cdot \|u\| + r_j\|u - \hat{u}\| \leq 2\varepsilon r_j \leq 2\varepsilon\|\theta\|.$$

$\square$

For $s \in (0, R]$, $\tau > 0$ and $g \sim N(0, 1)$ define

$$p_s(\tau) := \mathbb{P}\left( \left| |sg| - \sqrt{M} \right| \leq \tau \right).$$

**Lemma D.12** (Tail of slab). *There is a universal $C > 0$ such that for all $s \in (0, R]$ and $\tau \in (0, \sqrt{M}/2]$,*

$$p_s(\tau) \leq C \frac{\tau}{s} \exp\left(-\frac{M}{8s^2}\right), \tag{D.13}$$

$$\frac{\sqrt{p_s(\tau)}}{s} \leq C \frac{\sqrt{\tau}}{M^{3/4}}. \tag{D.14}$$

*Proof.* Let $Y := sg$. Then $Y \sim \mathcal{N}(0, s^2)$ and

$$p_s(\tau) = \mathbb{P}\big(\big||Y| - \sqrt{M}\big| \leq \tau\big) = \mathbb{P}\big(\sqrt{M} - \tau \leq |Y| \leq \sqrt{M} + \tau\big).$$

Since $\tau < \sqrt{M}$, we have $\sqrt{M} - \tau > 0$, and therefore

$$\{\sqrt{M} - \tau \leq |Y| \leq \sqrt{M} + \tau\} = \{Y \in [\sqrt{M} - \tau, \sqrt{M} + \tau]\} \cup \{Y \in [-(\sqrt{M} + \tau), -(\sqrt{M} - \tau)]\}.$$

Let $\varphi_s$ be the probability density function of $Y$. By symmetry of $\varphi_s$ (it is even), the two probabilities are equal, hence for $\tau \leq \sqrt{M}/2$ using the monotonicity of $\varphi_s$, we have

$$p_s(\tau) = 2\,\mathbb{P}\big(Y \in [\sqrt{M} - \tau, \sqrt{M} + \tau]\big) = 2 \int_{\sqrt{M}-\tau}^{\sqrt{M}+\tau} \varphi_s(y)\,dy \leq 4\tau\varphi_s(\sqrt{M} - \tau) \leq 4\tau\varphi_s(\sqrt{M}/2)$$

$$\leq \frac{4}{\sqrt{2\pi}}\frac{\tau}{s} \exp\left(-\frac{M}{8s^2}\right),$$

which gives (D.13).

To prove (D.14), we note that by (D.13) we have

$$\frac{\sqrt{p_s(\tau)}}{s} \leq C\sqrt{\tau} \cdot s^{-3/2} \exp\left(-\frac{M}{16s^2}\right).$$

The function $g(s) := s^{-3/2} \exp(-\frac{M}{16s^2})$ on $(0, \infty)$ is maximized at $s = \frac{\sqrt{M}}{\sqrt{12}}$, and $g\left(\frac{\sqrt{M}}{\sqrt{12}}\right) = 12^{3/4}e^{-3/4}M^{-3/4}$, which yields (D.14). $\qquad\square$

**Lemma D.13** (Uniform empirical slab control). *Let $x_1, \ldots, x_n \overset{\text{i.i.d.}}{\sim} \mathcal{N}(0, I_d)$. Fix $\tau \in (0, M/2]$ and any finite set $\mathcal{T} \subset \mathbb{R}^d$ with $\|t\| \leq R$ for all $t \in \mathcal{T}$. Also assume $R \leq C\sqrt{M}$ with $C$ a sufficiently large constant. Then there exist universal constants $c, C > 0$ such that, with probability at least*

$$1 - |\mathcal{T}| \exp\big(-cn\tau/\sqrt{M}\big),$$

*we have*

$$\frac{1}{n}\sum_{i=1}^n \mathbf{1}\{|\,|\langle x_i, t\rangle| - \sqrt{M}| \leq \tau\} \leq \frac{\tau\|t\|^2}{M^{3/2}}.$$

*Proof.* Fix $t \neq 0$ and set $s := \|t\|$. Since $x_i \sim \mathcal{N}(0, I_d)$, we have $\langle x_i, t\rangle \sim \mathcal{N}(0, s^2)$, hence $\langle x_i, t\rangle \overset{d}{=} sg_i$ with $\{g_i\}_{i=1}^n$ i.i.d. standard Gaussians. Therefore,

$$p := \mathbb{P}\Big(\big||\langle x_i, t\rangle| - \sqrt{M}\big| \leq \tau\Big) = \mathbb{P}\Big(\big||sg| - \sqrt{M}\big| \leq \tau\Big) = p_s(\tau).$$

Applying the slab-tail bound in (D.13) yields

$$p \leq C\frac{\tau}{s} \exp\Big(-\frac{M}{8s^2}\Big). \tag{D.15}$$

Let $a := \tau s^2/M^{3/2}$. Since $\mathbf{1}\{\big||\langle x_i, t\rangle| - \sqrt{M}\big| \leq \tau\} \in \{0, 1\}$ are i.i.d. with mean $p$, an application of Chernoff's bound (Theorem 2.3.1 in (Vershynin, 2018)) gives that

$$\mathbb{P}\Big\{\frac{1}{n}\sum_{i=1}^n \mathbf{1}\{\big||\langle x_i, t\rangle| - \sqrt{M}\big| \leq \tau\} \geq a\Big\} \leq \exp\Big(-na\log(a/(ep))\Big). \tag{D.16}$$

Now plug $a = \tau s^2 / M^{3/2}$ and the upper bound (D.15) into (D.16). We have

$$\log \frac{a}{ep} = \log\left(\frac{\tau s^2}{eM^{3/2}}\right) - \log p \geq \log\left(\frac{\tau s^2}{eM^{3/2}}\right) - \log\left(C\frac{\tau}{s}\right) + \frac{M}{8s^2} = \frac{M}{8s^2} + \log\left(\frac{s^3}{CeM^{3/2}}\right).$$

Therefore,

$$\mathbb{P}\left\{\frac{1}{n}\sum_{i=1}^{n}\mathbf{1}\{||\langle x_i, t\rangle| - \sqrt{M}| \leq \tau\} \geq \frac{\tau s^2}{M^{3/2}}\right\} \leq \exp\left(-n\frac{\tau s^2}{M^{3/2}}\left[\frac{M}{8s^2} + \log\left(\frac{s^3}{CeM^{3/2}}\right)\right]\right).$$

Finally, under the condition $R \leq C\sqrt{M}$ for some other sufficiently large constant $C$, we have $\log\left(\frac{s^3}{CeM^{3/2}}\right) \geq -\frac{M}{16s^2}$, hence the bracket above is at least $\frac{M}{16s^2}$. Plugging this lower bound gives

$$\mathbb{P}\left\{\frac{1}{n}\sum_{i=1}^{n}\mathbf{1}\{||\langle x_i, t\rangle| - \sqrt{M}| \leq \tau\} \geq \frac{\tau s^2}{M^{3/2}}\right\} \leq \exp\left(-n\frac{\tau s^2}{M^{3/2}} \cdot \frac{M}{16s^2}\right) = \exp\left(-\frac{n\tau}{16\sqrt{M}}\right),$$

and a union bound over $t \in \mathcal{T}$ gives the claim. $\qquad\square$

### D.2.2. PROOF OF LEMMA D.8

For any $\theta$,

$$\|\widehat{\mathcal{G}}(\theta) - \mathcal{G}(\theta)\| = \sup_{u \in \mathcal{S}^{d-1}}\left\langle u, \widehat{\mathcal{G}}(\theta) - \mathcal{G}(\theta)\right\rangle.$$

For $\theta \neq 0$ and $u \in \mathcal{S}^{d-1}$ define

$$Z_{\theta,u}(x) := \left(\sigma(\langle x, \theta\rangle) - \sigma(\langle x, \theta^\star\rangle)\right)\sigma'(\langle x, \theta\rangle)\langle u, x\rangle, \qquad W_{\theta,u}(x) := \frac{Z_{\theta,u}(x)}{\|\theta\|}.$$

Then $\left\langle u, \widehat{\mathcal{G}}(\theta)\right\rangle = \frac{1}{n}\sum_{i=1}^{n} Z_{\theta,u}(x_i)$ and therefore

$$\frac{\|\widehat{\mathcal{G}}(\theta) - \mathcal{G}(\theta)\|}{\|\theta\|} = \sup_{u \in \mathcal{S}^{d-1}}\left|\frac{1}{n}\sum_{i=1}^{n}(W_{\theta,u}(x_i) - \mathbb{E}[W_{\theta,u}(x)])\right|.$$

Thus it suffices to prove that, with probability at least $1 - 6e^{-cd}$,

$$\sup_{\theta \in \Theta}\sup_{u \in \mathcal{S}^{d-1}}\left|\frac{1}{n}\sum_{i=1}^{n}(W_{\theta,u}(x_i) - \mathbb{E}[W_{\theta,u}(x)])\right| \leq \epsilon. \tag{D.17}$$

Fix $\varepsilon, \eta \in (0, 1/10)$. Let $\mathcal{N}_\varepsilon$ be the annulus net from Lemma D.11, and let $V_\eta$ be an $\eta$-net of $\mathcal{S}^{d-1}$ with $|V_\eta| \leq (3/\eta)^d$. For each $(\theta, u)$, choose $(\hat{\theta}, \hat{u}) \in \mathcal{N}_\varepsilon \times V_\eta$ such that (D.11) holds and $\|u - \hat{u}\| \leq \eta$. For any $(\theta, u)$ and their approximant $(\hat{\theta}, \hat{u})$,

$$\left|\frac{1}{n}\sum_{i=1}^{n}(W_{\theta,u}(x_i) - \mathbb{E}[W_{\theta,u}(x)])\right| \leq \underbrace{\left|\frac{1}{n}\sum_{i=1}^{n}\left(W_{\hat{\theta},\hat{u}}(x_i) - \mathbb{E}[W_{\hat{\theta},\hat{u}}(x)]\right)\right|}_{T_1(\hat{\theta},\hat{u})} + \underbrace{\frac{1}{n}\sum_{i=1}^{n}\left|W_{\theta,u}(x_i) - W_{\hat{\theta},\hat{u}}(x_i)\right|}_{T_2(\theta,u)}$$

$$+ \underbrace{\mathbb{E}\left|W_{\theta,u}(x) - W_{\hat{\theta},\hat{u}}(x)\right|}_{T_3(\theta,u)}. \tag{D.18}$$

Taking $\sup_{\theta,u}$ yields

$$\sup_{\theta\in\Theta}\sup_{u\in\mathcal{S}^{d-1}}\left|\frac{1}{n}\sum_{i=1}^{n}(W_{\theta,u}(x_i)-\mathbb{E}[W_{\theta,u}(x)])\right| \leq \underbrace{\sup_{\hat{\theta}\in\mathcal{N}_\varepsilon,\hat{u}\in V_\eta}\left|\frac{1}{n}\sum_{i=1}^{n}\left(W_{\hat{\theta},\hat{u}}(x_i)-\mathbb{E}[W_{\hat{\theta},\hat{u}}(x)]\right)\right|}_{T_1}$$

$$+\underbrace{\sup_{\theta\in\Theta}\sup_{u\in\mathcal{S}^{d-1}}\frac{1}{n}\sum_{i=1}^{n}\left|W_{\theta,u}(x_i)-W_{\hat{\theta},\hat{u}}(x_i)\right|}_{T_2}$$

$$+\underbrace{\sup_{\theta\in\Theta}\sup_{u\in\mathcal{S}^{d-1}}\mathbb{E}\left|W_{\theta,u}(x)-W_{\hat{\theta},\hat{u}}(x)\right|}_{T_3}.$$

**Bounding $T_1$.** We use the sharper bound $|\sigma'(z)| = 2\,|z|\,\mathbf{1}\{|t|\leq\sqrt{M}\}\leq 2\,|z|$. Also $0\leq\sigma(z)\leq M$ for all $z$, hence $|\sigma(\langle x,\theta\rangle)-\sigma(\langle x,\theta^\star\rangle)|\leq M$. Therefore, for any $\theta\neq 0$,

$$|W_{\theta,u}(x)| = \frac{|Z_{\theta,u}(x)|}{\|\theta\|}\leq\frac{M\cdot 2\,|\langle x,\theta\rangle|\cdot|\langle u,x\rangle|}{\|\theta\|}=2M\,|\langle x,\theta/\|\theta\|\rangle|\,|\langle u,x\rangle|. \tag{D.19}$$

Since for Gaussian $x$, each linear form $\langle x,a\rangle$ with $\|a\|=1$ is sub-Gaussian with $\psi_2$ norm $\leq C$, Lemma D.9 and (D.19) imply

$$\|W_{\theta,u}(x)\|_{\psi_1}\leq CM\qquad\text{for all }\theta\neq 0,\ u\in\mathcal{S}^{d-1}.$$

Hence for each fixed $(\hat{\theta},\hat{u})$, the mean-zero variable $W_{\hat{\theta},\hat{u}}(x)-\mathbb{E}W_{\hat{\theta},\hat{u}}(x)$ has $\psi_1$ norm $\leq CM$. Applying Lemma D.10 and union-bounding over $\mathcal{N}_\varepsilon\times V_\eta$ gives

$$\mathbb{P}\Big\{T_1>\epsilon/3\Big\}\leq 2|\mathcal{N}_\varepsilon||V_\eta|\exp\Big(-cn\min\Big\{\frac{\epsilon^2}{M^2},\frac{\epsilon}{M}\Big\}\Big).$$

Using $|\mathcal{N}_\varepsilon|\leq(J+1)(3/\varepsilon)^d$ and $|V_\eta|\leq(3/\eta)^d$, we get that $T_1\leq\epsilon/3$ with probability at least $1-2e^{-cd}$ provided

$$n\ \geq\ C\max\left(\frac{M^2}{\epsilon^2},\frac{M}{\epsilon}\right)\left(d\log\frac{3}{\varepsilon}+d\log\frac{3}{\eta}+\log(J+1)\right). \tag{D.20}$$

**Bounding $T_2$ and $T_3$.** To bound this expression we utilize the simple identity

$$W_{\theta,u}(x)-W_{\hat{\theta},\hat{u}}(x)=W_{\theta,u}(x)-W_{\theta,\hat{u}}(x)+W_{\theta,\hat{u}}(x)-W_{\hat{\theta},\hat{u}}(x)$$

We proceed by bounding each of these terms. First, let us look at variations in $u$ with $\theta$ fixed. To this aim, we write

$$W_{\theta,u}(x)-W_{\theta,\hat{u}}(x)=\frac{(\sigma(\langle x,\theta\rangle)-\sigma(\langle x,\theta^\star\rangle))\sigma'(\langle x,\theta\rangle)}{\|\theta\|}\cdot\langle u-\hat{u},x\rangle.$$

Using $|\sigma(z_1)-\sigma(z_2)|\leq M$ for all $z_1,z_2$ and $|\sigma'(\langle x,\theta\rangle)|\leq 2\,|\langle x,\theta\rangle|$,

$$|W_{\theta,u}(x)-W_{\theta,\hat{u}}(x)|\leq 2M\frac{|\langle x,\theta\rangle|}{\|\theta\|}\,|\langle u-\hat{u},x\rangle|=2M\,|\langle x,\theta/\|\theta\|\rangle|\,|\langle u-\hat{u},x\rangle|.$$

By Corollary 7.3.3 in (Vershynin, 2018), we have that $\frac{1}{n}\sum_{i=1}^{n}x_ix_i^T\preceq 2I$ with probability at least $1-e^{-cd}$. Thus, by Cauchy-Schwarz we have

$$\frac{1}{n}\sum_{i=1}^{n}|\langle x,\theta/\|\theta\|\rangle|\,|\langle u-\hat{u},x\rangle|\leq\sqrt{\frac{1}{n}\sum_{i=1}^{n}\langle x,\theta/\|\theta\|\rangle^2}\sqrt{\frac{1}{n}\sum_{i=1}^{n}\langle u-\hat{u},x\rangle^2}\leq\sqrt{2}\cdot\sqrt{2}\,\|u-\hat{u}\|\leq 2\eta.$$

Thus on this event,

$$\sup_{\theta,u} \frac{1}{n} \sum_{i=1}^{n} |W_{\theta,u} - W_{\theta,\hat{u}}| \le 4M\,\eta. \tag{D.21}$$

Similarly, using $\mathbb{E} \langle x, a \rangle^2 = \|a\|^2$,

$$\sup_{\theta,u} \mathbb{E} |W_{\theta,u} - W_{\theta,\hat{u}}| \le 2M\,\eta. \tag{D.22}$$

Next, we turn to variations in $\theta$ with $u$ fixed (and the corresponding $\hat{u}$ fixed as well). We bound

$$W_{\theta,\hat{u}}(x) - W_{\hat{\theta},\hat{u}}(x) = \frac{Z_{\theta,\hat{u}}(x)}{\|\theta\|} - \frac{Z_{\hat{\theta},\hat{u}}(x)}{\|\hat{\theta}\|} = \underbrace{\frac{Z_{\theta,\hat{u}}(x) - Z_{\hat{\theta},\hat{u}}(x)}{\|\theta\|}}_{(A)} + \underbrace{Z_{\hat{\theta},\hat{u}}(x)\Big(\frac{1}{\|\theta\|} - \frac{1}{\|\hat{\theta}\|}\Big)}_{(B)}.$$

Term (B). By (D.11), $\|\theta\| \in [\|\hat{\theta}\|, (1+\varepsilon)\|\hat{\theta}\|]$, hence $\left|\frac{1}{\|\theta\|} - \frac{1}{\|\hat{\theta}\|}\right| \le \frac{\varepsilon}{\|\hat{\theta}\|}$. Using (D.19) with $\theta = \hat{\theta}$ (so $|Z_{\hat{\theta},\hat{u}}|/\|\hat{\theta}\| = |W_{\hat{\theta},\hat{u}}|$), we get

$$\left| Z_{\hat{\theta},\hat{u}}(x)\Big(\frac{1}{\|\theta\|} - \frac{1}{\|\hat{\theta}\|}\Big)\right| \le \varepsilon \left|W_{\hat{\theta},\hat{u}}(x)\right| \le C\,\varepsilon\,M\,\left|\Big\langle x, \hat{\theta}/\|\hat{\theta}\|\Big\rangle\right| |\langle \hat{u}, x \rangle|.$$

Thus, again using Cauchy-Schwarz with the identity $\frac{1}{n} \sum_{i=1}^{n} x_i x_i^T \preceq 2I$ which holds with high probability we have

$$\frac{1}{n} \sum_{i=1}^{n} \left| Z_{\hat{\theta},\hat{u}}(x_i)\Big(\frac{1}{\|\theta\|} - \frac{1}{\|\hat{\theta}\|}\Big)\right| \le C\varepsilon M \frac{1}{n} \sum_{i=1}^{n} \left|\Big\langle x_i, \hat{\theta}/\|\hat{\theta}\|\Big\rangle\right| |\langle \hat{u}, x_i \rangle|$$

$$\le C\varepsilon M \sqrt{\frac{1}{n} \sum_{i=1}^{n} \left|\Big\langle x_i, \hat{\theta}/\|\hat{\theta}\|\Big\rangle\right|^2} \sqrt{\frac{1}{n} \sum_{i=1}^{n} |\langle \hat{u}, x_i \rangle|^2}$$

$$\le 2C\varepsilon M.$$

Similarly,

$$\mathbb{E}\left[\left| Z_{\hat{\theta},\hat{u}}(x)\Big(\frac{1}{\|\theta\|} - \frac{1}{\|\hat{\theta}\|}\Big)\right|\right] \le C\varepsilon M.$$

Term (A). Let us write $E_\theta := \{|\langle x, \theta \rangle| \le \sqrt{M}\}$ and $E_{\hat{\theta}} := \{\left|\Big\langle x, \hat{\theta}\Big\rangle\right| \le \sqrt{M}\}$. Then, we have

$$\frac{\left|Z_{\theta,\hat{u}}(x) - Z_{\hat{\theta},\hat{u}}(x)\right|}{\|\theta\|} = 2\frac{|\langle x, \hat{u} \rangle|}{\|\theta\|} \left|(\sigma(\langle x, \theta \rangle) - \sigma(\langle x, \theta^\star \rangle)) \langle x, \theta \rangle \mathbf{1}\{E_\theta\} - \Big(\sigma(\langle x, \hat{\theta} \rangle) - \sigma(\langle x, \theta^\star \rangle)\Big) \langle x, \hat{\theta} \rangle \mathbf{1}\{E_{\hat{\theta}}\}\right|$$

$$\overset{(a)}{\le} 2\frac{|\langle x, \hat{u} \rangle|}{\|\theta\|} \left|\sigma(\langle x, \theta \rangle) - \sigma(\langle x, \hat{\theta} \rangle)\right| |\langle x, \theta \rangle \mathbf{1}\{E_\theta\}|$$

$$+ 2\frac{|\langle x, \hat{u} \rangle|}{\|\theta\|} \left|\sigma(\langle x, \hat{\theta} \rangle) - \sigma(\langle x, \theta^\star \rangle)\right| \left|\langle x, \theta \rangle \mathbf{1}\{E_\theta\} - \langle x, \hat{\theta} \rangle \mathbf{1}\{A_{\hat{\theta}}\}\right|$$

$$\overset{(b)}{\le} 4M |\langle x, \hat{u} \rangle| \left|\Big\langle x, (\theta - \hat{\theta})/\|\theta\|\Big\rangle\right| + 2M \frac{|\langle x, \hat{u} \rangle|}{\|\theta\|} \left|\langle x, \theta \rangle \mathbf{1}\{E_\theta\} - \langle x, \hat{\theta} \rangle \mathbf{1}\{E_{\hat{\theta}}\}\right|$$

$$\overset{(c)}{\le} 6M |\langle x, \hat{u} \rangle| \cdot \left|\Big\langle x, (\theta - \hat{\theta})/\|\theta\|\Big\rangle\right| + 2M^{3/2} |\langle x, \hat{u} \rangle| \cdot \frac{\mathbf{1}\{E_\theta \Delta E_{\hat{\theta}}\}}{\|\theta\|}. \tag{D.23}$$

Here, (a) follows from using $|ab - a'b'| \le |a - a'| |b| + |a'| |b - b'|$ with $a = \sigma(\langle x, \theta \rangle) - \sigma(\langle x, \theta^\star \rangle)$ and $b = \langle x, \theta \rangle \mathbf{1}\{E_\theta\}$, (b) from the fact that $\sigma$ is $2\sqrt{M}$ Lipschitz, $|\langle x, \theta \rangle \mathbf{1}\{E_\theta\}| \le \sqrt{M}$ and $\left|\sigma(\langle x, \hat{\theta} \rangle) - \sigma(\langle x, \theta^\star \rangle)\right| \le M$, and (c) follows from $\left|z\,\mathbf{1}\{|z| \le \sqrt{M}\} - z'\,\mathbf{1}\{|z'| \le \sqrt{M}\}\right| \le |z - z'| + \sqrt{M}\,\mathbf{1}\{\{|z| \le \sqrt{M}\} \Delta \{|z'| \le \sqrt{M}\}\}$ with $z = \langle x, \theta \rangle$, $z' = \langle x, \theta \rangle$ and $E_1 \Delta E_2$ denoting the symmetric difference of the sets $E_1, E_2$.

To control the first term in the above inequality we again use Cauchy-Schwarz with the identity $\frac{1}{n}\sum_{i=1}^{n} x_i x_i^T \preceq 2I$ which holds with high probability to conclude that

$$\frac{1}{n}\sum_{i=1}^{n} |\langle \hat{u}, x_i\rangle| \left|\left\langle x_i, (\theta - \hat{\theta})/\|\theta\|\right\rangle\right| \leq \sqrt{\frac{1}{n}\sum_{i=1}^{n}\langle u, x_i\rangle^2}\sqrt{\frac{1}{n}\sum_{i=1}^{n}\left\langle x_i, (\theta - \hat{\theta})/\|\theta\|\right\rangle^2} \leq 2\frac{\|\theta - \hat{\theta}\|}{\|\theta\|} \leq 4\varepsilon,$$

using (D.11). Furthermore, using Cauchy-Schwarz readily gives

$$\mathbb{E}\left(|\langle \hat{u}, x\rangle|\left|\left\langle x, (\theta - \hat{\theta})/\|\theta\|\right\rangle\right|\right) \leq 2\varepsilon.$$

Hence, the first term contributes $\leq CM\varepsilon$ to both $T_2$ and $T_3$.

It remains to bound the kink term involving $\mathbf{1}\{E_\theta \Delta E_{\hat{\theta}}\}/\|\theta\|$. If $E_\theta \Delta E_{\hat{\theta}}$ occurs, then necessarily $\left|\,|\hat{z}| - \sqrt{M}\,\right| \leq |z - \hat{z}| = \left|\left\langle x, \theta - \hat{\theta}\right\rangle\right|$. Thus for any $\tau \in (0, \sqrt{M}]$,

$$\mathbf{1}\{E_\theta \Delta E_{\hat{\theta}}\} \leq \mathbf{1}\{\left|\,|\hat{z}| - \sqrt{M}\,\right| \leq \tau\} + \mathbf{1}\{\left|\left\langle x, \theta - \hat{\theta}\right\rangle\right| \geq \tau\}.$$

Therefore, by using Cauchy-Schwarz with the identity $\frac{1}{n}\sum_{i=1}^{n} x_i x_i^T \preceq 2I$ which holds with high probability we have

$$\frac{1}{n}\sum_{i=1}^{n}|\langle \hat{u}, x_i\rangle|\frac{\mathbf{1}\{E_\theta(x_i)\Delta E_{\hat{\theta}}(x_i)\}}{\|\theta\|}$$

$$\leq \frac{\sqrt{\frac{1}{n}\sum_{i=1}^{n}\langle \hat{u}, x_i\rangle^2}}{\|\theta\|}\left(\sqrt{\frac{1}{n}\sum_{i=1}^{n}\mathbf{1}\{\left|\,\left|\left\langle x_i, \hat{\theta}\right\rangle\right| - \sqrt{M}\,\right| \leq \tau\}} + \sqrt{\frac{1}{n}\sum_{i=1}^{n}\mathbf{1}\{\left|\left\langle x_i, \theta - \hat{\theta}\right\rangle\right| \geq \tau\}}\right)$$

$$\leq \frac{\sqrt{2}}{\|\theta\|}\left(\sqrt{\frac{1}{n}\sum_{i=1}^{n}\mathbf{1}\{\left|\,\left|\left\langle x_i, \hat{\theta}\right\rangle\right| - \sqrt{M}\,\right| \leq \tau\}} + \sqrt{\frac{1}{n}\sum_{i=1}^{n}\frac{\left\langle x_i, \theta - \hat{\theta}\right\rangle^2}{\tau^2}}\right)$$

$$\leq \frac{\sqrt{2}}{\|\theta\|}\left(\sqrt{\frac{1}{n}\sum_{i=1}^{n}\mathbf{1}\{\left|\,\left|\left\langle x_i, \hat{\theta}\right\rangle\right| - \sqrt{M}\,\right| \leq \tau\}} + \frac{\sqrt{2}\|\theta - \hat{\theta}\|}{\tau}\right)$$

$$\leq \frac{\sqrt{2}}{\|\theta\|}\sqrt{\frac{1}{n}\sum_{i=1}^{n}\mathbf{1}\{\left|\,\left|\left\langle x_i, \hat{\theta}\right\rangle\right| - \sqrt{M}\,\right| \leq \tau\}} + 4\frac{\varepsilon}{\tau},$$

using $\|\theta - \hat{\theta}\| \leq 2\varepsilon\|\theta\|$.

Now we control the slab probability uniformly over $\hat{\theta} \in \mathcal{N}_\varepsilon$. By Lemma D.13 with $\mathcal{T} = \mathcal{N}_\varepsilon$, with probability at least $1 - 2|\mathcal{N}_\varepsilon|\exp(-cn\tau/\sqrt{M})$ we have

$$\sup_{\hat{\theta}\in\mathcal{N}_\varepsilon}\frac{1}{n}\sum_{i=1}^{n}\mathbf{1}\{\{\left|\,|\left\langle x_i, \hat{\theta}\right\rangle| - \sqrt{M}\,\right| \leq \tau\}\} \leq \frac{\tau\|\hat{\theta}\|^2}{M^{3/2}}.$$

On this event,

$$\sup_{\theta, u}\frac{1}{\|\theta\|}\sqrt{\frac{1}{n}\sum_{i=1}^{n}\mathbf{1}\{\left|\,\left|\left\langle x_i, \hat{\theta}\right\rangle\right| - \sqrt{M}\,\right| \leq \tau\}} \leq \sup_{\hat{\theta}\in\mathcal{N}_\varepsilon}\frac{1}{\|\hat{\theta}\|}\sqrt{\frac{1}{n}\sum_{i=1}^{n}\mathbf{1}\{\left|\,\left|\left\langle x_i, \hat{\theta}\right\rangle\right| - \sqrt{M}\,\right| \leq \tau\}} \leq C\frac{\sqrt{\tau}}{M^{3/4}},$$

which implies that

$$\sup_{\theta, u}\frac{1}{n}\sum_{i=1}^{n}|\langle \hat{u}, x_i\rangle|\frac{\mathbf{1}\{E_\theta(x_i)\Delta E_{\hat{\theta}}(x_i)\}}{\|\theta\|} \leq C\frac{\sqrt{\tau}}{M^{3/4}} + C\frac{\varepsilon}{\tau}. \tag{D.24}$$

The population bounds are simpler: following the same passages and applying Lemma D.12 gives

$$\sup_{\theta,u} \mathbb{E}\Big(|\langle \hat{u}, x\rangle| \frac{\mathbf{1}\{E_\theta \Delta E_{\hat{\theta}}\}}{\|\theta\|}\Big) \le C\frac{\sqrt{\tau}}{M^{3/4}} + C\frac{\varepsilon}{\tau}. \tag{D.25}$$

Combining (D.21), (D.22), (D.23), (D.24), (D.25) and adding term (B), we have that on the good events

$$T_2 + T_3 \le CM\eta + CM\varepsilon + CM^{3/4}\sqrt{\tau} + CM^{3/2}\frac{\varepsilon}{\tau}.$$

Let us now pick

$$\eta := c_1 \frac{\epsilon}{M}, \qquad \tau := \sqrt{M}\,\varepsilon^{2/3}, \qquad \varepsilon := c_2 \Big(\frac{\epsilon}{M}\Big)^3,$$

with small enough numerical constants $c_1, c_2$ so that $T_2 + T_3 \le \epsilon/3$. Here, the choice of $\tau = \sqrt{M}\varepsilon^{2/3}$ balances the last two terms $M^{3/4}\sqrt{\tau} = M\varepsilon^{1/3}$ and $M^{3/2}\varepsilon/\tau = M\varepsilon^{1/3}$. Let us gather all the high probability events:

- The failure probability of the event $\frac{1}{n}\sum_{i=1}^n x_i x_i^T \preceq 2I$ is $\le 2e^{-cd}$ if $n \ge Cd$ for a sufficiently large $C$.

- $T_1 \le \epsilon/3$ from (D.20) with failure probability $2e^{-cd}$ as long as

$$n \ \ge \ C\max\Big(\frac{M^2}{\epsilon^2}, \frac{M}{\epsilon}\Big)\Big(d\log\frac{3}{\varepsilon} + d\log\frac{3}{\eta} + \log(J+1)\Big),$$

  which is equivalent to

$$n \ \ge \ C\max\Big(\frac{M^2}{\epsilon^2}, \frac{M}{\epsilon}\Big)\Big(d\log\Big(\frac{M}{\epsilon}\Big) + \log(J+1)\Big).$$

- The slab-empirical event from Lemma D.13 with $\mathcal{T} = \mathcal{N}_\varepsilon$, which holds with probability at least

$$1 - 2|\mathcal{N}_\varepsilon|\exp(-cn\tau/\sqrt{M}) \ge 1 - 2e^{-cd}.$$

Since $\tau/\sqrt{M} = \varepsilon^{2/3} = c_2^{2/3}\epsilon^2/M^2$, this requires

$$n \ \ge \ C\frac{M^2}{\epsilon^2}\Big(d\log\frac{3}{\varepsilon} + \log(J+1)\Big).$$

These requirements are all of the form

$$n \ge C\frac{M^2}{\epsilon^2}\Big(d\log\Big(\frac{M}{\epsilon}\Big) + \log(J+1)\Big),$$

where we recall that $J$ is given by (D.12). A union bound gives failure probability $\le 6e^{-cd}$, and then $T_1 + T_2 + T_3 \le \epsilon$, proving (D.17) and concluding the argument.

$\square$

### D.3. Norm growth during the first phase

**Proposition D.14.** *Let $M$ be a large enough constant (independent of $n, d$), and $C, c > 0$ be constants (independent of $n, d, M$). Consider the truncated quadratic activation in (4.4) and let $\theta_t$ be obtained from the gradient descent iteration in (4.2), with learning rate $\eta \le 1/(40\sqrt{2}M^{3/2})$ and initialization norm $\|\theta_0\| \in [r, 1/4]$. Assume that*

$$n \ge CM^2\Big(d\log M + \log\Big(\log\Big(\frac{10}{r}\Big) + 2\Big)\Big). \tag{D.26}$$

*Let $t^*$ be the smallest $t$ such that $\|\theta_t\| \ge 1/4$. Then, with probability at least $1 - Ce^{-cd}$, the following results hold.*

1. **Upper bound on $t^*$.** We have that

$$t^* \leq \left\lceil \frac{2\log\left(\frac{1}{4\|\theta_0\|}\right)}{\log\left(1 + \frac{19}{25}\eta\right)} \right\rceil. \tag{D.27}$$

2. **Monotonic norm until $t^*$.** We have that $\|\theta_t\| > \|\theta_{t-1}\|$ for $t \leq t^*$.

3. **Norm never below $t^*$ again.** We have that $\|\theta_t\| \geq 1/4$ for all $t \geq t^*$.

4. **Upper bound on the norm.** We have that $\|\theta_t\| \leq 10$ for all $t \geq 0$.

*Proof.* We start by showing that, uniformly over all $\theta$ such that $\|\theta\| \in [r, 1/2]$,

$$-\langle \widehat{\mathcal{G}}(\theta), \theta \rangle \geq \frac{19}{50}\|\theta\|^2. \tag{D.28}$$

To that aim, we write

$$-\langle \widehat{\mathcal{G}}(\theta), \theta \rangle \geq -\langle \mathcal{G}(\theta), \theta \rangle - \|\widehat{\mathcal{G}}(\theta) - \mathcal{G}(\theta)\|\|\theta\| \geq -\langle \mathcal{G}(\theta), \theta \rangle - \frac{1}{50}\|\theta\|^2, \tag{D.29}$$

where the second inequality holds with probability at least $1 - 6e^{-cd}$ uniformly over all $\theta$ such that $\|\theta\| \in [r, 1/2]$ by Lemma D.8. To lower bound $\langle \mathcal{G}(\theta), \theta \rangle$, we proceed as follows:

$$\begin{aligned}
-\langle \mathcal{G}(\theta), \theta \rangle &= -\mathbb{E}\left[(\sigma(\langle x, \theta \rangle) - \sigma(\langle x, \theta^\star \rangle))\sigma'(\langle x, \theta \rangle)\langle x, \theta \rangle\right] \\
&= -\mathbb{E}\left[(\sigma(z) - \sigma(z^\star))\sigma'(z)z\right] \\
&= 2\left(\mathbb{E}\left[\sigma(z^\star)z^2\mathbf{1}(|z| \leq \sqrt{M})\right] - \mathbb{E}\left[z^4\mathbf{1}(|z| \leq \sqrt{M})\right]\right),
\end{aligned} \tag{D.30}$$

where we have defined $z = \langle x, \theta \rangle$ and $z^\star = \langle x, \theta^\star \rangle$. Let $\bar{r} = \|\theta\|$ and $\rho = \langle \theta, \theta^\star \rangle / \|\theta\|$. Then, as $x$ is a standard Gaussian vector, we have

$$z^\star = V, \qquad z = \bar{r}(\rho V + \sqrt{1 - \rho^2}W), \tag{D.31}$$

with $V, W$ i.i.d. standard Gaussian. We lower bound the first term in the RHS of (D.30) as

$$\begin{aligned}
\mathbb{E}\left[\sigma(z^\star)z^2\mathbf{1}(|z| \leq \sqrt{M})\right] &= \mathbb{E}\left[\sigma(z^\star)z^2\right] - \mathbb{E}\left[\sigma(z^\star)z^2\mathbf{1}(|z| > \sqrt{M})\right] \\
&\geq \mathbb{E}\left[\sigma(z^\star)z^2\right] - M\mathbb{E}\left[z^2\mathbf{1}(|z| > \sqrt{M})\right] \\
&\geq \mathbb{E}\left[(z^\star)^2z^2\right] - \mathbb{E}\left[(z^\star)^2z^2\mathbf{1}(|z^\star| > \sqrt{M})\right] - M\mathbb{E}\left[z^2\mathbf{1}(|z| > \sqrt{M})\right].
\end{aligned} \tag{D.32}$$

Next, we compute the first term in the RHS of (D.32) explicitly:

$$\mathbb{E}\left[(z^\star)^2z^2\right] = \mathbb{E}\left[V^2\bar{r}^2(\rho V + \sqrt{1 - \rho^2}W)^2\right] = \bar{r}^2(1 + 2\rho^2). \tag{D.33}$$

For the second term in the RHS of (D.32), we have

$$\begin{aligned}
\mathbb{E}\left[(z^\star)^2z^2\mathbf{1}(|z^\star| > \sqrt{M})\right] &= \bar{r}^2\left(\rho^2\mathbb{E}\left[V^4\mathbf{1}(|V| > \sqrt{M})\right] + (1 - \rho^2)\mathbb{E}\left[V^2\mathbf{1}(|V| > \sqrt{M})\right]\right) \\
&\leq \bar{r}^2\left(\frac{1}{250}\rho^2 + \frac{1}{5000}(1 - \rho^2)\right),
\end{aligned} \tag{D.34}$$

where we have used that $E\left[V^m\mathbf{1}(|V| > \sqrt{M})\right]$ ($m \in \{2, 4\}$) is decreasing in $M$ and $M \geq 20$. For the third term in the RHS of (D.32), note that $V' := \rho V + \sqrt{1 - \rho^2}W$ is standard Gaussian. Thus, we have

$$\begin{aligned}
M\mathbb{E}\left[z^2\mathbf{1}(|z| > \sqrt{M})\right] &= \bar{r}^2 M\mathbb{E}\left[(V')^2\mathbf{1}(|V'| > \sqrt{M}/\bar{r})\right] \\
&\leq \bar{r}^2 M\mathbb{E}\left[(V')^2\mathbf{1}(|V'| > 2\sqrt{M})\right] \\
&\leq 10^{-6}\bar{r}^2.
\end{aligned} \tag{D.35}$$

Here, in the second line we used that $E\left[(V')^2\mathbf{1}(|V'| > a)\right]$ is decreasing in $a$ and that $\bar{r} = \|\theta\| \leq 1/2$; in the third line, we used that the expression is decreasing in $M$ and $M$ is large enough. Finally, we upper bound the second term in the RHS of (D.30) as

$$\mathbb{E}\left[z^4\mathbf{1}(|z| \leq \sqrt{M})\right] \leq \mathbb{E}\left[z^4\right] = 3\bar{r}^4 \leq \frac{3}{4}\bar{r}^2, \tag{D.36}$$

where in the last step we use that $\bar{r} = \|\theta\| \leq 1/2$. By plugging these bounds into (D.29), we obtain

$$-\langle\widehat{\mathcal{G}}(\theta), \theta\rangle \geq \frac{2}{5}\|\theta\|^2 - \frac{1}{50}\|\theta\|^2, \tag{D.37}$$

which gives the claim in (D.28).

Now, if $r \leq \|\theta_t\| \leq 1/4 \leq 1/2$, then

$$\begin{aligned}
\|\theta_{t+1}\|^2 = \langle\theta_t - \eta\widehat{\mathcal{G}}(\theta_t), \theta_t - \eta\widehat{\mathcal{G}}(\theta_t)\rangle &= \|\theta_t\|^2 - 2\eta\langle\theta_t, \widehat{\mathcal{G}}(\theta_t)\rangle + \eta^2\|\widehat{\mathcal{G}}(\theta_t)\|^2 \\
&\geq \|\theta_t\|^2 - 2\eta\langle\theta_t, \widehat{\mathcal{G}}(\theta_t)\rangle \\
&\geq \left(1 + \frac{19}{25}\eta\right)\|\theta_t\|^2.
\end{aligned} \tag{D.38}$$

Thus, as $r \leq \|\theta_0\| \leq 1/4$, $\|\theta_t\|$ monotonically increases with $t$ (in a strict way), until it becomes $\geq 1/4$ for some $t^*$ satisfying (D.27). This proves the first two claims.

Next, we show that $\|\theta_t\| \geq 1/4$ for all $t \geq t^*$. Suppose, by contradiction, that this is not the case and let $\bar{t}$ be the smallest integer larger than $t^*$ such that $\|\theta_{\bar{t}}\| < 1/4$. Note that

$$\begin{aligned}
\sup_{\theta\in\mathbb{R}^d}\|\widehat{\mathcal{G}}(\theta)\| &= \sup_{\theta\in\mathbb{R}^d}\sup_{u\in\mathcal{S}^{d-1}}\langle u, \widehat{\mathcal{G}}(\theta)\rangle \\
&= \sup_{\theta\in\mathbb{R}^d}\sup_{u\in\mathcal{S}^{d-1}}\frac{1}{n}\sum_{i=1}^n\left(\sigma(\langle x_i, \theta\rangle) - \sigma(\langle x_i, \theta^\star\rangle)\right)\sigma'(\langle x_i, \theta\rangle)\langle x_i, u\rangle \\
&= \sqrt{\sup_{\theta\in\mathbb{R}^d}\frac{1}{n}\sum_{i=1}^n(\sigma(\langle x_i, \theta\rangle) - \sigma(\langle x_i, \theta^\star\rangle))^2(\sigma'(\langle x_i, \theta\rangle))^2}\sqrt{\sup_{u\in\mathcal{S}^{d-1}}\frac{1}{n}\sum_{i=1}^n\langle x_i, u\rangle^2} \\
&\leq 2\sqrt{2}M^{3/2},
\end{aligned} \tag{D.39}$$

where the third line follows from Cauchy-Schwarz and the last inequality uses that $(\sigma(\langle x_i, \theta\rangle) - \sigma(\langle x_i, \theta^\star\rangle))^2 \leq M^2$, $(\sigma'(\langle x_i, \theta\rangle))^2 \leq 4M$ for all $\theta \in \mathbb{R}^d$ and that $\frac{1}{n}\sum_{i=1}^n x_i x_i^\top \preceq 2I$ with probability at least $1 - 2e^{-cd}$ when $n$ is lower bounded as in (D.26). Let us express $\|\theta_{\bar{t}}\|$ as

$$\begin{aligned}
\|\theta_{\bar{t}}\|^2 &= \|\theta_{\bar{t}-1}\|^2 - 2\eta\langle\theta_{\bar{t}-1}, \widehat{\mathcal{G}}(\theta_{\bar{t}-1})\rangle + \eta^2\|\widehat{\mathcal{G}}(\theta_{\bar{t}-1})\|^2 \\
&\geq \|\theta_{\bar{t}-1}\|^2 - 2\eta\|\theta_{\bar{t}-1}\|\|\widehat{\mathcal{G}}(\theta_{\bar{t}-1})\| \\
&\geq \|\theta_{\bar{t}-1}\|^2 - \frac{1}{10}\|\theta_{\bar{t}-1}\|,
\end{aligned} \tag{D.40}$$

where the third line uses (D.39) and that $\eta \leq 1/(40\sqrt{2}M^{3/2})$. As $\|\theta_{\bar{t}}\| < 1/4$, we conclude that

$$\|\theta_{\bar{t}-1}\|^2 - \frac{1}{10}\|\theta_{\bar{t}-1}\| - \frac{1}{16} < 0,$$

which implies that

$$\|\theta_{\bar{t}-1}\| < \frac{\frac{1}{10} + \sqrt{\frac{1}{100} + \frac{1}{4}}}{2} \leq \frac{1}{2}.$$

However, as $\|\theta_{\bar{t}-1}\| \leq 1/2$, either $\|\theta_{\bar{t}-1}\|$ is in fact smaller than $r$ or (D.38) holds with $t = \bar{t} - 1$. In both cases, $\|\theta_{\bar{t}}\| \geq \|\theta_{\bar{t}-1}\|$. Thus, $\|\theta_{\bar{t}-1}\| < 1/4$ contradicting the minimality of $\bar{t}$. This proves the third claim.

It remains to show that $\|\theta_t\| \le 10$ for all $t$. To that aim, we start by showing that, uniformly over all $\theta$ such that $\|\theta\| \in [r, 10]$,

$$-\langle \widehat{\mathcal{G}}(\theta), \theta \rangle \le \left(6 + \frac{1}{50}\right) \|\theta\|^2 - \frac{9}{5}\|\theta\|^4. \tag{D.41}$$

Note that

$$-\langle \widehat{\mathcal{G}}(\theta), \theta \rangle \le -\langle \mathcal{G}(\theta), \theta \rangle + \|\widehat{\mathcal{G}}(\theta) - \mathcal{G}(\theta)\|\|\theta\| \le -\langle \mathcal{G}(\theta), \theta \rangle + \frac{1}{50}\|\theta\|^2, \tag{D.42}$$

where the second inequality holds with probability at least $1 - 6e^{-cd}$ uniformly over all $\theta$ such that $\|\theta\| \in [r, 10]$ by Lemma D.8. For the term $-\langle \mathcal{G}(\theta), \theta \rangle$, we use (D.30) and then obtain the following chain of inequalities:

$$\begin{aligned}
-\langle \mathcal{G}(\theta), \theta \rangle &= 2\left(\mathbb{E}\left[\sigma(z^\star)z^2 \mathbf{1}(|z| \le \sqrt{M})\right] - \mathbb{E}\left[z^4 \mathbf{1}(|z| \le \sqrt{M})\right]\right) \\
&\le 2\left(\mathbb{E}\left[(z^\star)^2 z^2\right] - \mathbb{E}\left[z^4\right] + \mathbb{E}\left[z^4 \mathbf{1}(|z| > \sqrt{M})\right]\right) \\
&= 2\left(\bar{r}^2(3\rho^2 + (1 - \rho^2)) - \bar{r}^4 + \mathbb{E}\left[z^4 \mathbf{1}(|z| > \sqrt{M})\right]\right) \\
&\le 2\left(3\bar{r}^2 - \bar{r}^4 + \mathbb{E}\left[z^4 \mathbf{1}(|z| > \sqrt{M})\right]\right).
\end{aligned} \tag{D.43}$$

Furthermore, we have

$$\begin{aligned}
\mathbb{E}\left[z^4 \mathbf{1}(|z| > \sqrt{M})\right] &= \bar{r}^4 \mathbb{E}\left[(V')^4 \mathbf{1}(|V'| > \sqrt{M}/\bar{r})\right] \\
&= \bar{r}^4 \mathbb{E}\left[(V')^4 \mathbf{1}(|V'| > \sqrt{M}/10)\right] \\
&\le \frac{1}{10}\bar{r}^4.
\end{aligned} \tag{D.44}$$

Here, in the second line we used that $E\left[(V')^4 \mathbf{1}(|V'| > a)\right]$ is decreasing in $a$ and that $\bar{r} = \|\theta\| \le 10$; in the third line, we used that the expression is decreasing in $M$ and $M$ is large enough. Combining (D.42), (D.43) and (D.44) gives (D.41).

Armed with this uniform bound, we now show that $\|\theta_t\| \le 10$ for all $t$. Suppose, by contradiction, that this is not the case and let $\bar{t}$ be the smallest integer $\ge 1$ such that $\|\theta_{\bar{t}}\| > 10$. The minimality of $\bar{t}$ readily implies that $\|\theta_{\bar{t}-1}\| \le 10$. Thus, we can apply the uniform bound we just proved to $\theta = \theta_{\bar{t}-1}$ and obtain

$$\begin{aligned}
\|\theta_{\bar{t}}\|^2 &= \|\theta_{\bar{t}-1}\|^2 - 2\eta\langle \theta_{\bar{t}-1}, \widehat{\mathcal{G}}(\theta_{\bar{t}-1})\rangle + \eta^2\|\widehat{\mathcal{G}}(\theta_{\bar{t}-1})\|^2 \\
&\le \|\theta_{\bar{t}-1}\|^2 + \left(6 + \frac{1}{50}\right)\|\theta_{\bar{t}-1}\|^2 - \frac{9}{5}\|\theta_{\bar{t}-1}\|^4 + \eta^2\|\widehat{\mathcal{G}}(\theta_{\bar{t}-1})\|^2 \\
&\le \left(7 + \frac{1}{50}\right)\|\theta_{\bar{t}-1}\|^2 - \frac{9}{5}\|\theta_{\bar{t}-1}\|^4 + \frac{1}{100},
\end{aligned} \tag{D.45}$$

where in the last step we use (D.39) and that $\eta \le 1/(40\sqrt{2}M^{3/2})$. Now, the RHS of (D.45) is $< 100$ for any $\|\theta_{\bar{t}-1}\|$, which gives the desired contradiction and concludes the proof. $\qquad\square$

## D.4. Angle does not decrease

**Proposition D.15.** *Fix any $\varphi \in (0, \pi/4)$. Let $M$ be a large enough constant (independent of $n, d$), and $C, c > 0$ be constants (independent of $n, d, M$, but possibly dependent on $\varphi$). Consider the truncated quadratic activation in (4.4). Let $\theta_t$ be obtained from the gradient descent iteration in (4.2), with learning rate $\eta \le 1/(40\sqrt{2}M^{3/2})$ and initialization norm $\|\theta_0\| \in [r, 1/4]$. Assume that (D.26) holds. Let $t^*$ be the smallest $t$ such that $\angle(\theta_t, \theta^\star) \le \varphi$. Then, with probability at least $1 - Ce^{-cd}$, $\angle(\theta_t, \theta^\star) \le \varphi$ for all $t \ge t^*$.*

We start by noting that there exist scalars $A(\theta, \theta^\star), B(\theta, \theta^\star)$ such that

$$\mathcal{G}(\theta) = A(\theta, \theta^\star)\,\theta - B(\theta, \theta^\star)\,\theta^\star. \tag{D.46}$$

**Lemma D.16.** *For every $\theta \in \mathbb{R}^d$,*

$$B(\theta, \theta^\star) = \mathbb{E}\big[\sigma'(\langle x, \theta \rangle)\sigma'(\langle x, \theta^\star \rangle)\big]. \tag{D.47}$$

*In particular, for $\sigma$ given by (4.4),*

$$B(\theta, \theta^\star) = 4\,\mathbb{E}\Big[\langle x, \theta \rangle \, \langle x, \theta^\star \rangle \, \mathbf{1}\{|\langle x, \theta \rangle| < \sqrt{M}, \, |\langle x, \theta^\star \rangle| < \sqrt{M}\}\Big]. \tag{D.48}$$

*Proof.* Define $f : \mathbb{R}^d \to \mathbb{R}$ by

$$f(x) := \left(\sigma(\langle x, \theta \rangle) - \sigma(\langle x, \theta^\star \rangle)\right)\sigma'(\langle x, \theta \rangle).$$

Then by (4.7), $\mathcal{G}(\theta) = \mathbb{E}[x\,f(x)]$. Thus, by Stein's lemma

$$\mathcal{G}(\theta) = \mathbb{E}[\nabla_x f(x)] = A(\theta, \theta^\star)\theta - \mathbb{E}\big[\sigma'(\langle x, \theta \rangle)\sigma'(\langle x, \theta^\star \rangle)\big]\theta^\star.$$

Extracting the coefficient of $\theta^\star$ gives $B$ and comparing with (D.46) yields (D.47). Substituting $\sigma'(z) = 2z\mathbf{1}\{|z| < \sqrt{M}\}$ yields (D.48). $\qquad\square$

**Lemma D.17** (Bounds on $A(\theta, \theta^\star)$ and $B(\theta, \theta^\star)$)**.** *Let $M \geq 1$ and $\phi := \angle(\theta, \theta^\star) \in [0, \pi]$ so that $\langle \theta, \theta^\star \rangle = \|\theta\| \cos\phi$. Then, the following results hold.*

1. **Uniform upper bound on $A(\theta, \theta^\star)$.** *For every $\theta \neq 0$,*

$$|A(\theta, \theta^\star)| \leq 4M + 4. \tag{D.49}$$

2. **Lower bound on $B(\theta, \theta^\star)$.** *For every $\theta$,*

$$B(\theta, \theta^\star) \;\geq\; 4\,\|\theta\|\left(\cos\phi - \sqrt{3}\,\sqrt{2\frac{\|\theta\|}{\sqrt{M}}e^{-\frac{M}{2\|\theta\|^2}} + 2e^{-\frac{M}{2}}}\right). \tag{D.50}$$

*Proof.* We use the identity

$$\langle \mathcal{G}(\theta), \theta \rangle = A(\theta, \theta^\star)\|\theta\|^2 - B(\theta, \theta^\star)\,\langle \theta, \theta^\star \rangle, \tag{D.51}$$

which follows by taking the inner product of (D.46) with $\theta$. Rearranging gives, for $\theta \neq 0$,

$$A(\theta, \theta^\star) = \frac{\langle \mathcal{G}(\theta), \theta \rangle + B(\theta, \theta^\star)\,\langle \theta, \theta^\star \rangle}{\|\theta\|^2}. \tag{D.52}$$

**Upper bound on $\langle \mathcal{G}(\theta), \theta \rangle$.** By (4.7),

$$\langle \mathcal{G}(\theta), \theta \rangle = \mathbb{E}\Big[\left(\sigma(\langle x, \theta \rangle) - \sigma(\langle x, \theta^\star \rangle)\right)\sigma'(\langle x, \theta \rangle)\,\langle x, \theta \rangle\Big].$$

As $0 \leq \sigma(z) \leq M$ for all $z$, we have

$$\left|\sigma(\langle x, \theta \rangle) - \sigma(\langle x, \theta^\star \rangle)\right| \leq M.$$

Furthermore, $\sigma'(\langle x, \theta \rangle) = 2\,\langle x, \theta \rangle\,\mathbf{1}\{|\langle x, \theta \rangle| < \sqrt{M}\}$, hence

$$\left|\sigma'(\langle x, \theta \rangle)\,\langle x, \theta \rangle\right| = 2\,\langle x, \theta \rangle^2\,\mathbf{1}\{|\langle x, \theta \rangle| < \sqrt{M}\} \leq 2\,\langle x, \theta \rangle^2,$$

which implies

$$\left|\left(\sigma(\langle x, \theta \rangle) - \sigma(\langle x, \theta^\star \rangle)\right)\sigma'(\langle x, \theta \rangle)\,\langle x, \theta \rangle\right| \leq 2M \cdot (2\,\langle x, \theta \rangle^2) = 4M\,\langle x, \theta \rangle^2.$$

Taking expectations and using $\mathbb{E}[\langle x, \theta \rangle^2] = \|\theta\|^2$ yields

$$\left|\langle \mathcal{G}(\theta), \theta \rangle\right| \leq 4M\,\|\theta\|^2. \tag{D.53}$$

**Upper bound on $|B(\theta, \theta^\star)|$.** From (D.48),

$$|B(\theta, \theta^\star)| = \left| 4 \, \mathbb{E}\left[ \langle x, \theta \rangle \, \langle x, \theta^\star \rangle \, \mathbf{1}\{| \langle x, \theta \rangle | < \sqrt{M}, \, | \langle x, \theta^\star \rangle | < \sqrt{M}\} \right] \right| \leq 4 \, \mathbb{E}| \langle x, \theta \rangle \, \langle x, \theta^\star \rangle |.$$

By Cauchy–Schwarz,

$$\mathbb{E}| \langle x, \theta \rangle \, \langle x, \theta^\star \rangle | \leq \sqrt{\mathbb{E}\big[ \langle x, \theta \rangle^2 \big] \mathbb{E}\big[ \langle x, \theta^\star \rangle^2 \big]}.$$

Now $\mathbb{E}[\langle x, \theta \rangle^2] = \|\theta\|^2$, $\mathbb{E}[\langle x, \theta^\star \rangle^2] = \|\theta^\star\|^2 = 1$. Therefore,

$$|B(\theta, \theta^\star)| \leq 4 \, \|\theta\|. \tag{D.54}$$

**Upper bound on $A(\theta, \theta^\star)$.** Using (D.52), (D.53), $| \langle \theta, \theta^\star \rangle | \leq \|\theta\|$, and (D.54),

$$|A(\theta, \theta^\star)| \leq \frac{4M\|\theta\|^2}{\|\theta\|^2} + \frac{|B(\theta, \theta^\star)| \, | \langle \theta, \theta^\star \rangle |}{\|\theta\|^2} \leq 4M + \frac{(4\|\theta\|) \cdot \|\theta\|}{\|\theta\|^2} = 4M + 4,$$

proving (D.49).

**Lower bound on $B(\theta)$.** Let

$$E := \{| \langle x, \theta \rangle | < \sqrt{M}, \, | \langle x, \theta^\star \rangle | < \sqrt{M}\}.$$

Then by (D.48),

$$B(\theta, \theta^\star) = 4 \, \mathbb{E}\left[ \langle x, \theta \rangle \, \langle x, \theta^\star \rangle \, \mathbf{1}\{E\} \right] = 4 \left( \mathbb{E}[\langle x, \theta \rangle \, \langle x, \theta^\star \rangle] - \mathbb{E}\left[ \langle x, \theta \rangle \, \langle x, \theta^\star \rangle \, \mathbf{1}\{E^c\} \right] \right).$$

Hence, using $z \geq -|z|$,

$$B(\theta) \geq 4 \left( \mathbb{E}[\langle x, \theta \rangle \, \langle x, \theta^\star \rangle] - \mathbb{E}\left[ | \langle x, \theta \rangle \, \langle x, \theta^\star \rangle | \, \mathbf{1}\{E^c\} \right] \right).$$

We have $\mathbb{E}[\langle x, \theta \rangle \, \langle x, \theta^\star \rangle] = \langle \theta, \theta^\star \rangle = \|\theta\| \cos\phi$. Also by applying Cauchy-Schwarz twice,

$$\begin{aligned}
\mathbb{E}\left[ | \langle x, \theta \rangle \, \langle x, \theta^\star \rangle | \, \mathbf{1}\{E^c\} \right] &\leq \sqrt{\mathbb{E}\big[ \langle x, \theta \rangle^2 \, \langle x, \theta^\star \rangle^2 \big]} \, \sqrt{\mathbb{P}(E^c)} \\
&\leq \sqrt[4]{\mathbb{E}\big[ \langle x, \theta \rangle^4 \big] \mathbb{E}\big[ \langle x, \theta^\star \rangle^4 \big]} \, \sqrt{\mathbb{P}(E^c)} \\
&\leq \sqrt{3} \, \|\theta\| \, \sqrt{\mathbb{P}(E^c)}.
\end{aligned} \tag{D.55}$$

Therefore

$$B(\theta, \theta^\star) \geq 4\|\theta\| \left( \cos\phi - \sqrt{3}\sqrt{\mathbb{P}(E^c)} \right). \tag{D.56}$$

Finally, by the union bound,

$$\mathbb{P}(E^c) \leq \mathbb{P}(| \langle x, \theta \rangle | \geq \sqrt{M}) + \mathbb{P}(| \langle x, \theta^\star \rangle | \geq \sqrt{M}).$$

Since $\langle x, \theta \rangle \sim \mathcal{N}(0, \|\theta\|^2)$ and $\langle x, \theta^\star \rangle \sim \mathcal{N}(0, 1)$, the standard Gaussian tail bound gives

$$\mathbb{P}(| \langle x, \theta \rangle | \geq \sqrt{M}) \leq 2 \frac{\|\theta\|}{\sqrt{M}} \exp\left( -\frac{M}{2\|\theta\|^2} \right), \qquad \mathbb{P}(| \langle x, \theta^\star \rangle | \geq \sqrt{M}) \leq 2 \exp\left( -\frac{M}{2} \right).$$

Substituting into (D.56) yields (D.50). $\qquad \square$

**Lemma D.18.** *Fix $\varphi \in (0, \pi/2)$. Let $\theta \neq 0$ satisfy $\angle(\theta, \theta^\star) \leq \varphi$, where $\|\theta^\star\| = 1$. Let the population gradient $\mathcal{G}(\theta)$ admit the decomposition*

$$\mathcal{G}(\theta) = A(\theta, \theta^\star) \, \theta - B(\theta, \theta^\star) \, \theta^\star \quad with \quad B(\theta, \theta^\star) \geq 0,$$

*and let $\widetilde{\mathcal{G}}(\theta)$ be any perturbed gradient satisfying*

$$\|\widetilde{\mathcal{G}}(\theta) - \mathcal{G}(\theta)\| \leq \Delta := B(\theta, \theta^\star) \frac{\tan\varphi}{1 + \tan\varphi}. \tag{D.57}$$

*Consider one step $\theta^+ = \theta - \eta\widetilde{\mathcal{G}}(\theta)$, where the step size $\eta > 0$ satisfies*

$$1 - \eta A(\theta, \theta^\star) \geq 0. \tag{D.58}$$

*Then, $\angle(\theta^+, \theta^\star) \leq \varphi$.*

*Proof.* Let $e := \widetilde{\mathcal{G}}(\theta) - \mathcal{G}(\theta)$, so $\|e\| \leq \Delta$. Consider the *population step* vector

$$\bar{\theta}^+ := \theta - \eta \mathcal{G}(\theta) = \theta - \eta\big(A(\theta, \theta^\star)\theta - B(\theta, \theta^\star)\theta^\star\big) = \alpha\,\theta + \eta B(\theta, \theta^\star)\theta^\star,$$

where $\alpha := 1 - \eta A(\theta, \theta^\star) \geq 0$ by (D.58). Then, we have

$$\theta^+ = \theta - \eta\widetilde{\mathcal{G}}(\theta) = \theta - \eta(\mathcal{G}(\theta) + e) = \bar{\theta}^+ - \eta e.$$

Let $\Pi_\perp := I - \theta^\star(\theta^\star)^\top$ denote the projection onto $(\theta^\star)^\perp$. For any nonzero vector $z$ with $\langle z, \theta^\star \rangle > 0$,

$$\tan(\angle(z, \theta^\star)) = \frac{\|\Pi_\perp z\|}{\langle z, \theta^\star \rangle}.$$

**Lower bound on the new parallel component.** Set $\phi := \angle(\theta, \theta^\star) \leq \varphi$. By Cauchy-Schwarz,

$$\langle \theta^+, \theta^\star \rangle = \langle \bar{\theta}^+ - \eta e, \theta^\star \rangle = \langle \bar{\theta}^+, \theta^\star \rangle - \eta \langle e, \theta^\star \rangle \geq \langle \bar{\theta}^+, \theta^\star \rangle - \eta\|e\| \geq \langle \bar{\theta}^+, \theta^\star \rangle - \eta\Delta.$$

Moreover,

$$\langle \bar{\theta}^+, \theta^\star \rangle = \langle \alpha\theta + \eta B(\theta, \theta^\star)\theta^\star, \theta^\star \rangle = \alpha \langle \theta, \theta^\star \rangle + \eta B(\theta, \theta^\star) \geq \alpha\|\theta\|\cos\phi + \eta B(\theta, \theta^\star),$$

since $\alpha \geq 0$ and $\langle \theta, \theta^\star \rangle \geq 0$ whenever $\angle(\theta, \theta^\star) \leq \varphi < \pi/2$. Hence, we have

$$\langle \theta^+, \theta^\star \rangle \geq \alpha\|\theta\|\cos\phi + \eta(B(\theta, \theta^\star) - \Delta). \tag{D.59}$$

As $\frac{\tan\varphi}{1+\tan\varphi} < 1$, $\Delta < B(\theta, \theta^\star)$, hence $\langle \theta^+, \theta^\star \rangle > 0$ and the tangent formula applies to $\theta^+$.

**Upper bound on the new orthogonal component.** Using the triangle inequality and $\|\Pi_\perp e\| \leq \|e\|$,

$$\|\Pi_\perp \theta^+\| = \|\Pi_\perp(\bar{\theta}^+ - \eta e)\| \leq \|\Pi_\perp \bar{\theta}^+\| + \eta\|\Pi_\perp e\| \leq \|\Pi_\perp \bar{\theta}^+\| + \eta\Delta.$$

Since $\Pi_\perp \theta^\star = 0$,

$$\|\Pi_\perp \bar{\theta}^+\| = \|\Pi_\perp(\alpha\theta + \eta B(\theta, \theta^\star)\theta^\star)\| = \|\alpha\,\Pi_\perp\theta\| = \alpha\,\|\Pi_\perp\theta\|.$$

We have $\|\Pi_\perp\theta\| = \|\theta\|\sin\phi$. Therefore,

$$\|\Pi_\perp \theta^+\| \leq \alpha\,\|\theta\|\sin\phi + \eta\Delta. \tag{D.60}$$

**Bound on the new angle.** Combining (D.59) and (D.60),

$$\tan(\angle(\theta^+, \theta^\star)) = \frac{\|\Pi_\perp \theta^+\|}{\langle \theta^+, \theta^\star \rangle} \leq \frac{\alpha\|\theta\|\sin\phi + \eta\Delta}{\alpha\|\theta\|\cos\phi + \eta(B(\theta, \theta^\star) - \Delta)}.$$

To ensure $\angle(\theta^+, \theta^\star) \leq \varphi$, it suffices that the right-hand side is at most $\tan\varphi$:

$$\frac{\alpha\|\theta\|\sin\phi + \eta\Delta}{\alpha\|\theta\|\cos\phi + \eta(B(\theta, \theta^\star) - \Delta)} \leq \tan\varphi.$$

Multiplying by the positive denominator and rearranging yields

$$\alpha\|\theta\|\sin\phi + \eta\Delta \leq \alpha\|\theta\|\cos\phi\tan\varphi + \eta\tan\varphi\, B(\theta, \theta^\star) - \eta\tan\varphi\,\Delta.$$

Rearranging the above the latter is equivalent to

$$\eta\Delta(1 + \tan\varphi) \leq \alpha\|\theta\|(\cos\phi\tan\varphi - \sin\phi) + \eta\tan\varphi\, B(\theta, \theta^\star)$$
$$= \alpha\|\theta\|\frac{\sin(\varphi - \phi)}{\cos\varphi} + \eta\tan\varphi\, B(\theta, \theta^\star)$$

Since $\phi \leq \varphi$ the first term is positive and a valid sufficient condition is

$$\eta\Delta(1 + \tan\varphi) \leq \eta B(\theta, \theta^\star)\tan\varphi,$$

which is satisfied by our choice of $\Delta$ in (D.57). This proves $\angle(\theta^+, \theta^\star) \leq \varphi$ and concludes the argument. $\qquad\square$

**Lemma D.19.** *Let $M \geq 1$ and $\theta \neq 0$ satisfy $\angle(\theta, \theta^\star) \leq \varphi < \pi/2$. Let $\theta^+ = \theta - \eta \widetilde{\mathcal{G}}(\theta)$, where $\eta \leq 1/(4M+4)$ and $\widetilde{\mathcal{G}}(\theta)$ is any perturbation of the population gradient $\mathcal{G}(\theta)$ in (4.7) satisfying*

$$\|\widetilde{\mathcal{G}}(\theta) - \mathcal{G}(\theta)\| \leq \left[ 4 \left( \cos\varphi - \sqrt{3} \sqrt{2 \frac{R}{\sqrt{M}} e^{-\frac{M}{2R^2}} + 2e^{-\frac{M}{2}}} \right) \right] \cdot \frac{\tan\varphi}{1+\tan\varphi} \|\theta\| \quad \text{for all } \theta \text{ s.t. } r \leq \|\theta\| \leq R. \quad \text{(D.61)}$$

*Then, one has $\angle(\theta^+, \theta^\star) \leq \varphi$.*

*Proof.* By Lemma D.17, $A(\theta, \theta^\star) \leq 4M+4$, so the assumption on the step size $\eta \leq \frac{1}{4M+4}$ implies $1 - \eta A(\theta, \theta^\star) \geq 0$. By assumption, (D.61) holds and note that

$$\left[ 4 \left( \cos\varphi - \sqrt{3} \sqrt{2 \frac{R}{\sqrt{M}} e^{-\frac{M}{2R^2}} + 2e^{-\frac{M}{2}}} \right) \right] \cdot \frac{\tan\varphi}{1+\tan\varphi} \|\theta\|$$

$$\leq \left[ 4\|\theta\| \left( \cos(\angle(\theta,\theta^\star)) - \sqrt{3} \sqrt{2 \frac{\|\theta\|}{\sqrt{M}} e^{-\frac{M}{2\|\theta\|^2}} + 2e^{-\frac{M}{2}}} \right) \right] \cdot \frac{\tan\varphi}{1+\tan\varphi} \quad \text{(D.62)}$$

$$\leq B(\theta, \theta^\star) \cdot \frac{\tan\varphi}{1+\tan\varphi} := \Delta,$$

where in the first inequality we use that $\angle(\theta, \theta^\star) \leq \varphi$ and $\|\theta\| \leq R$, and in the second inequality we use the lower bound (D.50) on $B(\theta, \theta^\star)$ from Lemma D.17. Thus, condition (D.57) holds and the result follows from Lemma D.18. □

At this point, the desired result showing that the angle does not decrease is a consequence of Lemmas D.19 and D.8.

*Proof of Proposition D.15.* We verify that the hypotheses of Lemma D.19 hold when taking $\theta = \theta_t$ and $\widetilde{\mathcal{G}}(\theta) = \widehat{\mathcal{G}}(\theta_t)$, where $\widehat{\mathcal{G}}(\theta_t)$ is the empirical gradient in (4.3). The upper bound $\eta \leq 1/(4M+4)$ required by Lemma D.19 is implied by the upper bound $\eta \leq 1/(40\sqrt{2}M^{3/2})$ for large enough $M$. By Proposition D.14, we have that $\|\theta_t\| \in [r, 10]$, with probability at least $1 - Ce^{-cd}$ uniformly for all $t \geq 0$. Hence, we can take $R = 10$. For large enough $M$,

$$\cos\varphi - \sqrt{3} \sqrt{2 \frac{R}{\sqrt{M}} e^{-\frac{M}{2R^2}} + 2e^{-\frac{M}{2}}} \geq \cos\varphi - \frac{1}{2\sqrt{2}} \geq \frac{1}{2} \cos\varphi,$$

where the last inequality holds for all $\varphi \in [0, \pi/4]$. Then, an application of Lemma D.8 with $\epsilon := 2\cos\varphi \frac{\tan\varphi}{1+\tan\varphi}$ gives that (D.61) holds, with probability at least $1 - Ce^{-cd}$. In fact, the lower bound on $n$ in (D.10) is implied by that in (D.26) (for a suitable choice of $C$ depending on $\varphi$). Then, the conclusion of Lemma D.19 holds: if $\angle(\theta_t, \theta^\star) \leq \varphi$, then $\angle(\theta_{t+1}, \theta^\star) \leq \varphi$, with probability at least $1 - Ce^{-cd}$ uniformly for all $t \geq 0$. This is equivalent to the desired claim and the proof is complete. □

## D.5. Uniform concentration of the Gram matrix of the Jacobian

For $\theta, \widetilde{\theta} \in \mathbb{R}^d$, let us define the random matrix

$$\widehat{H}(\theta, \widetilde{\theta}) := \frac{1}{n} \sum_{i=1}^n \sigma'(\langle x_i, \theta \rangle) \, \sigma'(\langle x_i, \widetilde{\theta} \rangle) \, x_i x_i^\top, \qquad H := \mathbb{E}[\widehat{H}(\theta, \widetilde{\theta})]. \quad \text{(D.63)}$$

**Lemma D.20.** *Let $\widehat{H}(\theta, \widetilde{\theta})$ be defined in (D.63) and the annulus $\Theta$ be defined in (D.9). Fix $\epsilon \in (0, 1/2)$ and let $M$ be sufficiently large. There exist constants $c, C > 0$ such that if*

$$n \geq C d \, \frac{M^2 \log^2\left(\frac{eM}{\epsilon}\right)}{\epsilon^2}, \quad \text{(D.64)}$$

*then with probability at least $1 - 3e^{-cd}$,*

$$\sup_{\theta, \widetilde{\theta} \in \Theta} \|\widehat{H}(\theta, \widetilde{\theta}) - H(\theta, \widetilde{\theta})\| \leq \epsilon. \quad \text{(D.65)}$$

### D.5.1. PRELIMINARY RESULTS

We start by providing some auxiliary probabilistic results. Recall that a real-valued random variable $X$ is said to be sub-exponential with parameters $(\sigma^2, b)$ if, for all $|\lambda| \leq \frac{1}{b}$,

$$\mathbb{E} \exp\big(\lambda(X - \mathbb{E}X)\big) \leq \exp\left(\frac{\lambda^2 \sigma^2}{2}\right). \tag{D.66}$$

For such subexponential random variables, we have the following standard refined Bernstein-type inequality, see Eq. (2.18) in (Wainwright, 2019).

**Theorem D.21** (Bernstein's-type inequality for sub-exponential sums)**.** *Let $X_1, \ldots, X_n$ be independent mean-zero random variables, where each $X_i$ is sub-exponential with parameters $(\sigma^2, b)$ in the sense of (D.66). Then, for every $s \geq 0$,*

$$\mathbb{P}\left(\frac{1}{n} \sum_{i=1}^{n} X_i \geq s\right) \leq 2 \exp\left(-\frac{1}{2} n \cdot \min\left\{\frac{s^2}{\sigma^2}, \frac{s}{b}\right\}\right). \tag{D.67}$$

**Lemma D.22.** *Let $Z \sim \mathcal{N}(0, 1)$ and $Y = (Z^2 - \tau)_+$, where $[\cdot]_+ = \max(\cdot, 0)$. Then, $Y$ is sub-exponential with parameters $(8, 128 e^{-\tau/2})$.*

*Proof.* Note that, for $t \geq 0$,

$$\mathbb{P}(Z^2 > t) \leq 2 e^{-t/2},$$

which implies that

$$\mathbb{P}(Y \geq t) \leq 2 e^{-t/2 - \tau/2}.$$

Thus, we obtain that, for any integer $k \geq 1$,

$$\begin{aligned}
\mathbb{E}[Y^k] &= \int_0^\infty k t^{k-1} \mathbb{P}(Y \geq t) dt \\
&\leq 2 e^{-\tau/2} k \int_0^\infty t^{k-1} e^{-t/2} dt \\
&\leq k! 2^{k+1} e^{-\tau/2}.
\end{aligned} \tag{D.68}$$

Let $X := Y - \mathbb{E}[Y]$. Then,

$$\mathbb{E}[|X|^k] \leq 2^{k-1} \left(\mathbb{E}[Y^k] + (\mathbb{E}[Y])^k\right) \leq 2^k \mathbb{E}[Y^k] \leq k! 2^{2k+1} e^{-\tau/2}, \tag{D.69}$$

where the second passage uses Jensen inequality.

We now claim that, for a random variable $X$ with zero mean, if there exists constants $v^2 > 0$ and $c > 0$ such that, for every integer $k \geq 2$,

$$\mathbb{E}[|X|^k] \leq \frac{k!}{2} v^2 c^{k-2}, \tag{D.70}$$

then for every $\lambda < 1/c$,

$$\log \mathbb{E}[e^{\lambda X}] \leq \frac{\lambda^2 v^2}{2(1 - c|\lambda|)}. \tag{D.71}$$

In particular, for $|\lambda| \leq 1/(2c)$, we have

$$\mathbb{E}[e^{\lambda X}] \leq e^{\lambda^2 v^2},$$

which means that $X$ is sub-exponential with parameters $b = 2c$ and $\sigma^2 = 2v^2$. Thus, from the bound on the moments in (D.69), we deduce that (D.70) holds with $c = 4$ and $v^2 = 64 e^{-\tau/2}$ and, therefore, the desired result holds with $b = 8$ and $\sigma^2 = 128 e^{-\tau/2}$.

It remains to prove the claim in (D.71). To that aim, note that

$$\mathbb{E}[e^{\lambda X}] = 1 + \lambda \mathbb{E}[X] + \sum_{k=2}^{\infty} \mathbb{E}[X^k] = 1 + \sum_{k=2}^{\infty} \mathbb{E}[X^k].$$

Upper bounding $\mathbb{E}[X^k]$ by $\mathbb{E}[|X^k|]$ and using (D.70) gives

$$\mathbb{E}[e^{\lambda X}] \leq 1 + \frac{v^2\lambda^2}{2}\sum_{k=2}^{\infty}(c|\lambda|)^{k-2} = 1 + \frac{v^2\lambda^2}{2(1 - c|\lambda|)},$$

where the last passage holds as $|\lambda| < 1/c$. Taking the $\log$ on both sides and using that $\log(1 + u) \leq u$ for all $u \geq 0$ concludes the argument. $\qquad\square$

We will use the following result on covering numbers for VC-subgraph classes.

**Theorem D.23** (Theorem 2.6.7 in (Van der Vaart & Wellner, 1996)). *Let $\mathcal{H}$ be a VC-subgraph class of real-valued functions on $\mathbb{R}^d$ with VC-subgraph dimension at most $V$ and envelope bound $|h| \leq B$ pointwise. Then there exist absolute constants $A, C > 0$ such that for every probability measure $Q$ and every $0 < \eta \leq B$,*

$$\log N(\eta, \mathcal{H}, L_2(Q)) \;\leq\; CV\log\Big(\frac{AB}{\eta}\Big).$$

We will also use the following Dudley-type bound for Rademacher averages.

**Theorem D.24** (Equation (5.48) in (Wainwright, 2019)). *There exists an absolute constant $C > 0$ such that, for any function class $\mathcal{F}$, conditionally on $x_1, \ldots, x_n$,*

$$\mathbb{E}_{\varepsilon}\Big[\sup_{f\in\mathcal{F}}\Big|\frac{1}{n}\sum_{i=1}^{n}\varepsilon_i f(x_i)\Big|\Big] \leq \frac{C}{\sqrt{n}}\int_0^{2r}\sqrt{\log N(\eta, \mathcal{F}, L_2(P_n))}\, d\eta,$$

*where $P_n$ is the empirical measure of $x_1, \ldots, x_n$ and $r := \sup_{f\in\mathcal{F}}\|f\|_{L_2(P_n)}$.*

Finally, we will use Bousquet's concentration inequality for suprema of bounded empirical processes.

**Theorem D.25** (Theorem 2.3 in (Bousquet, 2002)). *Let $\mathcal{F}$ be a class of measurable functions with $0 \leq f \leq b$. Let*

$$Z := \sup_{f\in\mathcal{F}}\sum_{i=1}^{n}\big(f(X_i) - \mathbb{E}f(X)\big), \qquad \sigma^2 := \sup_{f\in\mathcal{F}}\mathrm{Var}(f(X)).$$

*Then for all $t \geq 0$, with probability at least $1 - e^{-t}$,*

$$Z \leq \mathbb{E}Z + \sqrt{2t(n\sigma^2 + 2b\mathbb{E}Z)} + \frac{bt}{3}. \tag{D.72}$$

In particular, we will further upper bound the RHS of (D.72) as

$$\mathbb{E}Z + \sqrt{2t(n\sigma^2 + 2b\mathbb{E}Z)} + \frac{bt}{3} \leq 2\mathbb{E}Z + \sqrt{2tn\sigma^2} + \frac{4bt}{3}, \tag{D.73}$$

where we have used that $\sqrt{a + b} \leq \sqrt{a} + \sqrt{b}$.

### D.5.2. PROOF OF LEMMA D.20

Let $\mathcal{U}_{1/4}$ be a $1/4$-net of the unit sphere. Then, by Corollary 4.2.13 of (Vershynin, 2018), we have that $|\mathcal{U}_{1/4}| \leq 9^d$ and by Lemma 4.4.1 in (Vershynin, 2018), we have

$$\begin{aligned}
\Big\|\widehat{H}(\theta, \widetilde{\theta}) - H(\theta, \widetilde{\theta})\Big\| &= \sup_{u\in\mathcal{S}^{d-1}}\Big|u^\top(\widehat{H}(\theta, \widetilde{\theta}) - H(\theta, \widetilde{\theta}))u\Big| \\
&\leq \frac{1}{(1 - 1/4)^2}\max_{u\in\mathcal{U}_{1/4}}\Big|u^\top(\widehat{H}(\theta, \widetilde{\theta}) - H(\theta, \widetilde{\theta}))u\Big| \\
&\leq 2\max_{u\in\mathcal{U}_{1/4}}\Big|u^\top(\widehat{H}(\theta, \widetilde{\theta}) - H(\theta, \widetilde{\theta}))u\Big|.
\end{aligned} \tag{D.74}$$

For fixed $u \in \mathcal{U}_{1/4}$ and $\theta, \widetilde{\theta} \in \Theta$, define the scalar function

$$f_{u,\theta,\widetilde{\theta}}(x) := \sigma'(\langle x, \theta \rangle)\sigma'(\langle x, \widetilde{\theta} \rangle) \langle u, x \rangle^2.$$

Let $P_n$ be the empirical measure and $P$ the law of $x \sim \mathcal{N}(0, I_d)$. Then, we have

$$u^\top(\widehat{H}(\theta, \widetilde{\theta}) - H(\theta, \widetilde{\theta}))u = (P_n - P)f_{u,\theta,\widetilde{\theta}} := \frac{1}{n}\sum_{i=1}^n \left( f_{u,\theta,\widetilde{\theta}}(x_i) - \mathbb{E}[f_{u,\theta,\widetilde{\theta}}(x)] \right).$$

Thus, it suffices to bound

$$\max_{u \in \mathcal{U}_{1/4}} \sup_{\theta, \widetilde{\theta} \in \Theta} |(P_n - P)f_{u,\theta,\widetilde{\theta}}|.$$

To do so, we truncate the term $\langle u, x \rangle^2$. Fix $\tau \geq 1$. For $t \geq 0$, define

$$T_\tau(t) := \min\{t, \tau\}, \qquad R_\tau(t) := (t - \tau)_+.$$

For $u \in \mathcal{U}_{1/4}$, define

$$W_u(x) := \langle u, x \rangle^2, \qquad W_u^{(\tau)}(x) := T_\tau(W_u(x)), \qquad G_u^{(\tau)}(x) := R_\tau(W_u(x)).$$

Then, $W_u = W_u^{(\tau)} + G_u^{(\tau)}$, and therefore

$$f_{u,\theta,\widetilde{\theta}}(x) = f_{u,\theta,\widetilde{\theta}}^{(\tau)}(x) + r_{u,\theta,\widetilde{\theta}}^{(\tau)}(x),$$

where

$$f_{u,\theta,\widetilde{\theta}}^{(\tau)}(x) := \sigma'(\langle x, \theta \rangle)\sigma'(\langle x, \widetilde{\theta} \rangle) W_u^{(\tau)}(x), \qquad r_{u,\theta,\widetilde{\theta}}^{(\tau)}(x) := \sigma'(\langle x, \theta \rangle)\sigma'(\langle x, \widetilde{\theta} \rangle) G_u^{(\tau)}(x).$$

Thus, we have

$$\sup_{\theta, \widetilde{\theta} \in \Theta} |(P_n - P)f_{u,\theta,\widetilde{\theta}}| \leq \sup_{\theta, \widetilde{\theta} \in \Theta} |(P_n - P)f_{u,\theta,\widetilde{\theta}}^{(\tau)}| + \sup_{\theta, \widetilde{\theta} \in \Theta} |(P_n - P)r_{u,\theta,\widetilde{\theta}}^{(\tau)}|. \tag{D.75}$$

We start by bounding the second term in the RHS of (D.75). To that aim, first note that, for all $x$, $\theta$, and $\widetilde{\theta}$,

$$\left| \sigma'(\langle x, \theta \rangle)\sigma'(\langle x, \widetilde{\theta} \rangle) \right| \leq (2\sqrt{M})(2\sqrt{M}) = 4M, \tag{D.76}$$

which implies that

$$|r_{u,\theta,\widetilde{\theta}}^{(\tau)}(x)| \leq 4M\, G_u^{(\tau)}(x),$$

namely this term is automatically uniform in $\theta, \widetilde{\theta}$. Using $|(P_n - P)h| \leq P_n|h| + P|h|$ for any $h$, we have

$$\sup_{\theta, \widetilde{\theta} \in \Theta} |(P_n - P)r_{u,\theta,\widetilde{\theta}}^{(\tau)}| \leq 4M \left( P_n G_u^{(\tau)} + P G_u^{(\tau)} \right).$$

We now claim that, for any $\tau \geq 1$, there exist absolute constants $c, C > 0$ such that with probability at least $1 - 2e^{-cd}$,

$$\max_{u \in \mathcal{U}_{1/4}} \frac{1}{n}\sum_{i=1}^n G_u^{(\tau)}(x_i) \leq Ce^{-c\tau} + C\sqrt{\frac{e^{-c\tau}\, d}{n}} + C\frac{d}{n}. \tag{D.77}$$

**Proof of the claim in (D.77).** Fix $u \in \mathcal{U}_{1/4}$. Since $\langle x, u \rangle \sim \mathcal{N}(0, 1)$, we have $G_u^{(\tau)}(x) \overset{d}{=} (Z^2 - \tau)_+$, with $Z \sim \mathcal{N}(0, 1)$. By Lemma D.22, $G_u^{(\tau)}(x)$ is sub-exponential with parameters $(\sigma^2, b)$, with $\sigma^2 \leq Ce^{-c\tau}$ and $b \leq C$. Thus, an application of Theorem D.21 with $s = C\sqrt{\frac{e^{-c\tau}t}{n}} + C\frac{t}{n}$ yields for all $t \geq 1$,

$$\mathbb{P}\left( \frac{1}{n}\sum_{i=1}^n G_u^{(\tau)}(x_i) - \mathbb{E}G_u^{(\tau)}(x) \geq C\sqrt{\frac{e^{-c\tau}t}{n}} + C\frac{t}{n} \right) \leq 2e^{-t}.$$

Set $t = c_1 d$ and do a union bound over $|\mathcal{U}_{1/4}| \leq 9^d$; choosing $c_1$ large enough makes the union-bound failure probability $\leq 2e^{-cd}$ and concludes the proof. $\qquad\square$

From (D.68) with $k = 1$, we also have that

$$PG_u^{(\tau)} \leq Ce^{-c\tau},$$

which combined with (D.77) gives that

$$\sup_{u \in \mathcal{U}} \sup_{\theta, \widetilde{\theta} \in \Theta} |(P_n - P)r_{u,\theta,\widetilde{\theta}}^{(\tau)}| \leq CMe^{-c\tau} + CM\sqrt{\frac{e^{-c\tau}d}{n}} + CM\frac{d}{n}. \tag{D.78}$$

We now bound the first term in the RHS of (D.75). To that aim, fix $u \in \mathcal{U}_{1/4}$ and $\tau \geq 1$. Define the class

$$\mathcal{F}_{u,\tau} := \left\{ f_{u,\theta,\widetilde{\theta}}^{(\tau)}(\cdot) : \theta, \widetilde{\theta} \in \Theta \right\}.$$

As $0 \leq W_u^{(\tau)} \leq \tau$ and (D.76) holds, we have the uniform bound

$$\left| f_{u,\theta,\widetilde{\theta}}^{(\tau)}(x) \right| \leq 4M\tau, \qquad \text{for all } x, \theta, \widetilde{\theta}. \tag{D.79}$$

We further decompose

$$f_{u,\theta,\widetilde{\theta}}^{(\tau)}(x) = m_{u,\tau}(x)\, g_{\theta,\widetilde{\theta}}(x),$$

where $g_{\theta,\widetilde{\theta}}(x) := \langle x, \theta \rangle \langle x, \widetilde{\theta} \rangle \mathbf{1}\{|x^\top \theta| < \sqrt{M}\}\mathbf{1}\{|x^\top \widetilde{\theta}| < \sqrt{M}\}$ is the $(\theta, \widetilde{\theta})$-dependent factor and $m_{u,\tau}(x) := 4W_u^{(\tau)}(x)$ is fixed (for this choice of $u, \tau$). Note that $|m_{u,\tau}(x)| \leq 4\tau$ and $|g_\theta(x)| \leq M$. Hence, for any probability measure $Q$ on $\mathbb{R}^d$ and any $\theta, \theta', \widetilde{\theta}, \widetilde{\theta}'$,

$$\|f_{u,\theta,\widetilde{\theta}}^{(\tau)} - f_{u,\theta',\widetilde{\theta}'}^{(\tau)}\|_{L_2(Q)} \leq 4\tau \|g_{\theta,\widetilde{\theta}} - g_{\theta',\widetilde{\theta}'}\|_{L_2(Q)}. \tag{D.80}$$

Consequently, for every $\eta > 0$,

$$N(\eta, \mathcal{F}_{u,\tau}, L_2(Q)) \leq N\left(\frac{\eta}{4\tau}, \mathcal{G}, L_2(Q)\right), \qquad \mathcal{G} := \{g_{\theta,\widetilde{\theta}} : \theta, \widetilde{\theta} \in \mathbb{R}^d\}. \tag{D.81}$$

Note that the class $\mathcal{G} := \{g_{\theta,\widetilde{\theta}} : \theta, \widetilde{\theta} \in \mathbb{R}^d\}$ is VC-subgraph with VC-subgraph dimension $V \leq Cd$ for an absolute constant $C$. To prove this claim, we write out the set $G$ of subgraphs of the class $\mathcal{G}$, which is by definition

$$G := \left\{ \text{subgraph}(g_{\theta,\tilde{\theta}}) \subseteq \mathbb{R}^d \times \mathbb{R} : g_{\theta,\tilde{\theta}} \in \mathcal{G} \right\}, \text{ where } \text{subgraph}(g_{\theta,\tilde{\theta}}) = \left\{ (x, \xi) \in \mathbb{R}^d \times \mathbb{R} : \exists \xi \leq g_{\theta,\tilde{\theta}}(x) \right\}.$$

A membership test for an arbitrary $\text{subgraph}(g_{\theta,\tilde{\theta}}) \in G$ can be expressed as intersection of 5 polynomial inequalities of degree at most 2 in $x$ and $\xi$, i.e.,

$$(x, \xi) \in \text{subgraph}(g_{\theta,\tilde{\theta}}) \iff -\sqrt{M} \leq x^\top \theta \leq \sqrt{M} \wedge -\sqrt{M} \leq x^\top \theta \leq \sqrt{M} \wedge \xi \leq (x^\top \theta)(x^\top \tilde{\theta}).$$

The result of (Goldberg & Jerrum, 1993, Theorem 2.2) then implies that $\text{VC}(G) \leq Cd$, for $C$ an absolute constant, proving that $\mathcal{G}$ is VC-subgraph with VC-subgraph dimension $V \leq Cd$.

Thus, applying Theorem D.23 to $\mathcal{G}$ with $B = M$ and $V \leq Cd$, we obtain that, for all $Q$ and $0 < \eta \leq M$,

$$\log N(\eta, \mathcal{G}, L_2(Q)) \leq Cd \log\left(\frac{AM}{\eta}\right).$$

Using (D.81), we deduce that, for all $Q$ and all $0 < \eta \leq 4M\tau$,

$$\log N(\eta, \mathcal{F}_{u,\tau}, L_2(Q)) \leq Cd \log\left(\frac{A'M\tau}{\eta}\right), \tag{D.82}$$

for an absolute constant $A' > 0$.

Let $\varepsilon_1, \ldots, \varepsilon_n$ be i.i.d. Rademacher signs independent of the data. By symmetrization (see e.g. Lemma 2.3.1 in (Van der Vaart & Wellner, 1996) with $\Phi(t) = t$), we have

$$\mathbb{E}\Big[ \sup_{f \in \mathcal{F}_{u,\tau}} |(P_n - P)f| \Big] \leq 2\, \mathbb{E}\Big[ \sup_{f \in \mathcal{F}_{u,\tau}} \Big| \frac{1}{n} \sum_{i=1}^n \varepsilon_i f(x_i) \Big| \Big]. \tag{D.83}$$

Applying Theorem D.24 with $\mathcal{F} = \mathcal{F}_{u,\tau}$, $\sup_{f \in \mathcal{F}_{u,\tau}} \|f\|_\infty \leq b := 4M\tau$ from (D.79) and the entropy bound (D.82) with $Q = P_n$, we obtain (conditionally on the samples $x_1, \ldots, x_n$)

$$\mathbb{E}_\varepsilon\Big[ \sup_{f \in \mathcal{F}_{u,\tau}} \Big| \frac{1}{n} \sum_{i=1}^n \varepsilon_i f(x_i) \Big| \Big] \leq \frac{C}{\sqrt{n}} \int_0^{2b} \sqrt{d \log\Big( \frac{A'M\tau}{\eta} \Big)}\, d\eta.$$

Using the bound

$$\int_0^{2b} \sqrt{\log\Big( \frac{A'M\tau}{\eta} \Big)}\, d\eta \;\leq\; C\, b\, \sqrt{\log\Big( \frac{eA'M\tau}{b} \Big)} \leq C'b,$$

we conclude that

$$\mathbb{E}_\varepsilon\Big[ \sup_{f \in \mathcal{F}_{u,\tau}} \Big| \frac{1}{n} \sum_{i=1}^n \varepsilon_i f(x_i) \Big| \Big] \leq C\, b \sqrt{\frac{d}{n}}.$$

Combining with (D.83) yields

$$\mathbb{E}\Big[ \sup_{\theta, \widetilde{\theta} \in \Theta} |(P_n - P)f^{(\tau)}_{u,\theta,\widetilde{\theta}}| \Big] \leq C\,(4M\tau) \sqrt{\frac{d}{n}}. \tag{D.84}$$

We now apply Theorem D.25 to two shifted classes to handle two-sided deviations. Define $\mathcal{F}^+ := \{(f + b)/2 : f \in \mathcal{F}_{u,\tau}\}$ and $\mathcal{F}^- := \{(-f + b)/2 : f \in \mathcal{F}_{u,\tau}\}$. Note that $0 \leq (\pm f + b)/2 \leq b$ and

$$\sup_{f \in \mathcal{F}_{u,\tau}} \Big| \sum_{i=1}^n (f(x_i) - \mathbb{E}f) \Big| \leq 2\max\{Z^+, Z^-\},$$

where $Z^\pm$ are the suprema for $\mathcal{F}^\pm$. Combining Theorem D.25 (and more precisely the upper bound in (D.73)) with $\sigma^2 \leq \sup_{g \in \mathcal{F}^\pm} \mathbb{E}[g^2] \leq b^2$, dividing by $n$ and using $\mathbb{E}Z^\pm \leq (n/2)\mathbb{E}[\sup_{f \in \mathcal{F}_{u,\tau}} |(P_n - P)f|]$, together with (D.84), we obtain:

$$\sup_{\theta, \widetilde{\theta} \in \Theta} |(P_n - P)f^{(\tau)}_{u,\theta,\widetilde{\theta}}| \leq C\, b \sqrt{\frac{d}{n}} + C\, b \sqrt{\frac{t}{n}} + C\, b \frac{t}{n}, \qquad b = 4M\tau, \tag{D.85}$$

for each fixed $u$ and all $t \geq 1$, with probability at least $1 - 2e^{-t}$. Now choose $t = c_0 d$ with $c_0$ large enough and take a union bound over $u \in \mathcal{U}_{1/4}$ recalling that $|\mathcal{U}_{1/4}| \leq 9^d$. Thus,

$$\max_{u \in \mathcal{U}_{1/4}} \sup_{\theta, \widetilde{\theta} \in \Theta} |(P_n - P)f^{(\tau)}_{u,\theta,\widetilde{\theta}}| \leq C\, b \sqrt{\frac{d}{n}} + C\, b \frac{d}{n}, \qquad b = 4M\tau, \tag{D.86}$$

with probability at least $1 - e^{-cd}$. Combining (D.75), (D.78) and (D.86), we conclude that

$$\max_{u \in \mathcal{U}_{1/4}} \sup_{\theta, \widetilde{\theta} \in \Theta} |(P_n - P)f_{u,\theta,\widetilde{\theta}}| \leq C(4M\tau)\sqrt{\frac{d}{n}} + C(4M\tau)\frac{d}{n} + CM\left( e^{-c\tau} + \sqrt{\frac{e^{-c\tau}\, d}{n}} + \frac{d}{n} \right),$$

with probability at least $1 - 3e^{-cd}$. Using (D.74), we obtain

$$\sup_{\theta, \widetilde{\theta} \in \Theta} \|\widehat{H}(\theta, \widetilde{\theta}) - H(\theta, \widetilde{\theta})\| \leq 2 \max_{u \in \mathcal{U}_{1/4}} \sup_{\theta, \widetilde{\theta} \in \Theta} |(P_n - P)f_{u,\theta,\widetilde{\theta}}|$$

$$\leq C(M\tau)\sqrt{\frac{d}{n}} + C(M\tau)\frac{d}{n} + CM\sqrt{\frac{e^{-c\tau}d}{n}} + CM\frac{d}{n} + CMe^{-c\tau}. \tag{D.87}$$

Setting $\tau := C_0 \log\Big( \frac{eM}{\epsilon} \Big)$ with $C_0$ large enough so that $CMe^{-c\tau} \leq \epsilon/10$ and using (D.64) concludes the proof. $\qquad \square$

## D.6. Exponential convergence during the second phase

**Proposition D.26.** *Let $M$ be a large enough constant, $r \leq 1/4$ and fix $\varphi \leq cr^2/M$ for a small enough constant $c$ (independent of $n, d, M, r, R$). Consider the parameter set*

$$\Theta_\varphi := \left\{ \theta \in \mathbb{R}^d \setminus \{0\} : \angle(\theta, \theta^\star) \leq \varphi, \ r \leq \|\theta\| \leq R \right\}. \tag{D.88}$$

*Assume that*

$$n \geq Cd \frac{M^2}{r^4} \log^2 \left( \frac{M}{r^2} \right), \tag{D.89}$$

*for a large enough constant $C$ (independent of $n, d, M, r, R$). Consider the truncated quadratic activation in* (4.4), *and let $\theta_t$ be obtained from the gradient descent iteration in* (4.2), *with learning rate $\eta \leq cr^2/M^2$. Assume further that there exists $\bar{t}$ such that, for all $t \geq \bar{t}$, $\theta_t \in \Theta_\varphi$. Then, with probability at least $1 - 7e^{-cd}$, we have*

$$\|\theta_t - \theta^\star\|^2 \leq (1 - \eta\alpha)^{t - \bar{t}} \|\theta_\tau - \theta^\star\|^2,$$

*with $\alpha = cr^2$.*

We start with a preliminary geometric lemma. For $a, b \in \mathcal{S}^{d-1}$, define the sign-disagreement event

$$D(a, b) = \{x : \text{sign}(\langle x, a \rangle) \neq \text{sign}(\langle x, b \rangle)\}.$$

Let us also define $\mathcal{S}(\theta) := \{x : \langle x, \theta \rangle \langle x, \theta^\star \rangle < 0\}$ and note that $\mathcal{S}(\theta) = D(\theta, \theta^\star)$.

**Lemma D.27.** *Let $g \sim \mathcal{N}(0, I_d)$, and let $a, b, u \in \mathcal{S}^{d-1}$. Set $\alpha = \angle(a, b) \in [0, \pi]$. Then*

$$\mathbb{E}\left[ \langle g, u \rangle^2 \mathbf{1}\{D(a, b)\} \right] \leq \frac{2}{\pi} \alpha.$$

*Proof.* Let $S = \text{span}\{a, b\}$ have dimension $\dim S$, and write the orthogonal decompositions

$$u = u_S + u_\perp, \qquad g = g_S + g_\perp,$$

with $u_S, g_S \in S$ and $u_\perp, g_\perp \in S^\perp$, where $S^\perp$ denotes the subspace orthogonal to $S$. Since $g \sim \mathcal{N}(0, I_d)$, we have that $g_S \sim \mathcal{N}(0, I_{\dim S})$ is independent of $g_\perp \sim \mathcal{N}(0, I_{d - \dim S})$. Note that $D(a, b)$ depends only on $\langle g, a \rangle, \langle g, b \rangle$ and hence only on $g_S$.

If $\dim S \leq 1$, then $a = \pm b$, which implies that $\alpha \in \{0, \pi\}$. For $\alpha = 0$, $\mathbf{1}\{D(a, b)\} = 0$, hence both expectations are 0 and the bounds hold. For $\alpha = \pi$, $\mathbf{1}\{D(a, b)\} = 1$, $\mathbb{E}\left[ \langle g, u \rangle^2 \right] = 1$, $\mathbb{E}\left[ \langle g, u \rangle^4 \right] = 3$, hence the bounds hold. Thus, we assume below that $\dim S = 2$ (equivalently $\alpha \in (0, \pi)$). Define

$$X := \langle g_S, u_S \rangle, \qquad Y := \langle g_\perp, u_\perp \rangle,$$

so that $\langle g, u \rangle = X + Y$. Note that $Y$ is independent of $(X, \mathbf{1}\{D(a, b)\})$ and satisfies $\mathbb{E}[Y] = 0$, $\mathbb{E}[Y^2] = \|u_\perp\|^2$, and $\mathbb{E}[Y^4] = 3\|u_\perp\|^4$.

By rotational invariance in the plane $S$, choose an orthonormal basis of $S$ such that

$$a = e_1, \qquad b = \cos \alpha \, e_1 + \sin \alpha \, e_2.$$

Let us write $g_S \in \mathbb{R}^2$ in polar form

$$g_S = r(\cos \theta, \sin \theta),$$

where $\theta \sim \text{Unif}[0, 2\pi)$ is independent of $r \geq 0$, and $r^2 \sim \chi_2^2$. In particular,

$$\mathbb{E}[r^2] = 2, \qquad \mathbb{E}[r^4] = 8.$$

Furthermore, we have

$$\langle g_S, a \rangle = r \cos \theta, \qquad \langle g_S, b \rangle = r \cos(\theta - \alpha),$$

which implies that
$$D(a,b) = \{\cos\theta \cdot \cos(\theta - \alpha) < 0\}.$$

Note that $D(a,b)$ depends on $\theta$ and not on $r$. On the circle this set consists of two disjoint angular intervals, each of length $\alpha$, hence
$$\mathbb{P}(g \in D(a,b)) = \frac{2\alpha}{2\pi} = \frac{\alpha}{\pi}. \tag{D.90}$$

Using that $Y$ is independent of $(X, \mathbf{1}\{D(a,b)\})$ and $\mathbb{E}[Y] = 0$, we obtain
$$\mathbb{E}[(X+Y)^2 \mathbf{1}\{D(a,b)\}] = \mathbb{E}[X^2 \mathbf{1}\{D(a,b)\}] + 2\mathbb{E}[XY\mathbf{1}\{D(a,b)\}] + \mathbb{E}[Y^2 \mathbf{1}\{D(a,b)\}]$$
$$= \mathbb{E}[X^2 \mathbf{1}\{D(a,b)\}] + \mathbb{E}[Y^2]\mathbb{P}(g \in \mathbf{1}\{D(a,b)\}).$$

Next, we write $u_S = \|u_S\|_2 (\cos\beta, \sin\beta)$ for some $\beta$. Then, we have
$$X = \langle g_S, u_S \rangle = r\|u_S\| \cos(\theta - \beta),$$

and therefore
$$X^2 \mathbf{1}\{D(a,b)\} = r^2 \|u_S\|^2 \cos^2(\theta - \beta)\, \mathbf{1}\{D(a,b)\}.$$

Using the independence of $r$ and $\theta$ and the fact that $\cos^2\gamma \leq 1$ for all $\gamma$, we have
$$\mathbb{E}[X^2 \mathbf{1}\{D(a,b)\}] = \mathbb{E}[r^2]\|u_S\|^2\, \mathbb{E}\big[\cos^2(\theta - \alpha)\mathbf{1}\{D(a,b)\}\big] \leq \mathbb{E}[r^2]\|u_S\|^2\, \mathbb{P}(g \in \mathbf{1}\{D(a,b)\}) = 2\|u_S\|^2 \frac{\alpha}{\pi},$$

where we used (D.90) in the last step. Hence,
$$\mathbb{E}[\langle g, u \rangle^2 \mathbf{1}\{D(a,b)\}] \leq \big(2\|u_S\|^2 + \|u_\perp\|^2\big) \frac{\alpha}{\pi} \leq \frac{2}{\pi}\alpha,$$

since $\|u_S\|^2 + \|u_\perp\|^2 = \|u\|^2 = 1$. □

We are now ready to prove Proposition D.26.

*Proof of Proposition D.26.* First we show that
$$\|\nabla\mathcal{L}(\theta)\| \leq L\|\theta - \theta^\star\|,$$

with $L := 4M$. To do this, note that using Cauchy-Schwarz and the fact that $\frac{1}{n}\sum_{i=1}^n x_i x_i^\top \preceq 2I$ with probability at least $1 - 2e^{-cd}$ we have

$$\|\nabla\mathcal{L}(\theta)\| = \sup_{u \in \mathcal{S}^{d-1}} \frac{1}{n}\sum_{i=1}^n \left(\sigma(\langle x_i, \theta\rangle) - \sigma(\langle x_i, \theta^\star\rangle)\right) \sigma'(\langle x_i, \theta\rangle)(\langle x_i, u\rangle)$$

$$\leq \sqrt{M}\sqrt{\frac{1}{n}\sum_{i=1}^n \left(\sigma(\langle x_i, \theta\rangle) - \sigma(\langle x_i, \theta^\star\rangle)\right)^2}\sqrt{\frac{1}{n}\sum_{i=1}^n (\langle x_i, u\rangle)^2}$$

$$\leq \sqrt{2M}\sqrt{\frac{1}{n}\sum_{i=1}^n \left(\sigma(\langle x_i, \theta\rangle) - \sigma(\langle x_i, \theta^\star\rangle)\right)^2}$$

$$\leq \sqrt{8M}\sqrt{\frac{1}{n}\sum_{i=1}^n \left(\langle x_i, \theta - \theta^\star\rangle\right)^2}$$

$$\leq 4M\|\theta - \theta^\star\|,$$

where in the penultimate step we used the fact that $\sigma$ is $2\sqrt{M}$-Lipschitz. The key part of the argument is to show the one-point strong convexity
$$\left\langle \widehat{\mathcal{G}}(\theta), \theta - \theta^\star \right\rangle \geq \alpha\|\theta - \theta^\star\|^2, \tag{D.91}$$

with $\alpha = cr^2$ for some numerical constant $c > 0$ independent of $n, d, M, r, R$. Having established (D.91), if we let $\theta$ and $\theta^+ = \theta - \eta\nabla\mathcal{L}(\theta)$ be two subsequent gradient descent iterations such that $\theta, \theta^+ \in \Theta_\varphi$, we have

$$
\begin{aligned}
\|\theta^+ - \theta^\star\|^2 &= \|\theta - \theta^\star - \eta\nabla\mathcal{L}(\theta)\|^2 \\
&= \|\theta - \theta^\star\|^2 - 2\eta\langle\mathcal{L}(\theta), \theta - \theta^\star\rangle + \eta^2\|\nabla\mathcal{L}(\theta)\|^2 \\
&\leq \|\theta - \theta^\star\|^2 - 2\alpha\eta\|\theta - \theta^\star\|^2 + \eta^2 L^2\|\theta - \theta^\star\|^2 \\
&\leq (1 - \alpha\eta)\|\theta - \theta^\star\|^2,
\end{aligned}
\tag{D.92}
$$

where in the last line we used the fact $\eta \leq \frac{\alpha}{L^2}$. Hence, the desired result follows by iterating (D.92).

The rest of the argument consists in proving (D.91). Let us write the inner product

$$
\left\langle\widehat{\mathcal{G}}(\theta), \theta - \theta^\star\right\rangle = \frac{1}{n}\sum_{i=1}^n \left(\sigma\left(\langle x_i, \theta\rangle\right) - \sigma(\langle x_i, \theta^\star\rangle)\right)\sigma'(\langle x_i, \theta\rangle)\langle x_i, \theta - \theta^\star\rangle.
\tag{D.93}
$$

Now we apply the integral form of the mean value theorem

$$
\sigma(\langle x_i, \theta\rangle) - \sigma(\langle x_i, \theta^\star\rangle) = \left(\int_0^1 \sigma'\left(\langle x_i, \theta^\star + t(\theta - \theta^\star)\rangle dt\right)\right)\langle x_i, \theta - \theta^\star\rangle.
$$

Thus,

$$
\begin{aligned}
\langle\widehat{\mathcal{G}}(\theta), \theta - \theta^\star\rangle &= (\theta - \theta^\star)^\top \left(\int_0^1 \left(\frac{1}{n}\sum_{i=1}^n \sigma'\left(\langle x_i, \theta^\star + t(\theta - \theta^\star)\rangle dt\right)\sigma'(\langle x_i, \theta\rangle)x_i x_i^\top\right) dt\right)(\theta - \theta^\star) \\
&= (\theta - \theta^\star)^\top \left(\int_0^1 \widehat{H}(\theta^\star + t(\theta - \theta^\star), \theta)dt\right)(\theta - \theta^\star),
\end{aligned}
$$

where $\hat{H}$ is the Gram matrix of the Jacobian defined in (D.63). Therefore,

$$
\begin{aligned}
&\langle\widehat{\mathcal{G}}(\theta), \theta - \theta^\star\rangle - \mathbb{E}\left[\langle\widehat{\mathcal{G}}(\theta), \theta - \theta^\star\rangle\right] \\
&= (\theta - \theta^\star)^\top \left(\int_0^1 \left(\widehat{H}(\theta^\star + t(\theta - \theta^\star), \theta) - \mathbb{E}\left[\widehat{H}(\theta^\star + t(\theta - \theta^\star), \theta)\right]\right) dt\right)(\theta - \theta^\star).
\end{aligned}
$$

We lower bound the quantity $\mathbb{E}\left[\langle\widehat{\mathcal{G}}(\theta), \theta - \theta^\star\rangle\right]$. To that aim, let us define the event

$$
\mathcal{S} = \{\text{sign}(\langle x, \theta\rangle) = \text{sign}(\langle x, \theta^\star\rangle)\},
$$

and decompose (D.93) based on the alignment of the signs into two terms

$$
\begin{aligned}
\mathbb{E}\left[\langle\widehat{\mathcal{G}}(\theta), \theta - \theta^\star\rangle\right] &= \mathbb{E}\left[\left(\sigma\left(\langle x, \theta\rangle\right) - \sigma(\langle x, \theta^\star\rangle)\right)\sigma'(\langle x, \theta\rangle)\langle x, \theta - \theta^\star\rangle\mathbf{1}\{\mathcal{S}^c\}\right] \\
&\quad + \mathbb{E}\left[\left(\sigma\left(\langle x, \theta\rangle\right) - \sigma(\langle x, \theta^\star\rangle)\right)\sigma'(\langle x, \theta\rangle)\langle x, \theta - \theta^\star\rangle\mathbf{1}\{\mathcal{S}\}\right].
\end{aligned}
\tag{D.94}
$$

Towards bounding the first term in the RHS of (D.94), we write

$$
\left|\left(\sigma\left(\langle x, \theta\rangle\right) - \sigma(\langle x, \theta^\star\rangle)\right)\sigma'(\langle x, \theta\rangle)\langle x, \theta - \theta^\star\rangle\right| \leq 4M\langle x, \theta - \theta^\star\rangle^2,
$$

where we have used that $\sigma$ is $2\sqrt{M}$-Lipschitz. Thus, we obtain the lower bound

$$
\mathbb{E}\left[\left(\sigma\left(\langle x, \theta\rangle\right) - \sigma(\langle x, \theta^\star\rangle)\right)\sigma'(\langle x, \theta\rangle)\langle x, \theta - \theta^\star\rangle\mathbf{1}\{\mathcal{S}^c\}\right] \geq -4M\mathbb{E}\left[\langle x, \theta - \theta^\star\rangle^2\mathbf{1}\{\mathcal{S}^c\}\right] \geq -\frac{8M}{\pi}\varphi\|\theta - \theta^\star\|^2,
\tag{D.95}
$$

where the last passage follows from Lemma D.27.

Next, we claim the following lower bound on the second term of the RHS in (D.94):

$$
\mathbb{E}\left[\left(\sigma(\langle x, \theta\rangle) - \sigma(\langle x, \theta^\star\rangle)\right)\sigma'(\langle x, \theta\rangle)\left(\langle x, \theta\rangle - \langle x, \theta^\star\rangle\right)\mathbf{1}\{\mathcal{S}\}\right] \geq \frac{f(1/4)f(1/8)}{2^{19}}r^2(\cos\varphi - \sin\varphi)^2\|\theta - \theta^\star\|_2^2,
\tag{D.96}
$$

where $f(u) = (2\pi)^{-1/2}e^{-u^2/2}$ is the standard normal density.

**Proof of the claim in** (D.96). Let us write

$$a = \langle x, \theta \rangle, \qquad b = \langle x, \theta^\star \rangle, \qquad u = \theta - \theta^\star,$$

and note that the LHS of (D.96) can be expressed as

$$Q := \mathbb{E}\Big[(\sigma(a) - \sigma(b))\,\sigma'(a)\,(a - b)\,\mathbf{1}\{\mathcal{S}\}\Big].$$

As $a$ and $b$ have the same sign on $\mathcal{S}$, we have $(\sigma(a) - \sigma(b))\,\sigma'(a)\,(a - b) \geq 0$. Thus, adding an extra indicator only reduces $Q$ and we obtain the lower bound

$$Q \geq \mathbb{E}\Big[(\sigma(a) - \sigma(b))\,\sigma'(a)\,(a - b)\,\mathbf{1}\{\mathcal{S}\}\mathbf{1}\{|a| \leq \sqrt{M}, |b| \leq \sqrt{M}\}\Big].$$

On the event $\{|a| < \sqrt{M},\ |b| < \sqrt{M}\}$ we have $\sigma(a) = a^2$, $\sigma(b) = b^2$, and $\sigma'(a) = 2a$, hence

$$(\sigma(a) - \sigma(b))\,\sigma'(a)\,(a - b) = (a^2 - b^2)\cdot 2a \cdot (a - b) = 2a(a + b)(a - b)^2.$$

On $\mathcal{S} = \{ab \geq 0\}$, we further obtain that

$$a(a + b) = a^2 + ab \geq a^2.$$

Consequently, on $\{|a| < \sqrt{M},\ |b| < \sqrt{M}\} \cap \mathcal{S}$,

$$(\sigma(a) - \sigma(b))\,\sigma'(a)\,(a - b) \;\geq\; 2a^2(a - b)^2.$$

Thus

$$Q \;\geq\; 2\,\mathbb{E}\Big[a^2(a - b)^2\,\mathbf{1}\{\{|a| < \sqrt{M},\ |b| < \sqrt{M}\} \cap \mathcal{S}\}\Big].$$

Let $\phi := \angle(\theta, \theta^\star) \leq \varphi$ and $r \leq \|\theta\| \leq R$. Focusing on the 2-dimensional subspace spanned by $\theta^\star$ and $\theta$, pick an orthonormal basis $(e_1, e_2)$ for this plane with $e_1 = \theta^\star$. Then, we can write

$$x = g_1 e_1 + g_2 e_2 + x_\perp,$$

where $g_1, g_2 \overset{\text{iid}}{\sim} \mathcal{N}(0, 1)$ and $x_\perp$ is independent of $(g_1, g_2)$. Thus,

$$b = g_1, \qquad a = \|\theta\|(\cos\phi\, g_1 + \sin\phi\, g_2).$$

Define the event

$$\mathcal{B} := \left\{g_1 \in \left[\frac{1}{8}, \frac{1}{4}\right],\ |g_2| \leq \frac{1}{8}\right\}.$$

On $\mathcal{B}$, we have

$$\cos\phi\, g_1 + \sin\phi\, g_2 \;\geq\; \frac{1}{8}(\cos\phi - \sin\phi) \;>\; 0 \qquad (\text{since } \varphi < \pi/4),$$

hence $a > 0$ and also $b > 0$. Therefore $\mathcal{B} \subset \mathcal{S}$. Moreover, on $\mathcal{B}$, using the assumption on $M \geq 16R^2$ and $R \geq 1$, we have

$$|b| = |g_1| \leq \frac{1}{4} \leq \sqrt{M},$$

and

$$|a| \leq \|\theta\|(|g_1| + |g_2|) \leq R \cdot \frac{3}{8} \leq \sqrt{M},$$

so $\mathcal{B} \subset \{|a| < \sqrt{M},\ |b| < \sqrt{M}\}$. Finally, still on $\mathcal{B}$,

$$a \;\geq\; \frac{\|\theta\|}{8}(\cos\phi - \sin\phi) \;\geq\; r \cdot \frac{1}{8}(\cos\varphi - \sin\varphi),$$

and therefore

$$a^2 \;\geq\; \frac{r^2}{64}(\cos\varphi - \sin\varphi)^2.$$

Thus, using $\mathcal{B} \subset \{|a| < \sqrt{M}, |b| < \sqrt{M}\} \cap \mathcal{S}$ yields

$$Q \geq 2\,\mathbb{E}\Big[a^2(a-b)^2\,\mathbf{1}\{\mathcal{B}\}\Big] \geq \frac{r^2}{32}(\cos\varphi - \sin\varphi)^2\,\mathbb{E}\Big[(a-b)^2\,\mathbf{1}\{\mathcal{B}\}\Big].$$

Note that $a - b = \langle x, u \rangle$. Let us decompose $u = u_{\parallel} + u_{\perp}$ where $u_{\parallel} \in \mathrm{span}\{e_1, e_2\}$ and $u_{\perp} \perp \mathrm{span}\{e_1, e_2\}$. We also write $u_{\parallel} = u_1 e_1 + u_2 e_2$, which gives that

$$\langle x, u \rangle = u_1 g_1 + u_2 g_2 + \langle x_{\perp}, u_{\perp} \rangle.$$

Note that $\langle x_{\perp}, u_{\perp} \rangle \sim \mathcal{N}(0, \|u_{\perp}\|_2^2)$ is independent of $(g_1, g_2)$ and hence independent of $\mathcal{B}$. Therefore, we have

$$\mathbb{E}[(a - b)^2 \mathbf{1}\{\mathcal{B}\}] = \mathbb{E}[(\langle x, u \rangle)^2 \mathbf{1}\{\mathcal{B}\}] \geq \mathbb{E}[(u_1 g_1 + u_2 g_2)^2 \mathbf{1}\{\mathcal{B}\}].$$

Since $\mathcal{B}$ is symmetric in $g_2$, $\mathbb{E}[g_1 g_2 \mathbf{1}\{\mathcal{B}\}] = 0$, and hence

$$\mathbb{E}[(u_1 g_1 + u_2 g_2)^2 \mathbf{1}\{\mathcal{B}\}] = u_1^2\,\mathbb{E}[g_1^2 \mathbf{1}\{\mathcal{B}\}] + u_2^2\,\mathbb{E}[g_2^2 \mathbf{1}\{\mathcal{B}\}].$$

We now lower bound the two expectations. Because the standard normal density $f$ is decreasing on $[0, \infty)$,

$$\mathbb{P}(g_1 \in [1/8, 1/4]) = \int_{1/8}^{1/4} f(s)\,ds \geq \frac{1}{8}f(1/4), \qquad \mathbb{P}(|g_2| \leq 1/8) = \int_{-1/8}^{1/8} f(s)\,ds \geq \frac{1}{4}f(1/8).$$

Hence

$$\mathbb{P}(\mathcal{B}) \geq \frac{1}{32}f(1/4)f(1/8).$$

On $\mathcal{B}$ we have $g_1^2 \geq \frac{1}{64}$, so

$$\mathbb{E}[g_1^2 \mathbf{1}\{\mathcal{B}\}] \geq \frac{1}{64}\mathbb{P}(\mathcal{B}) \geq \frac{1}{2048}f(1/4)f(1/8).$$

For $g_2$, define the subset

$$\mathcal{B}' := \{g_1 \in [1/8, 1/4],\ |g_2| \in [1/16, 1/8]\} \subset \mathcal{B}.$$

On $\mathcal{B}'$ we have $g_2^2 \geq \frac{1}{16^2}$. Also,

$$\mathbb{P}(|g_2| \in [1/16, 1/8]) = 2\int_{1/16}^{1/8} f(s)\,ds \geq 2 \cdot \frac{1}{16}f(1/8) = \frac{1}{8}f(1/8),$$

so

$$\mathbb{P}(\mathcal{B}') \geq \mathbb{P}(g_1 \in [1/8, 1/4]) \cdot \mathbb{P}(|g_2| \in [1/16, 1/8]) \geq \frac{1}{8}\phi(1/4) \cdot \frac{1}{8}f(1/8) = \frac{1}{64}f(1/4)f(1/8).$$

Therefore

$$\mathbb{E}[g_2^2 \mathbf{1}\{\mathcal{B}\}] \geq \mathbb{E}[g_2^2 \mathbf{1}\{\mathcal{B}'\}] \geq \frac{1}{16^2}\mathbb{P}(\mathcal{B}') \geq \frac{1}{16384}f(1/4)f(1/8).$$

Combining the two bounds gives

$$\mathbb{E}[(a - b)^2 \mathbf{1}\{\mathcal{B}\}] \geq \frac{1}{16384}f(1/4)f(1/8)\,(u_1^2 + u_2^2) \geq \frac{1}{16384}f(1/4)f(1/8)\,\|u\|^2.$$

This allows us to conclude that

$$Q \geq \frac{r^2}{32}(\cos\varphi - \sin\varphi)^2 \cdot \frac{1}{16384}f(1/4)f(1/8)\,\|u\|_2^2 = \frac{f(1/4)f(1/8)}{524288}r^2(\cos\varphi - \sin\varphi)^2\,\|\theta - \theta^{\star}\|_2^2,$$

which is the bound in (D.96). $\qquad\square$

We are now ready to put everything together and conclude the proof. By combining (D.94), (D.95) and (D.96), we have

$$\mathbb{E}\Big[\langle \widehat{\mathcal{G}}(\theta), \theta - \theta^{\star} \rangle\Big] \geq \Big(c_1 r^2 - \frac{8M}{\pi}\varphi\Big)\|\theta - \theta^{\star}\|_2^2, \tag{D.97}$$

where we have used that, as $\varphi$ is small, $\cos\varphi - \sin\varphi$ is lower bounded by a strictly positive numerical constant $c_0$ and $c_1 := c_0 f(1/4)f(1/8)/524288$ is also a strictly positive numerical constant (independent of $n, d, M, r, R$). Next, we apply Proposition D.20 with $\epsilon = c_1 r^2/2$ (which is possible due to the lower bound on $n$ in (D.89)) and therefore we conclude that, with probability at least $1 - 3e^{-cd}$,

$$\langle \widehat{\mathcal{G}}(\theta), \theta - \theta^\star \rangle \geq \left( \frac{c_1}{2} r^2 - \frac{8M}{\pi} \varphi \right) \|\theta - \theta^\star\|_2^2 \geq c_2 r^2 \|\theta - \theta^\star\|_2^2, \tag{D.98}$$

where in the last step we use that $\varphi \leq cr^2/M$ for a small enough numerical constant $c$. This proves the one-point strong convexity in (D.91) and concludes the argument. $\square$

### D.7. Patching the two phases and concluding the argument

*Proof of Theorem 4.1.* As $\eta \leq c/M^2$ and $\delta \geq CM^4$, we can apply Proposition D.1, which gives that, with probability at least $1 - \frac{1}{d^2}$,

$$\angle(\theta_{t^*_{\angle}}, \theta^\star) \leq C_1(e^{-M/2} + M\delta^{-1/2}), \qquad \text{with } t^*_{\angle} = \frac{3\log d}{\log(1 + 1.99\eta)}, \tag{D.99}$$

for numerical constants $c_1, C_1 > 0$ (independent of $n, d, M, \delta$). Furthermore, the constraints $\eta \leq c/M^2$ and $\delta \geq CM^4$ also allow to apply Proposition D.14 with $r = d^{-15}$, which gives that, with probability at least $1 - Ce^{-cd}$,

$$\|\theta_t\| \geq 1/4, \qquad \text{for all } t \geq t^*_{\text{norm}} := \left\lceil \frac{2\log\left(\frac{1}{4\|\theta_0\|}\right)}{\log\left(1 + \frac{19}{25}\eta\right)} \right\rceil, \tag{D.100}$$

and that

$$\|\theta_t\| \leq 10, \qquad \text{for all } t \geq 0. \tag{D.101}$$

Next, we apply Proposition D.15 with $r = d^{-15}$, $\varphi = C_1(e^{-M/2} + M\delta^{-1/2})$ and $t^* = t^*_{\angle}$, which gives that, with probability at least $1 - Ce^{-cd}$,

$$\angle(\theta_t, \theta^\star) \leq C_1(e^{-M/2} + M\delta^{-1/2}), \qquad \text{for all } t \geq t^*_{\angle}. \tag{D.102}$$

Note that, as $\delta \geq CM^4$ for a large enough constant $C$, the upper bound on $\angle(\theta_t, \theta^\star)$ in (D.102) can be made $c_2/M$ for any small constant $c_2$. Hence, we can apply Proposition D.26 with $r = 1/4$, $R = 10$, $\varphi = c_2/M$ and $\bar{t} = \max(t^*_{\angle}, t^*_{\text{norm}})$, obtaining that, for all $t \geq \bar{t}$, with probability at least $1 - 7e^{-cd}$,

$$\|\theta_t - \theta^\star\|^2 \leq (1 - \eta\alpha)^{t - \bar{t}} \|\theta_{\bar{t}} - \theta^\star\|^2,$$

with $\alpha > 0$ a numerical constant independent of $n, d, M, \delta$. Note that $\|\theta_{\bar{t}} - \theta^\star\|^2 \leq 202$ by using (D.101). Furthermore, we have that $\max(t^*_{\angle}, t^*_{\text{norm}}) \leq C\log d/\eta$. Thus, a union bound on all these high-probability events gives the desired claim with probability at least $1 - 2/d^2$, thus concluding the argument. $\square$

