# OpenReview forum: "Full-Batch Gradient Descent Outperforms One-Pass SGD: Sample Complexity Separation in Single-Index Learning"
_ICML.cc/2026/Conference — ICML 2026 regular_

### Official Review · Reviewer_r6BH · 2026-03-04

**Soundness:** 4
**Presentation:** 4
**Significance:** 3
**Originality:** 2
**Overall Recommendation:** 3
**Confidence:** 3

**Summary:**

This paper looks at how gradient descent behaves on a simple single-index model. They show two contrasting results . First, if you just use the plain quadratic activation, full-batch gradient flow basically does nothing unless you have way more than $d\log d$ samples so full-batch updates don’t help compared to a single pass of SGD. But then, with a simple tweak, truncating the quadratic activation, they show full-batch gradient flow only needs $d$ samples. to get weak recovery, while one-pass SGD still needs $d\log d$. So, they show a clear separation result. Training with GD happens in two phases: first, the weights quickly align in the right direction, and then they converge exponentially fast to the true target once they’re close enough. Training with GD happens in two phases: the weights first quickly align in the right direction, and then they converge exponentially fast to the true target once they’re close enough. The truncated quadratic activation makes the math work out, but the main takeaway is that gradient descent can reliably find the global minimum here, showing that you can have nice landscape under the right conditions.

**Compliance With Llm Reviewing Policy:**

Affirmed.

**Key Questions For Authors:**

1) Your main results rely on the truncated quadratic activation, do you have any intuition or preliminary experiments suggesting similar behavior with more standard activations like ReLU or sigmoid?

2) The separation between full-batch GD and one-pass SGD is elegant, but do you expect it to hold beyond the single-index model, say in multi-index models or deeper architectures? Can you also provide some intuition here?

3) Could you clarify whether the linear in $d$ sample complexity is only for the realizable SIM case? Can you say something about non-realizable more realistic settings?

**Limitations:**

yes

**Strengths And Weaknesses:**

$\textbf{Strengths}$
The paper gives a really clear and rigorous picture of how GD behaves on a single-index model. One of its strongest points is showing the two-phase dynamics: first fast alignment of weights, then exponential convergence once close to the target. The truncated quadratic activation lets them get finite-sample guarantees, and it highlights a clear separation between full-batch GD and one-pass SGD. Overall, it shows that even very simple non-linear networks can be well-behaved under the right conditions, which is encouraging for understanding training dynamics in a more principled way.

$\textbf{Weaknesses}$
On the flip side, the results rely heavily on the truncated quadratic activation, which isn’t exactly standard in practice, so it’s unclear how much the theory extends to more natural activations like ReLU or sigmoid, although the authors mention that they expect it would hold more broadly. The model itself is very simple, a single-index model, so the findings might not directly carry over to more complex networks or higher-dimensional features. Overall, while I understand the importance of a separation result, this only applies in this very specific setup, so I’m not convinced about its practical impact.

---

> ### Author Rebuttal · Authors · 2026-03-30
>
> We thank the reviewer for the thoughtful comments. We address the three questions below.
>
> **Q1.** We first note that we focus on activations for which the sample complexity of online SGD is at least $d \log d$, as motivated by phase retrieval. This covers all even activations, and the most natural choice is the quadratic function.
> That said, we conjecture that our positive results hold more generally for any bounded activation. The necessity of boundedness follows directly from the argument of Theorem 3.1: informally, if the bulk of the spectrum of the random matrix on which we perform power iteration diverges, then a super-linear sample size is required for a BBP transition (and hence weak recovery) to occur.
>
> To extend weak recovery to gradient flow on the correlation loss with a general activation, the challenge is to prove a uniform-in-$\theta$ BBP transition for the matrix $A(\theta) = \frac{1}{n}\sum_{i=1}^n y_i \frac{\sigma'(\langle x_i, \theta\rangle)}{\langle x_i, \theta\rangle} x_i x_i^\top$. Since $y_i = \sigma(\langle x_i, \theta^\star\rangle)$, this reduces to establishing a BBP transition for a multi-index model (with the two indices corresponding to $\theta$ and $\theta^\star$). Precise characterizations of such transitions have been developed in [Kovačević et al., 2025; Defilippis et al., 2025], which would serve as the starting point.
>
> To extend strong recovery for gradient descent on the squared loss, one can follow a two-phase strategy similar to Theorem 4.1: (i) a first phase starting from small initialization, where the parameter aligns with the signal before the norm grows large enough for truncation to become active; and (ii) a refinement phase via concentration arguments. We expect the concentration arguments to extend to general activations with bounded higher-order derivatives, though substantial technical work is required. The first phase is more delicate, but we expect it to go through as long as $\sigma''(0) > 0$. We regard this as an interesting future direction beyond the scope of the current contribution.
>
> Finally, for activations with information exponent 1 (e.g., ReLU, sigmoid), the sample complexity of online SGD has no additional $\log$ factor, so the separation we study does not arise. We provide supporting numerical evidence that linear sample complexity is achievable for sigmoid activation in [[this figure]](https://ibb.co/B2vQyDJ3): we run online SGD with step size proportional to $1/d$ as prescribed by [Ben Arous et al. 2021], and report the achieved overlap vs. time (i.e., step size $\times$ number of SGD steps). We observe perfect collapse across dimensionality $d$, suggesting a runtime (and therefore sample size) exactly linear in $d$.
>
> **Q2.** We expect the separation to extend to multi-index models, for which the natural generalization of the information-exponent-2 setting is the class of leap-exponent-2 functions, see [Abbe et al., 2023]. For this class, [Montanari & Wang, 2026] (which appeared on arXiv after the ICML deadline) study the sample complexity of recovering the first leap-2 direction and give evidence that full-batch GD succeeds with $n = \Theta(d)$ samples, while a lower bound analogous to [Ben Arous et al., 2021] predicts that online SGD requires super-linear sample size. We note that [Montanari & Wang, 2026] do not provide a proof of convergence of full-batch GD, which remains an outstanding challenge for multi-index models.
>
> **Q3.** Our results focus on the realizable case. When the target function is misspecified (i.e., with mismatched link function), gradient-based learning decomposes into two components: (i) identifying the hidden direction (search phase), and (ii) fitting the nonlinear link function (which requires multiple student neurons). Intuitively, the initial search phase should behave similarly to the well-specified setting, since the student neurons are nearly orthogonal to the ground truth; we therefore expect that variants of our techniques can establish weak recovery for this phase. For the second phase, multiple student neurons are needed to approximate the unknown link function once constant overlap is achieved — this process can be complex and goes beyond our current analysis. That said, it is plausible that link fitting occurs on an $T=O(1)$ timescale and can be characterized using techniques from [Berthier et al., 2025]. We intend to investigate this in future work.
>
> We would be happy to answer any follow-up questions in the discussion period.

---

> > ### Author Rebuttal · Reviewer_r6BH · 2026-04-04
> >
> > Thank you for the discussion.
> >
> > I would like to clarify my current stance after reading the rebuttal as well as the comments by reviewer Cy18.
> >
> > Initially, I found the paper technically and conceptually interesting and was inclined to view the contribution as novel, particularly due to the clean separation result for full-batch GD. My main reservation at that stage was about the generality of the setting (e.g., reliance on truncated quadratic activations and single-index models), which is why I asked you the questions above.
> >
> > However, after carefully going through the references and points raised by reviewer Cy18 as well as your answers, I am now less certain about how to position the contribution relative to existing work. In particular, several prior results appear closely related in spirit, and I also find it strange that the authors stick to the view the fact that "theoretical advantage of fullbatch [...] remains unclear". Based on my understanding, this statement seems misleading, as there are well-established results showing clear statistical separations in nonlinear settings.
> >
> > Given this, I would appreciate a more precise and explicit positioning of the present work within the existing literature, in particular clarifying how it differs from these prior results. Is this work discussing only results about Equation 1.1 (i.e. a single index model? or are you trying to argue something more?)
> >
> > As mentioned already, at this point, I remain quite interested in the technical results, but my final assessment is conditional on a clearer understanding of how this work is placed into the broader literature.

---

> > > ### Author Response · Authors · 2026-04-06
> > >
> > > We thank the reviewer for the engagement and the opportunity to clarify. We believe that most of the relevant points were already present either in our original manuscript or in our previous responses to Reviewer Cy18, and we summarize them below.
> > >
> > > ---
> > >
> > > *"This statement seems misleading, as there are well-established results showing clear statistical separations in nonlinear settings"*
> > >
> > > As detailed in our responses to Reviewer Cy18, all references mentioned by the reviewer study different algorithms or model settings, and we already cited and discussed these references in the original manuscript. We recap the key points and our proposed changes and clarifications below.
> > >
> > > 1. The works (Dandi et al., 2024; Lee et al., 2024; Arnaboldi et al., 2024) show that reusing each data point twice may lower the effective information exponent; we explicitly discuss this mechanism in our original manuscript (Section 1.2, l. 137–148) and explain why it does not apply to our setting: for even activation functions, the exponent remains 2 after arbitrary preprocessing. Moreover, we argue this benefit is primarily computational — single-pass SGD on a modified loss achieves the same statistical efficiency — rather than the genuinely statistical benefit we establish for full-batch GD. That said, we acknowledge that this discussion should be more prominent, and we have proposed a concrete revision to l. 14-19 (see our latest reply to Reviewer Cy18) that explicitly scopes our contribution relative to the two-pass loss modification mechanism.
> > >
> > > 2. Regarding the sentence "Notably, most existing analyses of full-batch GD (and its variants) still require $n \gtrsim d\text{polylog}d$, and therefore do not imply a statistical separation from one-pass SGD", this description applies to (Netrapalli et al., 2013; Candès et al., 2015; Sun et al., 2018; Chen et al., 2019), cited immediately after. The next sentence (l. 76) explicitly acknowledges prior works showing "the optimization landscape can be benign at finite $n/d$," citing (Sarao Mannelli et al., 2020b). We do not think this constitutes overclaiming.
> > >
> > > 3. The claim that (Sarao Mannelli et al., 2020b) establish a statistical separation from online SGD does not hold upon inspection: this result requires an overparameterized network with $m > d$ student neurons, and there is no corresponding negative result for online SGD in this regime, so no separation can be inferred. The online SGD analysis in (Arnaboldi et al., 2023) does not rigorously apply to the proportional $m, d$ regime, and (Martin et al., 2024) analyze population gradient flow with no sample complexity guarantee. Combining results across these different settings does not constitute a documented statistical separation — we elaborated on this point at length in our replies to Reviewer Cy18.
> > >
> > > 4. We stand by our claim that "to our knowledge, this is the first convergence rate and strong recovery guarantee for learning single-index models with information exponent 2 using standard full-batch GD in the proportional (finite $n/d$) regime." (Sarao Mannelli et al., 2020b) present a positive landscape result at finite $n/d$ (explicitly acknowledged in l. 78), but contains no analysis of the learning dynamics on the empirical landscape. The convergence rate analysis of (Sarao Mannelli et al., 2020b) applies to the population loss (corresponding to $n=\infty$) and not to finite $n/d$. To avoid any possible misunderstanding, we will further emphasize that our result gives the first end-to-end convergence rate for GD training on the empirical loss (i.e., finite $n/d$).
> > >
> > > We emphasize that all our technical claims stand untouched, and the proposed changes amount to minor text revisions that more precisely contextualize our contributions within the existing literature — contributions whose novelty all reviewers have acknowledged.
> > >
> > > ---
> > >
> > > *"Is this work discussing only results about Equation 1.1, or are you trying to argue something more?"*
> > >
> > > Our technical results are indeed about the single-index model in Equation (1.1) with a quadratic (or truncated quadratic) activation. The broader message is that full-batch GD can achieve a statistical advantage over online SGD in a canonical nonlinear setting (phase retrieval), through a mechanism (data reuse over many passes) that is fundamentally different from the two-pass loss modification identified in prior work. We believe this is a clean and informative finding, but we do not claim it extends automatically to all nonlinear models.

---

### Official Review · Reviewer_SY96 · 2026-03-08

**Soundness:** 4
**Presentation:** 4
**Significance:** 3
**Originality:** 3
**Overall Recommendation:** 5
**Confidence:** 3

**Summary:**

This paper studies the sample complexity of learning a quadratic (information exponent 2) single index model with full-batch gradient descent and compares it to the sample complexity of online SGD as studied by Ben Arous et al [1]. The analysis of [1] shows that the sample complexity scales like $d\log(d)$. The paper shows that by modifying the activation function, it is possible to obtain a sample complexity of order $d$ i.e
\begin{equation}\tag{SGD}
n\simeq d\log(d),
\end{equation}
and this paper obtains the following guarantee
\begin{equation}\tag{Full-batch GD}
n\simeq d.
\end{equation}
To do so, the paper splits the analysis into two parts:
* Proving that the landscape is actually composed of a unique local minimizer (which is the desired $\theta^{*}$) when $n/d$ is constant.
* Giving some guarantees on the time required for the optimization dynamics to actually converge.

The main observation is that in this setup, the gradient descent is performing
\begin{equation} \frac{d}{dt}\theta(t)=P_{\theta(t)}A^{*}\theta(t),\end{equation}
which is reminiscent of a power method. So, one expect to converge to the top eigenvector of the matrix, and so, this paper studies the top eigenvector of this matrix when n,d are large.

The second part uses a decomposition into two time scales of the dynamics, which is interesting on its own. For using this two-time scales decompositon, the initialization is taken very small $d^{-15}$.

**Compliance With Llm Reviewing Policy:**

Affirmed.

**Final Justification:**

I believe my concerns have been adequately addressed by the rebuttal. I keep my score.

**Key Questions For Authors:**

* __Tensor power method__ The computationally best method to learn a single index with information exponent $k^{*}$ is to use the __partial-trace__ algorithm from [1] or a tensor PCA-like algorithm in the spherical case and multi-index case [2,3,4]. For example, taking $\sigma(z)=z^{k}$, we obtain that the gradient should be like
\begin{equation}
\nabla \mathcal{L}(\theta)=\frac{k}{n}\sum_{i=1}^{n}y_i x_i \langle x_i,\theta\rangle^{k-1}=T_{x,y}[\theta,\ldots,\theta].
\end{equation}
which suggests a contraction of an empirical order-$(k+1)$ tensor against $\theta^{\otimes (k-1)}$. Is it possible to formalize the early GD dynamics as a tensor power iteration on a tensor related to the partial-trace / tensor-unfolding estimators known to be optimal in these models

* __Noiseless Model__ Which parts of your analysis are specific to the noiseless model $y_i=\sigma(\langle x_i,\theta^*\rangle)$, and which ones can be adapted to the noisy model i.e $y_i=\sigma(x_i)+\varepsilon$? Since the strong-recovery result in Section 4 proves exact recovery in the realizable setting from small initialization, which steps of that argument fundamentally use a noiseless model? For instance, does the initial alignment phase survive under noise while the final geometric-contraction phase breaks down?

* __Generalization of the proof__ Can your proof techniques for the decoupling of the two phases be applied to [1], or is it too problem-dependent?

[1]  Berthier, Montanari and Zhou 'Learning time-scales in two-layers neural networks'

**Limitations:**

yes

**Strengths And Weaknesses:**

## Strengths
* The paper is well-written and clear.
* The paper provides the first benefits or reusing data without changing the training in a somewhat 'adhoc' way.
* The spectral/landscape analysis is interesting in its own right
* The optimization analysis in Section 4 appears technically novel, in particular the two-phase argument leading to a logarithmic iteration bound.

## Weakness
* The paper mixes two optimization settings: spherical gradient flow on the correlation loss in Section 3, and Euclidean GD on the squared loss in Section 4. While the switch is motivated by the need to learn the norm from small initialization, it makes it harder to isolate exactly which ingredient is responsible for the improvement: data reuse, truncation of the activation, the change of loss, the change of geometry, or the small-initialization regime. I would appreciate a clearer discussion of how these two results fit together and to what extent the Section 4 guarantees should be viewed as a continuation of the Section 3 story rather than a separate result.

---

> ### Author Rebuttal · Authors · 2026-03-30
>
> We thank the reviewer for appreciating our work and for the insightful comments. As for the weakness related to the different setups in Section 3 and 4, we expect several of these differences to be purely technical (in particular gradient descent vs. gradient flow, and hard truncation vs. smooth truncation), see our response to Reviewer prTm. We answer the three questions below.
>
> **Q1.** This is a great point. The main question is what is the first derivative of the activation that is not 0 at 0. Here, we focus on the second derivative because the activation is quadratic. If the activation is a monomial of degree $k$, then the early GD dynamics can be formalized as power iteration on the $k$-tensor, exactly as suggested by the reviewer. While we expect the analysis in this case to be substantially harder, one could certainly draw inspiration from the rich literature on tensor power methods.
>
> **Q2.** As long as the noise is additive and independent from the signal, we expect the analyses of both Theorem 3.2 and 4.1 to go through. As for Theorem 3.2, the fixed matrix $A^*$ still exhibits a BBP transition under the additive noise model. As for Theorem 4.1, one can incorporate noise in the analysis of the GD trajectory. The idea is to view the noise as inducing an additive perturbation of the gradient.  Since the first phase is short, this additive perturbation cannot increase the final accuracy much more than the noise floor. The final phase is based on a contraction or convergence argument. Thus, even if there are infinitely many steps, the additive perturbation can at most increase the error induced by noise by a constant factor. That said, while we expect this general strategy to work, this still requires careful technical arguments which are beyond the scope of the current paper.
>
> **Q3.** As mentioned by the reviewer, our approach shares with [1] the division in two phases/time-scales. However, we note that the model being fit in [1] (two-layer neural network in the mean-field regime) is quite different from the one considered in our work (single-index model, matched to the data model). We believe the general spirit of our multi-phase analysis to be in principle helpful for this problem as well, but clearly the phases are different and the population landscape is of a different nature, so this would require significant modifications. However, our concentration arguments can still be quite helpful and, in particular, we think that uniform concentration of the gradient around its population can be achieved for such a one-hidden layer model using a strategy similar to the one put forward in our work.
>
> We would be happy to answer any follow-up questions in the discussion period.

---

> > ### Author Rebuttal · Reviewer_SY96 · 2026-04-01
> >
> > Thank you to the authors for your thoughtful responses. I am satisfied with your answers to Questions 1 and 3. Concerning Question 2, I do not yet have sufficient familiarity with the underlying techniques to be fully convinced, but this was primarily a matter of curiosity and does not affect my overall evaluation of the paper.

---

### Official Review · Reviewer_Cy18 · 2026-03-11

**Soundness:** 4
**Presentation:** 3
**Significance:** 2
**Originality:** 2
**Overall Recommendation:** 2
**Confidence:** 3

**Summary:**

This paper studies the sample complexity and convergence rate for one-pass SGD and full-batch GD on isotropic Gaussian single-index models with information exponent 2, focusing the discussion on the phase retrieval problem $$.

The main results are:

- No sample-complexity separation for weak recovery with correlation loss: The authors show that both spherical gradient descent and one-pass SGD require $n \gtrsim d\log d$
  samples in order to achieve weak recovery with correlation loss. In particular, full-batch GD does not improve the statistical efficiency relative to one-pass SGD in this case.

- Improved efficiency via truncated activations: When the activation function is truncated, the optimization landscape changes qualitatively. In this case, full-batch gradient flow weakly recovers with $n \gtrsim d$ samples, removing the $\log d$ factor and outperforming one-pass SGD with the same activation.

- Strong recovery with square loss: The authors show that full-batch GD on the square loss with truncated square actition weakly recovers with $n=\Theta(d)$, retaining the $\log{d}$ time complexity.

Overall, the paper provides a rigorous analysis of the separation between full-batch GD/GF and one-pass SGD in a nonlinear high-dimensional model. The results highlight how subtle changes in the link function can qualitatively modify the optimization landscape and lead to a statistical separation.

**Compliance With Llm Reviewing Policy:**

Affirmed.

**Key Questions For Authors:**

- The definition of strong recovery eq. (2.1) needs to be fixed, $\theta_{\star}$ is not exactly identifiable in phase retrieval due to the $Z_{2}$ symmetry.
- In L050-L053 (right), this discussion is specific to one-pass. Would be better to precise.

**Limitations:**

Limitations are discussed in the *Concluding remarks*.

**Strengths And Weaknesses:**

**Strengths**: The paper is clearly written and is easy to follow. It addressess a question of interest to the ICML community, notably the statistical advantage of batch reusing in descent-based algorithms.

**Weaknessess**: The major problem in this paper is that most of the core questions put forward in the introduction as a motivation have been addressed in previous literature. At first, I thought this was due to the lack of knowledge of the related papers. However, while some important references are indeed missing, the authors do seem to be aware of at least an important part of the related literature, deciding however to discuss it only in the related works.

While it is common practice to put forward your own results and its relevance, the extent in which this is done in this work in detriment of the related literature, is in my opinion unacceptable. This is a pity, since the technical results on the paper are novel and of potential interest. Nevertheless, in its current form the paper do require a major rewritting. Below, I list the major points leading to my assessment.

- Statistical advantages of batch reusing: The statistical separation between one-pass SGD and other first order methods that reuse data in non-linear, Gaussian single-index models is well understood and documented:
    - The first instance of this (as the authors acknowledge in the related works) is that weak recovery can be achieved with $n=\Theta(d)$ with both AMP and spectral methods (Mondelli & Montanari, 2018; Lu & Li 2020).
    - In the context of full-batch GD, (Sarao Mannelli et al., 2020b) proved that a large-width two-layer neural network fully achieves full recovery in the phase retrieval problem with $n=2d$ and $T=O(\log{d})$. In exactly the same setting, (Arnaboldi et al., 2023) showed that one-pass SGD requires $n=O(d\log d)$, a result which also appeared in (Martin et al., 2024). Therefore, the statistical separation, both in weak and strong recovery, between one-pass SGD and full-batch GD for a wide quadratic NN learning in the phase retrieval target is well-documented. The authors seem aware of this since most of these papers are cited in the text, without explicitly discussing these results.
    - (Dandi et al., 2024) showed that a statistical separation for weak-recovery between full-batch GD and one-pass SGD already appears at the second-step in Gaussian single-index models with odd activation function (e.g. ${\rm He}_{3}$). Again, this paper is cited without the result being explicitly discussed.
    - (Mignacco et al., 2021) showed that persistent SGD -- multi-pass SGD where sample is kept in the mini-batch for a random time -- strongly solves phase retrieval in the proportional asymptotics, while GD in the same limit fails.

- $T = O(\log{d})$ convergence rate: The fact that full-batch GD does not improve on the convergence rate is also well-documented:
    - Bayes-optimal AMP -- which is the optimal first order method -- fails to weakly solve phase retrieval with $n=\Theta(d)$ and $T=\Theta(1)$ steps. Since full-batch GD is a first order method, it is no surprise that it also fails in the same regime. Rigorous side information results (Barbier et al., 2019), as well as random initialization results for AMP in similar problems (Li et al., 2023), also widely support that $T=O(\log(d))$ is necessary. Outside of the escope of AMP.
    - As already mentioned above, (Sarao Mannelli et al., 2020b) showed that $T=O(\log{d})$ is required for a quadratic NN to strongly solve phase retrieval with full-batch GD with $n=2d$ samples.

- Truncation helps: (Bonnaire et al., 2024) showed that regularizing the Hessian of the phase retrieval loss at initialization, with a construction very similar to the soft-truncation of Section 3.2, allows to weakly solve the problem in linear time, with a BBP analysis.

Given this extensive body of literature, I believe sentences such as:

- L014-L019 (left):
> *However, beyond linear regression, the theoretical advantage of fullbatch gradient descent (GD, which always reuses all the data) over one-pass stochastic gradient descent (online SGD, which uses each data point only once) remains unclear.*
- L012-L014 (right):
> *Despite this progress, it remains poorly understood whether comparable multi-pass speedups occur in nonlinear models and, if so, what mechanisms drive them*
- L072-074 (left):
> *Notably, most existing analyses of full-batch GD (and its variants) still require n ≳ d polylog d, and therefore do not imply a statistical separation from one-pass SGD*
- L373-376 (left):
> *To our knowledge, this is the first convergence rate and strong recovery guarantee for learning single-index models with information exponent 2 using standard full-batch GD in the proportional (finite n/d) regime.*

cannot be really justified. For this reason, I am not able to support this manuscript for publication in its current format. I would strongly encourage the authors to make a major revision to adjust these claims and include a detailed discussion of how each result in each section relates to what is known for this problem.

n.b.: Note that I am not claiming the technical results in this paper appeared in previous work, but the general claims in their motivation and discussion are not grounded in their current format.

- (Arnaboldi et al., 2023) Escaping mediocrity: how two-layer networks learn hard generalized linear models with SGD, https://arxiv.org/abs/2305.18502
- (Mignacco et al., 2021) Stochasticity helps to navigate rough landscapes: comparing gradient-descent-based algorithms in the phase retrieval problem. MLST 2021
- (Li et al., 2023) Approximate message passing from random initialization with applications to Z2 synchronization, PNAS 2023
- (Bonnaire et al., 2024) The Role of the Time-Dependent Hessian in High-Dimensional Optimization, https://arxiv.org/abs/2403.02418

---

> ### Author Rebuttal · Authors · 2026-03-30
>
> We thank the reviewer for acknowledging that "the technical results on the paper are novel and of potential interest." However, we respectfully disagree with the central claim that the motivating questions of our paper have been addressed in previous literature, and we stand by our main claims (e.g., those in l. 12-14, 14-19, 72-74, 373-376 flagged in the review). We address each major point below.
>
> *Statistical separation between one-pass SGD and other first order methods that reuse data in non-linear, Gaussian single-index models is well understood*
>
> Our goal is not to show a statistical separation between online SGD and *other first-order methods* such as AMP. As stated in the last paragraph on page 1, it is well-known that certain first-order methods achieve linear-in-$d$ sample complexity in phase retrieval, but it remains unclear whether full-batch GD from random initialization removes the log-factor. Thus, we study sample and time complexity of full-batch GD and elucidate the conditions under which multi-pass provides statistical benefit, see l. 66-70.
>
> *Statistical separation … between one-pass SGD and full-batch GD for a wide quadratic NN learning in the phase retrieval target is well-documented*
>
> This is incorrect. The positive result for empirical risk minimization in [Sarao Mannelli et al., 2020] requires $m>d$ student neurons and does not cover $m=1$. [Arnaboldi et al., 2023] consider an SDE approximation that takes $d\to\infty$ first; based on our reading, this result does not give sharp sample complexity of online SGD in learning one quadratic neuron when $m>d$ — in fact, their chosen learning rate $\eta \sim 1/(md)$ implies $n \gtrsim d^2$ sample size which is clearly suboptimal. [Martin et al., 2024] only studies the time complexity of population gradient flow with no statistical guarantee. To our knowledge, none of these results — individually or in combination — imply a statistical separation between online SGD and full-batch GD for learning a single quadratic neuron. We will clarify this in the revision.
>
> *Comparison with [Dandi et al. 2024]*
>
> This mechanism is discussed in l. 137-148: reusing the same data point twice induces a label preprocessing that may lower the information exponent. This does not apply to our phase retrieval setting (with generative exponent 2), and moreover, we argue that this benefit is primarily computational: in settings where [Dandi et al. 2024] established a sample complexity separation between single-pass and full-batch update, single-pass SGD on a modified loss achieves the same statistical efficiency. This contrasts with our setting, where the advantage of full-batch GD is genuinely statistical and cannot be replicated by loss modification.
>
> *Comparison with [Mignacco et al., 2021]*
>
> While this paper also studies multi-pass updates, the DMFT analysis assumes a warm start and does not cover the initial search phase. Crucially, after a warm-start is achieved, the descent phase for online SGD takes $O(d)$ samples regardless of the information exponent – see Theorem 1.5 in [Ben Arous et al. 2021]. Therefore, establishing a separation (beyond constant factors) between online and full-batch updates requires analyzing dynamics from random initialization, which [Mignacco et al., 2021] does not do. We will discuss this in the revision.
>
> *$T \sim \log d$ convergence rate*
>
> Our goal is not to show that the $\log d$ convergence rate is unimprovable. Rather, we rigorously establish an **end-to-end** sample and runtime complexity guarantee for full-batch GD achieving **strong recovery** from **random initialization** — to our knowledge, no such result exists in the literature: sharp analyses of GD trajectories in the proportional limit (e.g., via DMFT) are typically rigorous only at finite time horizon, and none of the references apply to the well-specified phase retrieval problem ($m=1$) that we study, but rather focus on AMP and overparameterized networks.
>
> *Truncation and [Bonnaire et al., 2024]*
>
> We do not claim truncation is a novel modification — prior works on phase retrieval have considered similar loss truncation (e.g., [Chen & Candès, 2017]), though mostly in local rather than random-initialization settings. More importantly, to our knowledge, it has not been clarified whether such modification is an artifact of the analysis or necessary for optimization to succeed. In spherical gradient setting, we prove truncation is indeed necessary: for unbounded quadratic activation, dynamics converge to an uninformative fixed point at finite $n/d$ (Thm. 3.1). We also note that certain derivations in [Bonnaire et al., 2024] are mathematically incorrect — see concurrent work [Montanari and Wang, 2026] for details.
>
> We finally thank the reviewer for the questions: (i) we will fix (2.1) mentioning the sign symmetry; and (ii) we will clarify that the discussion in l. 50-53 is for one-pass SGD.
>
> We would be happy to answer any follow-up questions in the discussion period.

---

> > ### Author Rebuttal · Reviewer_Cy18 · 2026-04-03
> >
> > I thank the authors for their rebuttal. However, I am concerned that there may have been a misunderstanding regarding my main point. If my initial comments were unclear, I apologize and attempt to restate them more precisely below.
> >
> > I have carefully read the paper and recognize the technical contributions it makes relative to the existing literature; this is explicitly acknowledged in my original review.
> >
> > My primary concern instead pertains to **how these contributions are positioned** within the broader phase retrieval literature. In this respect, the rebuttal reinforces my impression that the authors are indeed familiar with the relevant body of work. It is therefore surprising that they maintain the claims made in lines 12–14, 14–19, 72–74, and 373–376 without further qualification. In their current form, these statements appear imprecise and would benefit from clarification and contextualization.
> >
> > - L014-L019 (left):
> > > However, beyond linear regression, the theoretical advantage of fullbatch gradient descent (GD, which always reuses all the data) over one-pass stochastic gradient descent (online SGD, which uses each data point only once) remains unclear.
> >
> > (Dandi et al. 2024) shows that $f_{\star}(x)=he_{3}(\langle \theta_{\star},x\rangle)$ can be weakly learned with GD with $n=\Theta(d)$, while by (Ben Arous et al., 20219) with one-pass SGD one needs $n=\Theta(d^{2})$. This goes beyond linear regression, and the result is very clear.
> >
> > - L012-L014 (right):
> > > Despite this progress, it remains poorly understood whether comparable multi-pass speedups occur in nonlinear models and, if so, what mechanisms drive them
> >
> > (Arnaboldi et al., 2024; Lee et al., 2024) clearly showed the advantage of multi-pass over single-pass in non-linear models, and extensively discussed the mechanism driving it.
> >
> > - L072-074 (left):
> > > Notably, most existing analyses of full-batch GD (and its variants) still require $n \geq d polylog d$, and therefore do not imply a statistical separation from one-pass SGD
> >
> > (Sarao Mannelli et al.,2020b) showed that full-batch GD requires $n=\Theta(2d)$ to learn the phase retrieval target with an overparametrized network, showing a statistical separation from the one-pass SGD result of (Arnaboldi et al., 2023; Martin et al., 2024).
> >
> > - L373-376 (left):
> > > To our knowledge, this is the first convergence rate and strong recovery guarantee for learning single-index models with information exponent 2 using standard full-batch GD in the proportional (finite n/d) regime.
> >
> > Again, (Sarao Mannelli et al.,2020b) showed that learning the phase retrieval target (which is a single-index model with IE 2) with standard full-batch GD in the proportional limit is possible.
> >
> > Taken together, these examples suggest that the above statements require additional qualification to be accurate. At their current level of generality, they risk being misleading. I would be willing to reconsider my assessment if the authors provide a clear list of revisions addressing these points and appropriately situate their contributions within the existing literature.
> >
> > Additionally, although not directly relevant to the paper, I got confused by something the authors wrote in their rebuttal:
> > > We also note that certain derivations in [Bonnaire et al., 2024] are mathematically incorrect — see concurrent work [Montanari and Wang, 2026] for details.
> >
> > Can the authors be more precise of what is mathematically incorrect in Bonnaire et al., (2024)? Given the strength of the criticism directed at a published work, it would be important -- particularly if this discussion is made public -- to clearly substantiate and delineate this claim. As suggested by the authors, I have consulted the discussion in Montanari and Wang (2026) concerning Bonnaire et al., (2024):
> >
> > > Namely, they assume that gradient descent converges to ‘marginal states’, and use the replica method together
> > with random matrix theory to study spectrum of the Hessian in these states. While providing the correct picture, we point out that this type of analysis of gradient descent is not —in general— asymptotically exact.
> >
> > However, I found this discussion to be somewhat vague and ultimately inconclusive; it does not clearly identify which specific aspects, if any, are mathematically incorrect.

---

> > > ### Author Response · Authors · 2026-04-06
> > >
> > > We thank the reviewer for a continued discussion. We comment on refs raised above and propose clarifications for the final version. All these clarifications are minor text changes, and we reiterate that we already cited and discussed all references mentioned above by the reviewer in the submitted manuscript.
> > >
> > > ---
> > >
> > > Regarding (Dandi et al., 2024), as highlighted in our previous reply, we explicitly discuss this mechanism in l. 137-144 (right), explaining why it does not apply to our setting. This mechanism, also shared by (Lee et al., 2024; Arnaboldi et al., 2024), is primarily computational: in settings where all three works establish a statistical separation between single-pass and full-batch updates, single-pass SGD on a modified loss achieves the same efficiency. This contrasts multi-pass analyses in linear regression, where the statistical benefit comes from seeing data points more than once and cannot be achieved by loss modification. We will move this discussion from Section 1.2 to the first page, since the reviewer feels it is not sufficiently emphasized.
> > >
> > > Furthermore, (Dandi et al., 2024) use each data point *exactly twice*, and this is not enough to achieve weak recovery for an even activation (and hence for the quadratic link we consider). Similarly, (Lee et al., 2024; Arnaboldi et al., 2024) also use each data point exactly twice, which cannot explain the advantage of multi-pass ($>2$ passes) over single-pass in nonlinear models. To qualify the scope of our work, we will modify l. 14-19 as follows: "However, beyond linear regression and **a loss modification mechanism achieved by two passes on the data**, the theoretical advantage of **multi-pass** full-batch gradient descent (GD, which always reuses all the data) over one-pass stochastic gradient descent (online SGD, which uses each data point only once) remains unclear."
> > >
> > > More generally, the statistical advantages of full-batch updates and how they depend on problem structure (e.g., truncated vs. unbounded nonlinearity, spherical vs. Euclidean optimization) is not thoroughly investigated, which is why we state that the mechanism of multi-pass speedup in nonlinear models is not well understood. As this point created confusion, we will add the clarification currently in l. 137-144 to l. 11 (right): this amounts to a one-line introduction of the loss modification mechanism after the sentence on linear models.
> > >
> > > ---
> > >
> > > Regarding the sentence "Notably, most existing analyses of full-batch GD still require $n \gtrsim d\text{polylog}d$, and therefore do not imply a statistical separation from one-pass SGD", this description applies to (Netrapalli et al., 2013; Candès et al., 2015; Sun et al., 2018; Chen et al., 2019), cited immediately after. The next sentence (l. 76) explicitly acknowledges prior works showing "the optimization landscape can be benign at finite $n/d$," citing (Sarao Mannelli et al., 2020b). We do not think this constitutes overclaiming.
> > >
> > > More importantly, as highlighted in our previous replies, the positive result of (Sarao Mannelli et al., 2020b) holds in the overparameterized regime ($m > d$), a different setting from the online SGD analysis of (Arnaboldi et al., 2023) — in fact, taking their prescribed learning rate for $m > d$ yields $n \gtrsim d^2$ which is clearly suboptimal. Furthermore, (Martin et al., 2024) study population dynamics and do not present any sample complexity for online SGD. Hence, **combining (Sarao Mannelli et al., 2020b) with (Arnaboldi et al., 2023; Martin et al., 2024) does not yield a statistical separation** between full-batch GD and online SGD, as settings are incomparable. Statistical separation can only be claimed when examining the same model and algorithm, as in our upper bound on gradient flow with correlation loss together with the lower bound on the same setting from (Ben Arous et al., 2021).
> > >
> > > ---
> > >
> > > We stand by our exact claim that "to our knowledge, this is the first convergence rate and strong recovery guarantee for learning single-index models with information exponent 2 using standard full-batch GD in the proportional (finite $n/d$) regime."
> > >
> > > Note that (Sarao Mannelli et al., 2020b) presents a positive landscape result at finite $n/d$ (explicitly acknowledged in l. 78), but does not study the learning dynamics on the **empirical landscape**: **the convergence rate analysis only applies to the population loss (corresponding to $n=\infty$) and not to finite $n/d$**. However, to avoid any possible misunderstanding, we will further emphasize that our result gives the first end-to-end convergence rate for **GD training on the empirical loss** (i.e., finite $n/d$).
> > >
> > > ---
> > >
> > > Finally, (Bonnaire et al., 2024) ignore an interaction term between the data and parameters, see the "rest" block defined in the lower right of Equation (85) of (Montanari and Wang, 2026), which is characterized via DMFT. This is orthogonal to our contributions, so we did not expand for space reasons.

---

### Official Review · Reviewer_prTm · 2026-03-13

**Soundness:** 3
**Presentation:** 3
**Significance:** 3
**Originality:** 3
**Overall Recommendation:** 5
**Confidence:** 3

**Summary:**

The paper studies the sample-complexity benefits of full-batch gradient descent over single-pass SGD in learning a single-index model with quadratic activation by (again) a single neuron with quadratic activation and correlation loss . In particular, the paper studies weak recovery and strong recovery for this data model for the spherical gradient flow and by comparing their results with known SGD lower-bounds, obtain that: (1) full-batch GD alone does not lead to improved sample complexities for weak recovery from a random initialization (Thm 3.1).  (2) By smoothly truncating the activation, it is possible to improve the previous sample complexity for full-batch GD to $n=\Theta (d)$ (Thm 3.2)

**Compliance With Llm Reviewing Policy:**

Affirmed.

**Final Justification:**

The authors' response was sufficient. I believe this work can be of potential interest to the community. Therefore I recommend acceptance.

**Key Questions For Authors:**

Please see weaknesses in the last section.

**Limitations:**

yes

**Strengths And Weaknesses:**

Strengths:

- The separation between SGD and full-batch methods is an interesting and timely research direction which is becoming popular recently. Although the results apply to a very specific setting (quadratic activation, single neuron, correlation loss etc), it strongly shows the benefit of joint full-batch GD and the architectural design (activation consideration) for improving the sample complexity. Earlier work on online SGD for information-exponent-2 single-index models shows the $n=d~ log(d)$ barrier, while spectral methods and AMP were already known to achieve linear sample complexity. What was missing was a proof that plain full-batch GD itself can close that gap in a nonlinear model. The paper is also sound and presented well and the proof sketch for different theorems highlight the main steps of the proof and contributions.

Weaknesses:

- The sample-complexity separation is proved for spherical gradient flow on the correlation loss with a smooth truncation, whereas the strong-recovery and runtime result is proved for Euclidean GD on the squared loss with a hard truncation. It remains unclear how can these results be unified.

- It is unclear how the results or approach extend to other data assumptions. The truncation is doing a lot of work, and the robustness of the phenomenon beyond this setting remains unclear. For example, can other activations or architectures learn this model with a similar sample complexity?

---

> ### Author Rebuttal · Authors · 2026-03-30
>
> We thank the reviewer for the positive evaluation of our work. We discuss below the two weaknesses.
>
> **W1.** We opted to start with spherical gradient flow and correlation loss, since this is the same setup for the lower bound of [Ben Arous et al., 2021]. The choice of gradient descent and square loss is motivated by our analysis in two phases (first the angle shrinks while the truncation does not occur and, at the same time, the norm grows; then the error decreases geometrically as we are in the vicinity of the optimum), as well as by prior works on GD dynamics under small initialization. This being said, we expect several of the differences in the settings to be purely technical:
>
> (i) We expect all results to hold under either a smooth or a hard truncation. The hard truncation in Theorem 4.1 simplifies some of the population arguments (reducing some algebraic manipulations) and simplifies concentration arguments involved in the analysis of gradient descent with square loss. With some additional work, a variety of truncations can however be accommodated. We decided to focus on this setting for clarity of exposition of the core technical ideas. The smooth truncation in Theorem 3.2 is needed to guarantee well-posedness and adequate regularity of the solution to the gradient flow with correlation loss, and it simplifies the application of the stable manifold theorem in the analysis. We speculate that a similar conclusion holds for gradient descent with correlation loss and an activation with a discontinuous derivative.
>
> (ii) We also expect the difference between gradient descent (with small enough step size) and gradient flow to be negligible (and that such a difference can be explicitly tracked with a more detailed analysis).
>
> **W2.** Due to space constraints, let us focus our discussion on a more general class of activations. We first note that we are interested in activations such that the sample complexity of online SGD is at least $d\log d$. This covers e.g. all even activations, and the most natural choice is the quadratic function.
> Having said that, we conjecture that our positive results hold more generally as long as the activation function is bounded. The necessity of a bounded activation comes directly from the argument of Theorem 3.1: informally, if the bulk of the spectrum of the random matrix on which we are doing power iteration diverges, then a super-linear sample size is required for a BBP transition (and, hence, for weak recovery) to occur.
>
> To show weak recovery for gradient flow with correlation loss and a general activation, the challenge is to prove a (uniform-in-$\theta$) BBP transition for the matrix $A(\theta)=\frac{1}{n}\sum_{i=1}^n y_i \frac{\sigma’(\langle w_i, \theta\rangle)}{\langle w_i, \theta\rangle}x_i x_i^T$. Recall that $y_i=\sigma(\langle w_i, \theta^* \rangle)$. Thus, the problem can be related to showing a BBP transition for a multi-index model (the two indices corresponding to $\theta$ and $\theta^*$). A precise characterization for such phase transitions has been developed in [Kovacevic et al., 2025; Defilippis et al., 2025], so this would be the starting point of the analysis.
>
> Next, to show strong recovery for gradient descent with square loss and a general activation, one can use a strategy in two phases similar to that employed to prove Theorem 4.1: the first phase starts from a small initialization so that the parameter aligns in the right direction before the norm grows and the truncation becomes active; and the second phase refines the estimate via a concentration argument. We expect most of the concentration arguments to hold for a general activation, but substantial technical work is required, as well as additional assumptions on the activation (e.g., bounded higher order derivatives). The first phase is a bit more delicate, but we also expect it to go through as long as the second derivative of the activation at $0$ is $>0$. We regard this as an interesting future direction which is however out of the scope of the current contribution.
>
> We would be happy to answer any follow-up questions in the discussion period.

---

> > ### Author Rebuttal · Reviewer_prTm · 2026-04-01
> >
> > Thank you for your response. I believe this work is potentially interesting for the community. I increase my score to accept.

---

### Decision · Program_Chairs · 2026-04-30

**Decision:**

Accept (regular)

**Comment:**

The paper studies gradient-based learning in a quadratic single-index model, with a focus on when reusing data can provide an advantage over one-pass SGD. The main technical contributions are to show that, in the plain quadratic setting, full-batch gradient methods do not improve the weak-recovery sample complexity relative to one-pass SGD, whereas with a truncated activation, they can achieve weak recovery with a sample complexity linear in the dimension. The paper also provides a strong recovery guarantee together with a convergence-rate analysis from small initialization. Overall, the reviewers agreed that the paper presents technical results that may be of interest to the ICML community.

The main discussion concerned how these contributions should be positioned relative to prior work. In particular, the discussion clarified that the most precise contribution is in the well-specified single-index setting considered here. The reviewers also noted that some earlier works address related but not identical mechanisms, for example through two-pass or loss-modification effects rather than the precise full-batch multi-pass setting studied here (Dandi et al. '24). Likewise, part of the discussion focused on distinguishing the present results from prior positive results in broader or different model classes, such as overparameterized networks (Sarao Mannelli et al. '20). At the same time, there was broad agreement that the technical results themselves are novel, but that the introduction and discussion should be more carefully positioned and narrower in scope than in the current version.

This is, therefore, a somewhat borderline call. Reviewer Cy18 remained unsatisfied with the currently proposed changes, and I think these concerns should be taken seriously. In particular, the final version should more carefully qualify broad statements about whether the advantage of multi-pass full-batch gradient methods in nonlinear models remains unclear, should make explicit that the main novelty is in the specific well-specified single-index setting studied here, and should revise the comparison to prior work so that distinctions in model, mechanism, and scope are stated precisely, including with respect to (Dandi et al. '24) and (Sarao Mannelli et al. '20). On balance, I believe the technical contribution is strong enough to support acceptance, but I encourage the authors to carefully reconsider these points and reformulate the disputed statements in a more precise and measured way.